# The ubiquitin ligase UBR5 suppresses proteostasis collapse in pluripotent stem cells from Huntington's disease patients

Seda Koyuncu [1], Isabel Saez[1], Hyun Ju Lee[1], Ricardo Gutierrez-Garcia[1], Wojciech Pokrzywa[1,2], Azra Fatima[1], Thorsten Hoppe [1] & David Vilchez [1]

Induced pluripotent stem cells (iPSCs) undergo unlimited self-renewal while maintaining their potential to differentiate into post-mitotic cells with an intact proteome. As such, iPSCs suppress the aggregation of polyQ-expanded huntingtin (HTT), the mutant protein underlying Huntington's disease (HD). Here we show that proteasome activity determines HTT levels, preventing polyQ-expanded aggregation in iPSCs from HD patients (HD-iPSCs). iPSCs exhibit high levels of UBR5, a ubiquitin ligase required for proteasomal degradation of both normal and mutant HTT. Conversely, loss of UBR5 increases HTT levels and triggers polyQ-expanded aggregation in HD-iPSCs. Moreover, UBR5 knockdown hastens polyQ-expanded aggregation and neurotoxicity in invertebrate models. Notably, UBR5 overexpression induces polyubiquitination and degradation of mutant HTT, reducing polyQ-expanded aggregates in HD-cell models. Besides HTT levels, intrinsic enhanced UBR5 expression determines global proteostasis of iPSCs preventing the aggregation of misfolded proteins ensued from normal metabolism. Thus, our findings indicate UBR5 as a modulator of super-vigilant proteostasis of iPSCs.

[1] Institute for Genetics and Cologne Excellence Cluster for Cellular Stress Responses in Aging-Associated Diseases (CECAD), University of Cologne, Joseph Stelzmann Strasse 26, 50931 Cologne, Germany. [2] Laboratory of Protein Metabolism in Development and Aging, International Institute of Molecular and Cell Biology, Trojdena Street 4, 02-109 Warsaw, Poland. These authors contributed equally: Seda Koyuncu, Isabel Saez. Correspondence and requests for materials should be addressed to D.V. (email: dvilchez@uni-koeln.de)

As the origin of multicellular organisms, a series of cellular quality control mechanisms must operate at high fidelity in pluripotent stem cells[1]. In culture, embryonic stem cells (ESCs) derived from blastocysts do not undergo senescence and can replicate indefinitely while maintaining their capacity to differentiate into all cell lineages[2]. Alternatively, somatic cells can be reprogrammed to generate induced pluripotent stem cells (iPSCs), which are similar to ESCs in many respects, such as their gene expression, potential for differentiation and ability to replicate continuously[3]. This unlimited self-renewal capacity requires stringent quality control mechanisms, including increased DNA damage responses and antioxidant defense systems[1,4–7]. Growing evidence indicates that pluripotent stem cells also have intrinsic mechanisms to maintain the integrity of the proteome, a critical process for organismal development and cell function[7–9]. Hence, defining the mechanisms of super-vigilant proteostasis in these cells is of central importance.

The proteostasis network is formed by multiple integrated processes that control the concentration, folding, location and interactions of proteins from their synthesis through their degradation[10]. Defects in proteostasis lead to the accumulation of damaged, misfolded and aggregated proteins that may alter the immortality of pluripotent stem cells. During the asymmetric divisions invoked by these cells, the passage of damaged proteins to progenitor cells could compromise organismal development and aging. Thus, pluripotent stem cells have a tightly regulated proteostasis network linked with their intrinsic characteristics and biological function[1,7]. While ESC identity requires enhanced global translational rates[11], these cells also exhibit high levels of distinct chaperones to assure proper protein folding[5,9]. For instance, ESCs have increased assembly of the TRiC/CCT (T-complex protein-1 (TCP-1) ring complex)/(chaperonin containing TCP-1) complex[12], a chaperonin that facilitates the folding of approximately 15% of the proteome and reduces the aggregation of disease-related mutant proteins[13]. To terminate damaged proteins, ESCs possess a powerful proteolytic machinery induced by high levels of PSMD11/RPN6[14–16], a scaffolding subunit that promotes the assembly of active proteasomes[16,17].

Remarkably, pluripotent stem cells are able to maintain enhanced proteostasis while proliferating indefinitely in their undifferentiated state[1,7,8]. However, the differentiation process triggers a rewiring of the proteostasis network that reduces their ability to sustain proteome integrity[7–9]. In addition, post-mitotic and progenitor cells as well as somatic stem cells undergo a progressive decline in their protein folding and clearance activities with age[8,18,19]. This demise of proteostasis is linked with the onset of age-related disorders such as Alzheimer's, Parkinson's and Huntington's disease (HD)[10,18]. On the other hand, the proteostasis network of somatic cells is rewired during cell reprogramming to generate iPSCs with high assembly of active TRiC/CCT and proteasome complexes, resembling ESCs[9,12,16,20].

HD is a fatal neurodegenerative disorder characterized by cognitive deficits, psychosis and motor dysfunction. The disease is inherited in a dominant manner and caused by mutations in the *huntingtin* (*HTT*) gene, which translates into an expanded polyglutamine stretch (polyQ)[21]. The wild-type *HTT* gene encodes a large protein of approximately 350 kDa that contains 6–35 polyQ repeats. In individuals affected by HD, HTT contains greater than 35 polyQ repeats[21]. Although loss of normal HTT function could also be a determinant of HD[22], the dominant inheritance pattern of the disease and numerous experiments in model organisms indicate that gain of function of mutant HTT is toxic and induces neurodegeneration[21,23,24]. PolyQ-expanded HTT is prone to aggregation, and the accumulation of mutant HTT fibrils as well as intermediate oligomers formed during the aggregation/disaggregation process contributes to neurodegeneration[21,23,24]. The longer

the polyQ-expanded repeat, the earlier HD symptoms (e.g., neurodegeneration) typically appear[21]. However, the length of the pathological polyQ does not affect survival, self-renewal and pluripotency of iPSCs derived from HD patients (HD-iPSCs), which can proliferate indefinitely as control iPSCs[25,26]. Moreover, HD-iPSCs do not accumulate polyQ-expanded inclusions[12,25,27]. These findings indicate that iPSCs have increased mechanisms to maintain proteostasis of mutant HTT. Once differentiated into neural progenitors and neurons, these cells exhibit HD-associated phenotypes such as altered gene expression, increased vulnerability to excitotoxic stressors and cumulative risk of death over time[25,28]. However, HD neurons lack polyQ aggregates and robust neurodegeneration phenotype[12,25,27], supporting a proteostasis-rejuvenating process during cell reprogramming that allows for HD-iPSC differentiation into neurons with an intact proteome.

Although cumulative evidence indicates a strong link between HD-related changes and proteasomal dysfunction[29], the mechanisms by which the proteasome recognizes polyQ-expanded HTT are poorly understood. With the high levels of proteasome activity exhibited by iPSCs[16], we ask whether these cells have an intrinsic E3 ubiquitin ligase network to regulate proteostasis of polyQ-expanded HTT. We find that iPSCs have increased levels of UBR5, a HECT domain E3 enzyme that promotes proteasomal degradation of both mutant and wild-type HTT. Notably, an impairment of mutant HTT levels induced by UBR5 downregulation triggers the accumulation of polyQ-expanded aggregates in HD-iPSCs. Prompted by these findings, we examine whether modulation of UBR5 impinges on polyQ-expanded aggregation in distinct models. Since HD iPSC-derived neurons lack aggregates even upon proteasome inhibition, we assess *Caenorhabditis elegans* and human cells lines that accumulate polyQ-expanded aggregates. We find that loss of UBR5 worsens polyQ-expanded aggregation and neurotoxicity in *C. elegans* models. Notably, ectopic expression of UBR5 is sufficient to promote polyubiquitination of HTT, resulting in decreased levels and aggregation of mutant HTT in human cell models. Thus, we identify UBR5 as a potential modulator of HTT proteostasis by studying immortal pluripotent stem cells.

## Results

### The proteasome suppresses mutant HTT aggregation in iPSCs.
iPSCs derived from HD patients do not accumulate polyQ-expanded HTT aggregates[12,27]. Since pluripotent stem cells exhibit high proteasome activity[16], we assessed whether this activity is required to prevent mutant HTT aggregation in iPSCs generated from two individuals with juvenile-onset HD (i.e., Q71 and Q180)[25] (Supplementary Table 1). Indeed, downregulation of proteasome activity triggered accumulation of mutant HTT aggregates, which were mostly located in the cytoplasm (Fig. 1a and Supplementary Fig. 1a–b). Although the size of aggregates was generally smaller in iPSCs that express longer polyQ repeats (Q180), these cultures exhibited a higher percentage of aggregate-containing cells (Fig. 1a). Moreover, proteasome inhibition also induced a high percentage of mutant HTT aggregation in iPSCs derived from an individual with adult-onset HD (Q57) (Supplementary Fig. 1c–d). On the contrary, we did not detect accumulation of polyQ aggregates in three distinct control iPSCs upon proteasome inhibition (Fig. 1a and Supplementary Fig. 1e). Likewise, proteasome inhibition did not induce HTT aggregation in two isogenic counterparts of the Q180-iPSC line (Supplementary Fig. 2), in which the 180 CAG expansion was corrected to a nonpathological repeat length[30].

To further examine the link between proteasome activity and polyQ-expanded aggregation in iPSCs, we tested different concentrations of proteasome inhibitor. Lower concentrations

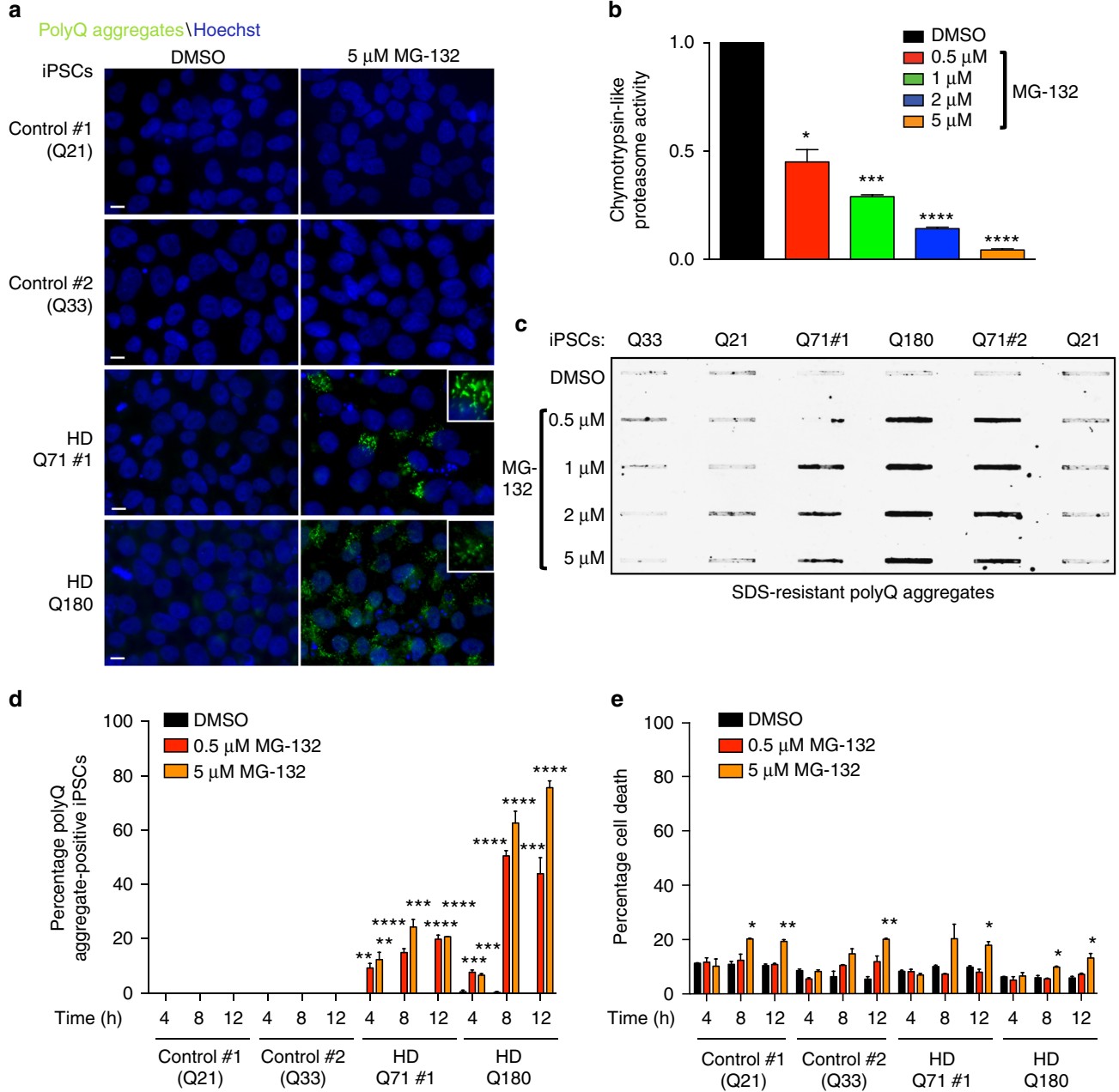

**Fig. 1** Proteasome inhibition triggers mutant HTT aggregation in HD-iPSCs. **a** Immunocytochemistry of control and HD-iPSC lines treated with 5 μM MG-132 for 12 h. We used an antibody against polyQ-expanded protein to detect mutant HTT aggregates. Cell nuclei were stained with Hoechst 33342. Scale bar represents 10 μm. The images are representative of six independent experiments. **b** Chymotrypsin-like proteasome activity in HD Q180-iPSC line treated with MG-132 for 12 h (relative slope to Q180-iPSCs treated with DMSO). Graph represents the mean ± s.e.m. of three independent experiments. **c** Filter trap analysis of the indicated control and HD-iPSCs. Proteasome inhibition with MG-132 for 12 h results in increased levels of polyQ aggregates in HD-iPSCs lines (detected by anti-polyQ-expansion diseases marker antibody). However, proteasome inhibition does not induce accumulation of polyQ aggregates in control iPSC lines. The images are representative of five independent experiments. **d** Graph represents the percentage of polyQ aggregate-positive cells/total nuclei in the indicated iPSC lines (mean ± s.e.m., 3 independent experiments, 300–350 total cells per treatment for each line). **e** Graph represents the percentage of propidium iodide-positive cells/total nuclei in the indicated iPSC lines (mean ± s.e.m., 3 independent experiments, 500–600 total cells per treatment for each line). For each time point, MG-132-treated lines were statistically compared with their respective DMSO-treated line. All the statistical comparisons were made by Student's t-test for unpaired samples. *P < 0.05, **P < 0.01, ***P < 0.001, ****P < 0.0001

reduced approximately 50% of the proteasome activity, which was sufficient to induce mutant HTT aggregation as assessed by both filter trap and immunofluorescence experiments (Fig. 1b–d and Supplementary Fig. 3). Since these analyses were performed in iPSCs treated with proteasome inhibitor for 12 h, we examined whether this treatment reduces cell viability, a process that could trigger proteostasis collapse and, in turn, dysregulation of protein

aggregation. Whereas different concentrations of proteasome inhibitor resulted in polyQ-expanded aggregation (Fig. 1c, d), only higher concentrations induced a mild increase (~8%) in cell death (Fig. 1e). In addition, we found accumulation of mutant HTT aggregates at an earlier time point (4 h) of the proteasome inhibition treatment when higher concentrations did not trigger cell death (Fig. 1d, e). Thus, these results indicate that a decline in

proteasome activity promotes mutant HTT aggregation in HD-iPSCs, a process that cannot only be explained by impairment of cell viability. Remarkably, control and HD-iPSCs exhibited similar sensitivity to proteasome inhibition (Fig. 1e), suggesting that mutant HTT aggregation does not induce cell death in iPSCs.

**Proteasome dysfunction impairs HTT levels in iPSCs.** With the link between proteasome inhibition and mutant HTT aggregation, we asked whether HTT levels are regulated by the proteasome in HD-iPSCs. For this purpose, we characterized in our model two antibodies that recognize HTT and polyQ-expanded proteins, respectively[12,31]. First, we validated that these antibodies

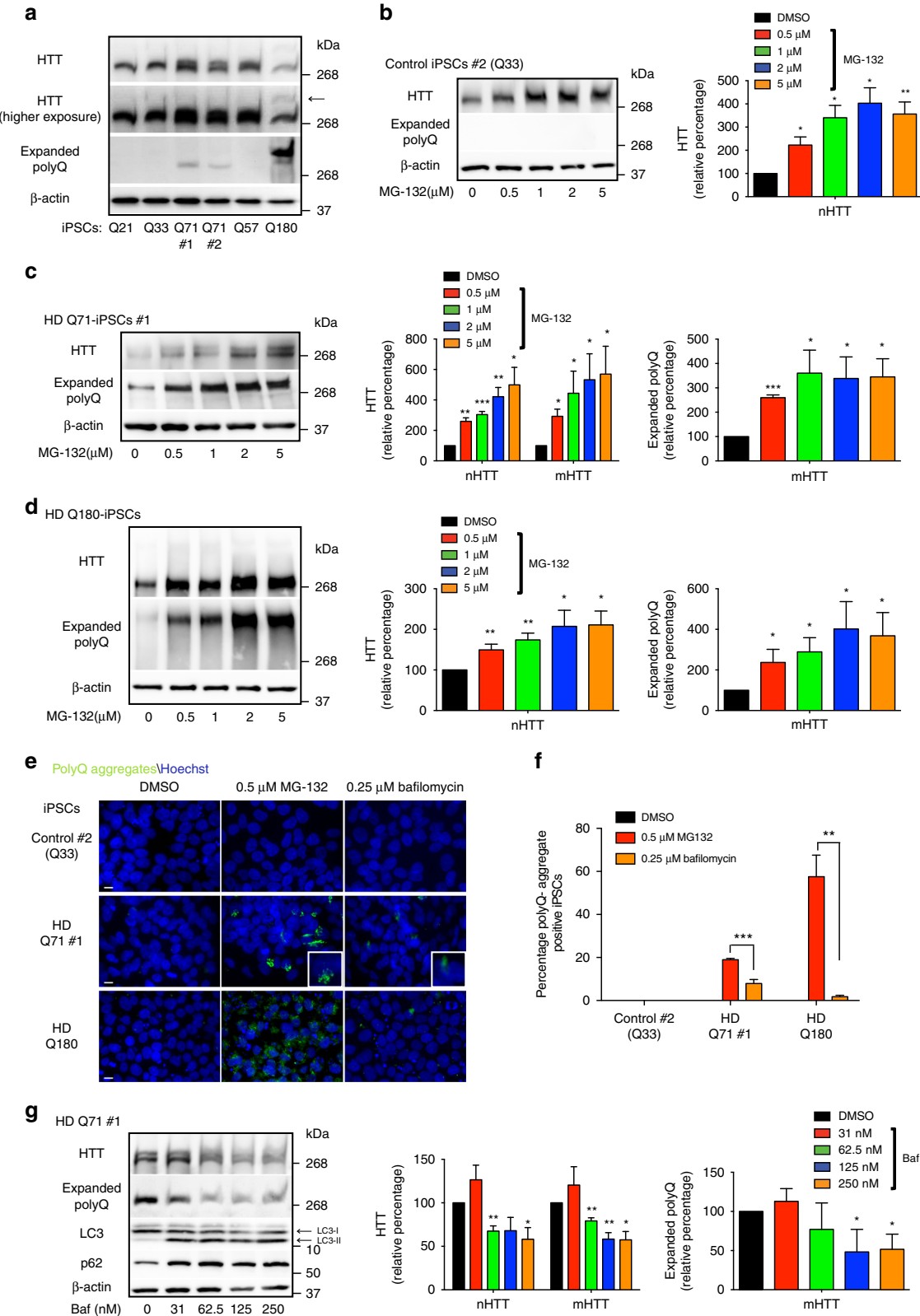

detect endogenous levels of HTT in iPSCs (Supplementary Fig. 4a–c). The HD-iPSCs used in this study express one mutant allele of HTT but also one normal copy (Supplementary Table 1)[12,25,32–34]. Since the length of the polyQ stretch diminishes the electrophoretic mobility of proteins[31], we could discriminate normal HTT and mutant HTT in both HD Q71 and Q180-iPSC lines by western blot using anti-HTT antibody (Fig. 2a and Supplementary Fig. 4a). In HD Q57-iPSCs, the differences in the electrophoretic mobilities of both alleles were marginal and they were not efficiently separated on western blot assays (Fig. 2a). We also observed that anti-HTT antibody was less immunoreactive to mutant HTT (Fig. 2a and Supplementary Fig. 4d). These differences were more pronounced in HD-iPSCs that express longer polyQ repeats (Q180), as mutant HTT was only detected after high exposure times in these cells (Fig. 2a). Thus, we used the antibody to polyQ-expanded proteins to examine the expression of mutant HTT in these cells (Fig. 2a). We confirmed that this antibody only recognizes mutant HTT on western blots and the intensity of the signal correlates with the length of the polyQ expansion (Fig. 2a and Supplementary Fig. 4d), as previously reported[31]. Although Q57 expansion was not detected with anti-polyQ-expanded proteins by western blot, this antibody strongly detected mutant HTT in Q71 and Q180 lines (Fig. 2a).

Once we characterized these antibodies in iPSCs, we assessed whether proteasome dysfunction impairs HTT levels. In control iPSCs, proteasome inhibition resulted in upregulated HTT protein levels (Fig. 2b and Supplementary Fig. 5a, b). In HD-iPSCs, proteasome dysregulation not only increased the amounts of normal HTT but also aggregation-prone HTT (Fig. 2c, d and Supplementary Fig. 5c, d). Although these results suggest a direct link between proteasome activity with HTT levels and aggregation, another possibility is that HTT dysregulation ensues from a global proteostasis collapse induced by proteasome inhibition. To assess this hypothesis, we inhibited the autophagy–lysosome system, which also modulates proteostasis of HTT through its proteolytic activity[29]. Although autophagy inhibition induced mutant HTT aggregation in HD Q71-iPSCs, these aggregates were less compact and the percentage of aggregate-containing cells were lower when compared to proteasome inhibition treatment (Fig. 2e, f and Supplementary Fig. 7a, b). In HD Q180-iPSCs, autophagy inhibition only induced aggregation in a low percentage of cells (Fig. 2e, f). Whereas proteasome inhibition increased HTT levels, autophagy downregulation resulted in decreased amounts of HTT protein (Fig. 2g and Supplementary Fig. 8a–c). This decrease in HTT levels was not associated with potential changes in cell viability or a compensatory upregulation of the proteasome, as we did not find significant changes in

these parameters upon autophagy inhibition (Supplementary Fig. 8d–e). The differences between autophagy and proteasome inhibition supported a direct role of the proteasome in HTT degradation of iPSCs. In these lines, proteasome dysfunction results in increased levels of both normal HTT and aggregation-prone HTT. Since the impairment of clearance of misfolded proteins is key to their accumulation[19], the increase in mutant HTT levels upon proteasome inhibition could contribute to diminish the ability of HD-iPSCs to suppress HTT aggregation.

**UBR5 prevents mutant HTT aggregation in HD-iPSCs**. To assess whether pluripotent stem cells have an intrinsic proteostasis to regulate HTT levels, we examined their E3 ubiquitin ligase network. For this purpose, we analyzed available quantitative proteomics data[35] and found 26 E3 ligases significantly increased in hESCs compared with their differentiated neuronal counterparts (Supplementary Table 2). Notably, UBR5 was one of the most upregulated E3 enzymes (Supplementary Table 2). UBR5-shows a striking preference for Lys48 linkages of ubiquitin[36], which is the primary signal for proteasomal degradation[19]. Under proteotoxic stress, UBR5 cooperates with Lys11-specific ligases to produce K11/K48 heterotypic chains, promoting proteasomal clearance of misfolded nascent polypeptides[36]. Recently, a genome-wide association analysis has identified that genetic variations in the chromosome region containing *UBR5* gene hasten the clinical onset of HD[37]. Remarkably, loss of UBR5 reduces the modification of overexpressed HTT protein with K11/K48-linked ubiquitin chains in a human cell line[36]. Thus, increased endogenous expression of UBR5 could provide a link between proteostasis and regulation of HTT levels in pluripotent stem cells.

Besides its downregulation during differentiation of hESCs (Supplementary Fig. 9a and Supplementary Table 2), we confirmed that UBR5 protein levels are also increased in both control and HD-iPSC compared with their neuronal counterparts (Fig. 3a–c and Supplementary Fig. 10). In all the lines tested, we observed that UBR5 downregulation is already significant when iPSCs differentiate into neural progenitor cells (NPCs) (Fig. 3a–c and Supplementary Fig. 10). The decrease in UBR5 protein levels correlated with a downregulation of messenger RNA (mRNA) amounts during differentiation (Fig. 3d–f and Supplementary Fig. 9b).

Since UBR5 promotes microRNA-mediated transcript destabilization in mouse ESCs[38] and may be involved in transcriptional repression during development[39], we examined whether UBR5 downregulation during differentiation correlates with

**Fig. 2** Loss of proteasome activity increases HTT levels in iPSCs. **a** Western blot of iPSCs with antibodies to total HTT, polyQ-expanded proteins and β-actin. Arrow indicates mutant HTT detected with total HTT antibody in HD Q180-iPSCs. The images are representative of three independent experiments. **b** Western blot of control iPSCs #2 treated with MG-132 (12 h). The graph represents the relative percentage values to DMSO-treated iPSCs of normal huntingtin (nHTT) detected with total HTT antibody and corrected for β-actin loading control (mean ± s.e.m., three independent experiments). **c** Western blot of HD Q71-iPSC line #1 treated with MG-132 (12 h). Graphs represent the relative percentage values to DMSO-treated iPSCs (corrected for β-actin) of nHTT and mutant HTT (mHTT) detected with antibodies to total HTT and polyQ-expanded proteins (mean ± s.e.m., three independent experiments). **d** Western blot of HD Q180-iPSCs treated with MG-132 (12 h). The graphs represent the relative percentage values to DMSO-treated iPSCs (corrected for β-actin) of nHTT and mHTT detected with antibodies to total HTT and polyQ-expanded proteins, respectively (mean ± s.e.m., four independent experiments). Supplementary Fig. 6 presents a higher exposure time of the same membrane for a better comparison of mHTT levels detected with HTT antibody. **e** Immunocytochemistry of control and HD-iPSCs treated with 0.5 μM MG-132 or 0.25 μM bafilomycin for 12 h. PolyQ-expanded and Hoechst 33342 staining were used as markers of mutant HTT aggregates and nuclei, respectively. Scale bar represents 10 μm. The images are representative of three independent experiments. **f** Graph represents the percentage of polyQ aggregate-positive cells/total nuclei in the indicated iPSC lines treated with MG-132 or bafilomycin for 12 h (mean ± s.e.m., four independent experiments, 300–350 total cells per treatment for each line). **g** Western blot of HD Q71-iPSCs #1 treated with bafilomycin (12 h) using antibodies to total HTT, polyQ-expanded proteins, LC3 and P62. Graphs represent the relative percentage values to DMSO-treated iPSCs (corrected for β-actin) of nHTT and mHTT (mean ± s.e.m., three independent experiments). Statistical comparisons were made by Student's *t*-test for unpaired samples. *$P < 0.05$, **$P < 0.01$, ***$P < 0.001$

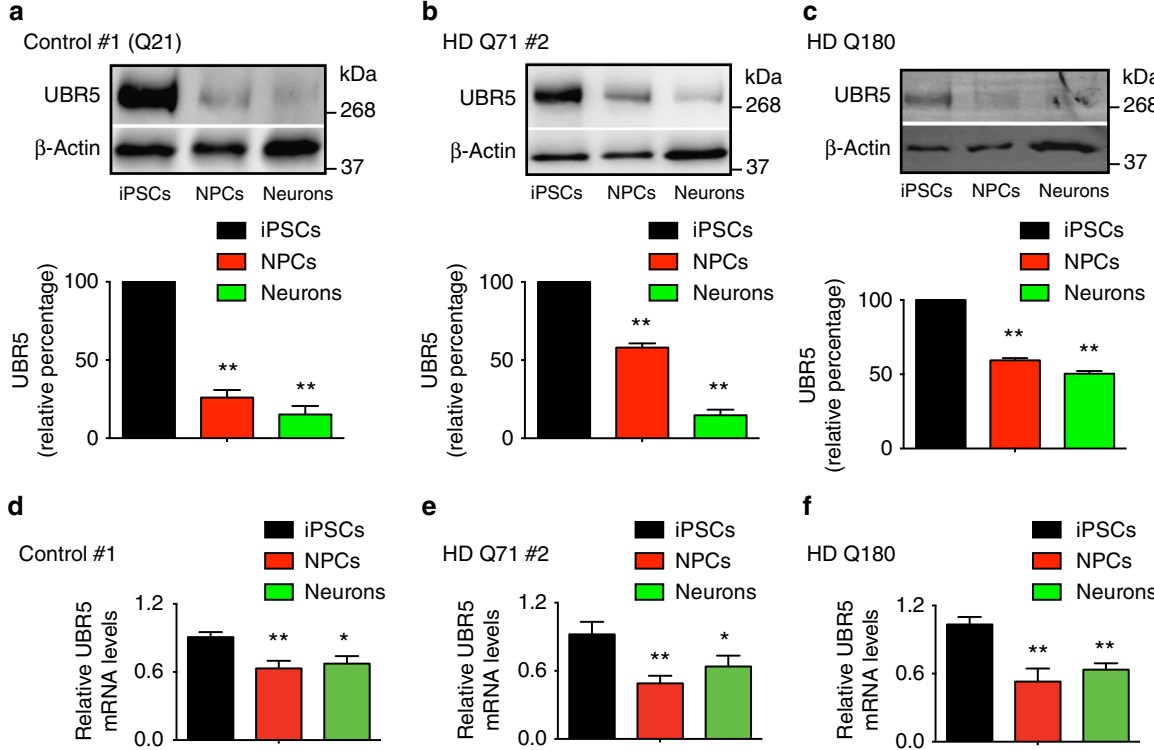

**Fig. 3** The levels of UBR5 decrease during differentiation of iPSCs. **a–c** Western blot of UBR5 levels in control #1 (**a**), HD Q71 #2 (**b**) and HD Q180 (**c**) iPSC lines compared with their neural progenitor cell (NPC) and striatal neuron counterparts. The graphs represent the UBR5 relative percentage values to the respective iPSCs corrected for β-actin loading control (mean ± s.e.m. of three independent experiments for each cell line). **d–f** Quantitative PCR (qPCR) analysis of UBR5 mRNA levels in control #1 (**d**), HD Q71 #2 (**e**) and HD Q180 (**f**) iPSC lines compared with their differentiated counterparts. Graphs (*UBR5* relative expression to iPSCs) represent the mean ± s.e.m. of three independent experiments for each line. *$P < 0.05$, **$P < 0.01$

increased HTT mRNA levels. However, the amounts of HTT mRNA were either downregulated or not significantly changed with iPSC differentiation (Supplementary Fig. 11). As a more formal test, we assessed whether knockdown of UBR5 alters the transcript levels of HTT in distinct iPSC/hESC lines and found no differences (Fig. 4a and Supplementary Fig. 12). On the contrary, knockdown of UBR5 induced an increase in the protein levels of HTT in human pluripotent stem cells (Fig. 4b–e and Supplementary Fig. 13). We assessed three independent control iPSCs as well as two hESC lines and obtained similar results (Fig. 4b, c and Supplementary Fig. 13a–c). In HD Q71 and Q180-iPSC lines, the loss of UBR5 not only impaired the levels of normal HTT but also mutant HTT at a similar extent (Fig. 4d, e and Supplementary Fig. 13d). Although we could not discriminate normal and polyQ-expanded HTT in HD Q57-iPSCs, we confirmed an upregulation of total HTT protein levels in these cells upon UBR5 knockdown (Supplementary Fig. 13e). Since loss of UBR5 did not impair HTT mRNA levels (Fig. 4a and Supplementary Fig. 12), our data supported a role of this E3 enzyme in post-translational regulation of HTT. To examine whether UBR5 modulates HTT levels in a proteasome-dependent manner, we blocked proteasomal degradation in iPSCs. Notably, UBR5 knockdown and MG-132-treated cells exhibited similar levels of HTT (Fig. 4b–e). Most importantly, UBR5 downregulation did not further increase the levels of HTT in both control and HD-iPSCs with reduced proteasome activity (Fig. 4b–e), indicating a role of UBR5 in proteasomal degradation of HTT. Given that pluripotent stem cells exhibit high levels of proteasome activity compared with their differentiated counterparts[16], we assessed whether UBR5 is required for this activity. However, loss of UBR5 did not affect global proteasome activities in iPSCs (Fig. 4f and Supplementary Fig. 14), suggesting a specific link between UBR5 and HTT

modulation. Besides UBR5, other E3 ligases are also increased in pluripotent stem cells (Supplementary Table 2 and Supplementary Fig. 15a). We knocked down four of these upregulated enzymes (i.e., UBE3A, RNF181, UBR7, TRIM71) and found no differences in HTT levels of HD-iPSCs (Supplementary Fig. 15b–g). Moreover, UBR5 interacted with both normal and polyQ-expanded HTT in HD-iPSCs, whereas we were not able to detect this interaction with a distinct upregulated E3 enzyme (Fig. 4g and Supplementary Fig. 16).

Taken together, these results suggest that intrinsic high expression of UBR5 determines HTT levels in iPSCs, a process that could contribute to the remarkable ability of these cells to maintain proteostasis of mutant HTT. In support of this hypothesis, loss of UBR5 triggered the accumulation of mutant HTT aggregates in all the HD-iPSCs tested as we confirmed by immunocytochemistry and filter trap experiments (Fig. 5a–e and Supplementary Fig. 17a). In contrast, we did not observe accumulation of polyQ aggregates in control iPSCs as well as corrected isogenic counterparts of HD Q180-iPSCs despite the upregulation of normal HTT levels (Fig. 5a, b, f, g and Supplementary Figs. 17b, 18).

Notably, proteasome inhibition did not further increase the accumulation of aggregates induced by loss of UBR5 in HD-iPSCs (Fig. 5h–l and Supplementary Fig. 19), indicating that UBR5 regulates proteostasis of mutant HTT via its proteasomal degradation. To further determine the impact of UBR5 on HTT levels and aggregation, we examined other components of the ubiquitin–proteasome system (UPS) network previously associated with HTT regulation. In particular, we focused on UBE3A, an E3 enzyme that promotes proteasomal degradation of polyQ-expanded HTT in cell lines[40]. In addition, we examined UBE2K, an E2 enzyme that was found to interact with HTT in a yeast two-

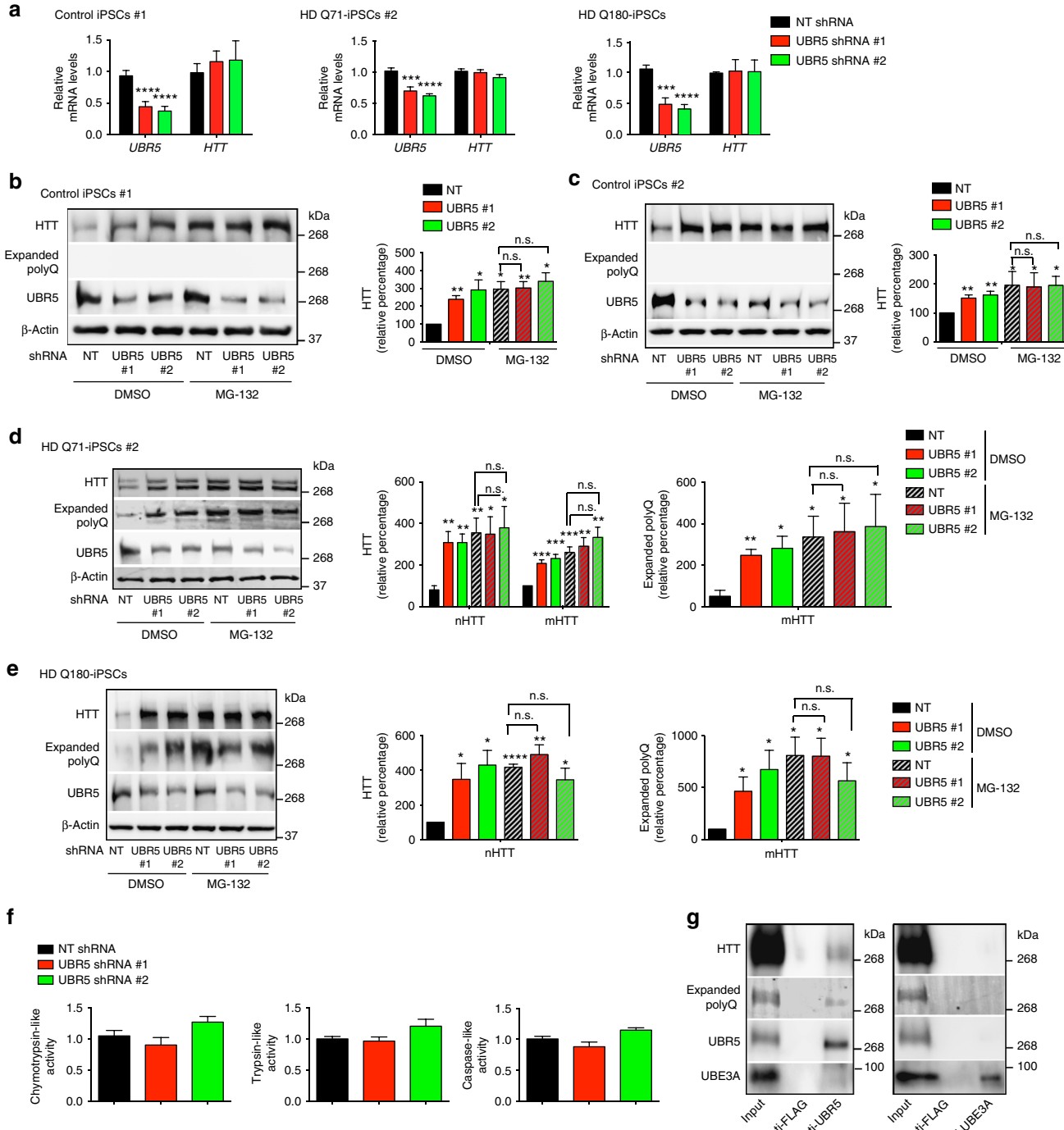

**Fig. 4** Loss of UBR5 impairs HTT protein levels in iPSCs. **a** qPCR analysis of UBR5 and HTT mRNA levels in control iPSCs #1 ($n = 4$ independent experiments), HD Q71-iPSCs #2 ($n = 3$) and HD Q180-iPSCs ($n = 3$). Graphs (relative expression to non-targeting (NT) shRNA) represent the mean ± s.e. m. **b**, **c** Western blot analysis of the indicated control iPSC lines with antibodies to HTT and polyQ-expanded proteins. Proteasome inhibitor treatment: 5 μM MG-132 for 12 h. The graphs represent the HTT relative percentage values to DMSO-treated NT shRNA iPSCs corrected for β-actin loading control (mean ± s.e.m. of three independent experiments). **d** Western blot analysis of HD Q71-iPSC line #2 upon UBR5 knockdown. The graphs represent the relative percentage values to DMSO-treated NT shRNA iPSCs (corrected for β-actin) of nHTT and mHTT detected with antibodies to total HTT and polyQ-expanded proteins (mean ± s.e.m. of four independent experiments). Proteasome inhibitor treatment: 5 μM MG-132 for 12 h. **e** Western blot analysis of HD Q180-iPSCs upon UBR5 knockdown. Proteasome inhibitor treatment: 5 μM MG-132 for 12 h. The graphs represent the relative percentage values to DMSO-treated NT shRNA iPSCs (corrected for β-actin) of nHTT and mHTT detected with antibodies to total HTT and polyQ-expanded proteins, respectively (mean ± s.e.m. of three independent experiments). **f** Proteasome activities in HD Q71-iPSCs #1 upon UBR5 knockdown (relative slope to NT shRNA iPSCs). Graphs represent the mean ± s.e.m. of three independent experiments. **g** Co-immunoprecipitation with UBR5, UBE3A and FLAG antibodies in HD Q71-iPSC line #1 followed by western blot with HTT, polyQ-expanded HTT, UBR5 and UBE3A antibodies. The images are representative of three independent experiments. All the statistical comparisons were made by Student's t-test for unpaired samples. *P < 0.05, **P < 0.01, ***P < 0.001, ****P < 0.0001

hybrid screen and reduce polyQ aggregation in cell lines[41,42]. Although both UBE3A and UBE2K are upregulated in pluripotent stem cells[35], their knockdown did not increase mutant HTT levels and aggregation in HD-iPSCs (Fig. 6a, b and Supplementary Fig. 20). Thus, these results further support a specific role of UBR5 as a key determinant of HTT levels in iPSCs. Whereas the UPS regulates HTT degradation facilitating its proteostasis, the chaperone network is essential to modulate different steps of the aggregation process[29]. Remarkably, pluripotent stem cells exhibit an intrinsic chaperone network that could contribute to

preventing mutant HTT aggregation[12]. For instance, these cells have increased levels of HTT-interacting protein K (HYPK)[35], a chaperone that reduces polyQ-expanded HTT aggregates in mouse neuroblastoma cell lines[43]. However, loss of HYPK did not trigger aggregation of mutant HTT in HD-iPSCs (Fig. 6b). Pluripotent stem cells also exhibit increased assembly of the TRiC/CCT complex[12], a chaperonin that suppresses mutant HTT aggregation[44,45]. In hESCs/iPSCs, increased assembly of the TRiC/CCT complex is induced by high levels of CCT8 subunit[12]. As UBR5 knockdown, loss of CCT8 triggers aggregation of

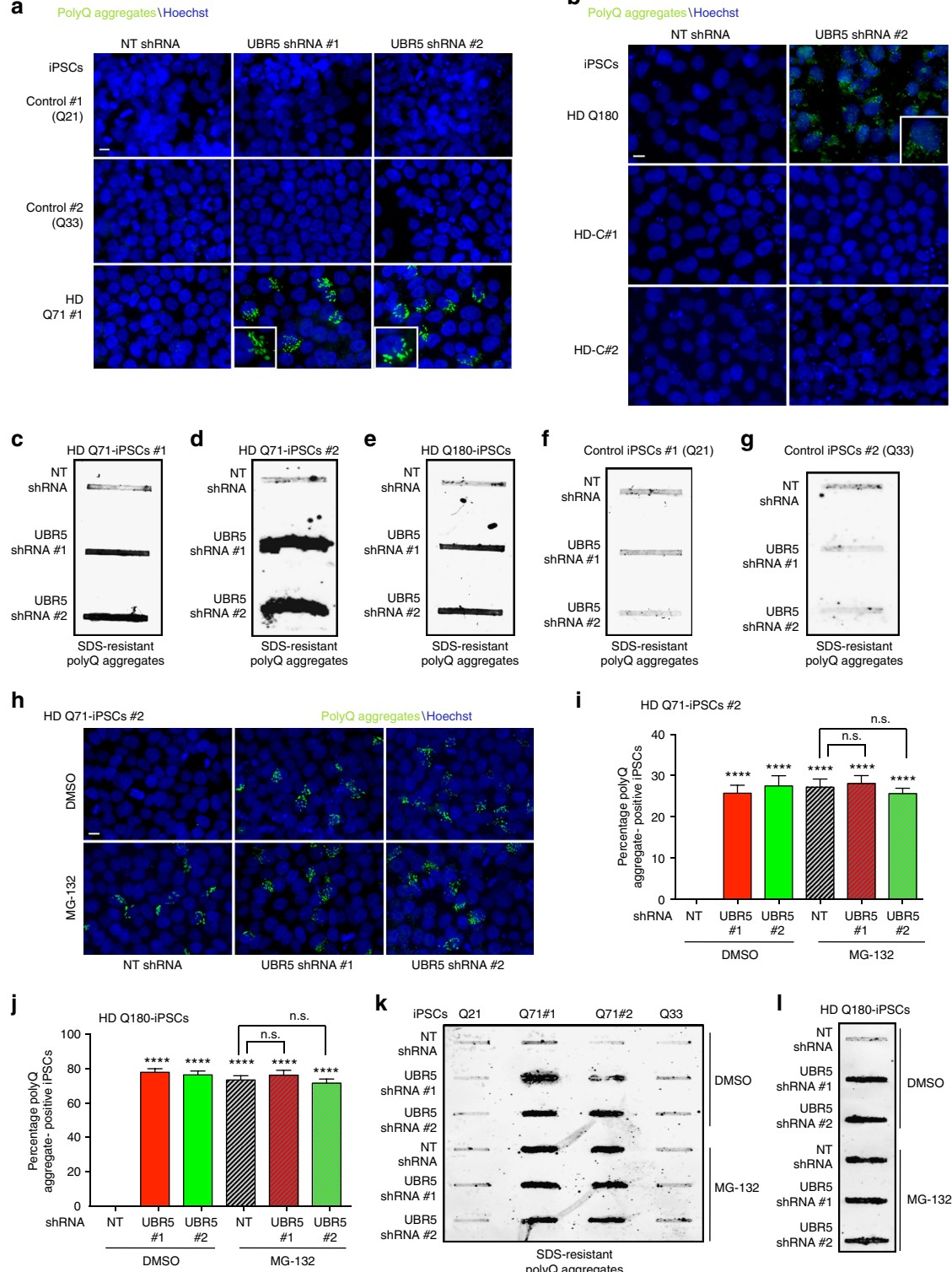

mutant HTT in HD-iPSCs (Fig. 6b)[12]. In contrast, CCT8 downregulation did not affect HTT protein levels in HD-iPSCs (Fig. 6a), supporting a direct role of the TRiC/CCT complex in mutant HTT aggregation rather than degradation. Altogether, our results indicate a key role of UBR5 in monitoring HTT levels, a process that could facilitate mutant HTT regulation by other proteostasis nodes such as enhanced TRiC/CCT complex.

**Loss of UBR5 triggers aggresome formation in hESCs/iPSCs**. Besides its role in HTT regulation, we asked whether UBR5 modulates other disease-associated proteins. To assess this hypothesis, we first examined ataxin-3 (ATXN3), a distinct polyQ-containing protein. An abnormal expansion of CAG triplets (>52) in the *ATXN3* gene causes Machado–Joseph disease (MJD), a neurodegenerative disorder characterized by neuronal loss in the cerebellum and progressive ataxia[46,47]. Here we used iPSCs derived from two individuals with MJD (MJD-iPSCs) (Supplementary Table 1), both expressing one normal copy of *ATXN3* and one mutant allele with 74 CAG repeats (Fig. 6c). As previously reported[48], these MJD-iPSCs did not accumulate polyQ-expanded aggregates (Fig. 6d). To examine whether the UPS regulates proteostasis of ATXN3 in iPSCs, we downregulated proteasome activity by using MG-132 proteasome inhibitor. In contrast with HD-iPSC lines, the treatment with proteasome inhibitor for 12 h induced acute cell death and detachment of MJD-iPSCs. To reduce these effects, we performed our analysis at an earlier time point of the treatment (6 h) (Supplementary Fig. 21). Notably, proteasome inhibition did not change the levels of normal and mutant ATXN3, whereas the amounts of HTT were upregulated in these cells (Fig. 6c). Thus, our results suggest that the UPS does not modulate the levels of ATXN3 in iPSCs. However, proteasome inhibition triggered polyQ-expanded ATXN3 aggregation (Fig. 6d), a process that could be linked with the global proteostasis collapse induced by this treatment. Although loss of UBR5 increased HTT levels in MJD-iPSCs, we did not find changes in either normal or mutant ATXN3 (Fig. 6e, f). Moreover, we found that UBR5 binds HTT but it does not interact with wild-type or mutant ATXN3 in MJD-iPSCs (Supplementary Fig. 22a). In contrast to global proteasome inhibition, UBR5 downregulation did not induce aggregation of polyQ-expanded ATXN3 (Fig. 6g). This phenotype differed from HD-iPSCs lines, where upregulation of mHTT levels upon UBR5 knockdown was sufficient to trigger polyQ-expanded HTT aggregation (Fig. 5). Thus, these data indicate that not all polyQ-containing proteins are modulated by UBR5.

To further assess the role of UBR5 in the proteostasis of aggregation-prone proteins, we used iPSCs carrying mutations in the RNA-binding protein FUS[49]. These mutations are linked with amyotrophic lateral sclerosis (ALS), a fatal neurodegenerative disorder characterized by loss of motor neurons and concomitant progressive muscle atrophy[49,50]. Although wild-type FUS shuttles between the nucleus and cytoplasm, it is mostly localized in the nucleus in normal conditions[51]. However, many of the ALS-related FUS mutations disrupt the nuclear import of the protein, resulting in aberrant localization and aggregation of FUS in the cytoplasm[52,53]. As previously reported[49], wild-type FUS was essentially located in the nucleus of iPSCs (Fig. 7a). Likewise, FUS was predominantly detected in the nucleus of iPSCs (FUS$^{R521C/wt}$) derived from a patient affected by ALS in mid-late age[49,50] (Fig. 7a). In iPSCs expressing a FUS variant linked with severe and juvenile ALS (FUS$^{P525L/P525L}$), the protein was mostly located in the cytoplasm[49] (Fig. 7a). Under oxidative stress, the cells form stress granules (SGs) where mutant variants of FUS are recruited[54]. Despite the induction of cytoplasmic SGs, wild-type FUS remains in the nucleus in iPSCs under oxidative stress[49]. On the contrary, mutant variants of FUS colocalize with SGs[49]. Similar to oxidative stress, proteasome inhibition induced the accumulation of SGs in iPSCs (Fig. 7a). In control iPSCs, FUS remained in the nucleus upon proteasome dysfunction (Fig. 7a). Although FUS signal was mostly nuclear in FUS$^{R521C}$-iPSCs upon proteasome inhibition, we also found cytoplasmic FUS speckles co-localizing with SGs as reported for oxidative stress[49] (Fig. 7a). On the other hand, FUS$^{P525L}$ variant showed a strong co-localization with SGs under proteasome inhibition (Fig. 7a). Remarkably, loss of UBR5 did not stimulate SG formation or FUS delocalization in control or ALS-iPSC lines (Fig. 7a). In addition, UBR5 knockdown did not impair wild-type or mutant FUS levels, whereas HTT was upregulated in these cells (Fig. 7b). Accordingly, UBR5 interacted with HTT in both control and ALS-iPSC lines, but we could not detect interaction with wild-type or mutant FUS in these cells (Supplementary Fig. 22b, c). Thus, these results indicate a specific role of UBR5 on HTT regulation. Although UBR5 was dispensable for ATXN3 and FUS proteostasis, we cannot discard a role of UBR5 in the control of other aggregation-prone proteins associated with disease.

Besides its role in HTT modulation, we asked whether UBR5 also determines the global proteostatic ability of pluripotent cells. Under normal conditions, misfolded proteins are refolded by chaperones or terminated via proteolytic systems[10]. Metabolic and environmental conditions (e.g., heat stress) challenge the structure of proteins, increasing the load of misfolded and damaged proteins. When proteolytic systems are overwhelmed, misfolded proteins accumulate into aggresomes. In these lines, we observed that heat stress induces the accumulation of aggresomes in pluripotent stem cells despite their increased proteolytic ability

**Fig. 5** UBR5 suppresses mutant HTT aggregation in HD-iPSCs. **a** Immunocytochemistry of control iPSC line #1 (Q21), control iPSC line #2 (Q33) and HD Q71-iPSC line #1 upon UBR5 knockdown. PolyQ-expanded and Hoechst 33342 staining were used as markers of aggregates and nuclei, respectively. Scale bar represents 10 μm. The images are representative of four independent experiments. **b** Immunocytochemistry of Q180-iPSCs and two isogenic counterparts (i.e., HD-C#1 and HD-C#2), in which the 180 CAG expansion was corrected to a nonpathological repeat length. Scale bar represents 10 μm. The images are representative of three independent experiments. **c–g** Filter trap analysis of the indicated control and HD-iPSC lines upon UBR5 knockdown with anti-polyQ-expansion diseases marker antibody. The images are representative of at least three independent experiments for each cell line. **h** Immunocytochemistry of HD Q71-iPSC line #2 upon knockdown of UBR5. PolyQ-expanded and Hoechst 33342 staining were used as markers of aggregates and nuclei, respectively. Scale bar represents 10 μm. Proteasome inhibitor treatment: 5 μM MG-132 for 12 h. The images are representative of four independent experiments. **i** Graph represents the percentage of polyQ aggregate-positive cells/total nuclei in HD Q71-iPSC line #2 (mean ± s.e.m., 3 independent experiments, 500–600 total cells per condition). Proteasome inhibitor treatment: 5 μM MG-132 for 12 h. **j** Graph represents the percentage of polyQ aggregate-positive cells/total nuclei in HD Q180-iPSCs (mean ± s.e.m., 3 independent experiments, 300–400 total cells per condition). Proteasome inhibitor treatment: 5 μM MG-132 for 12 h. **k** Filter trap analysis of the indicated control and HD-iPSC lines with anti-polyQ-expansion diseases marker antibody. Proteasome inhibitor treatment: 5 μM MG-132 for 12 h. The images are representative of three independent experiments. **l** Filter trap analysis of HD Q180-iPSCs with anti-polyQ-expansion diseases marker antibody. Proteasome inhibitor treatment: 5 μM MG-132 for 12 h. The images are representative of three independent experiments. All the statistical comparisons were made by Student's *t*-test for unpaired samples. ****$P < 0.0001$

(Fig. 7c). Notably, loss of UBR5 was sufficient to induce the accumulation of aggresomes in these cells (Fig. 7c). Thus, our data suggest that UBR5 not only regulates the levels of specific proteins such as HTT, but is also involved in the degradation of misfolded proteins ensued during normal metabolism.

**Mutant HTT aggregates do not affect neural differentiation.** Loss of UBR5 did not impair the levels of pluripotency and germ layer markers in control iPSCs (Supplementary Fig. 23a–c). With the pronounced accumulation of mutant HTT aggregates induced by UBR5 knockdown in HD-iPSCs (Fig. 5), we asked whether

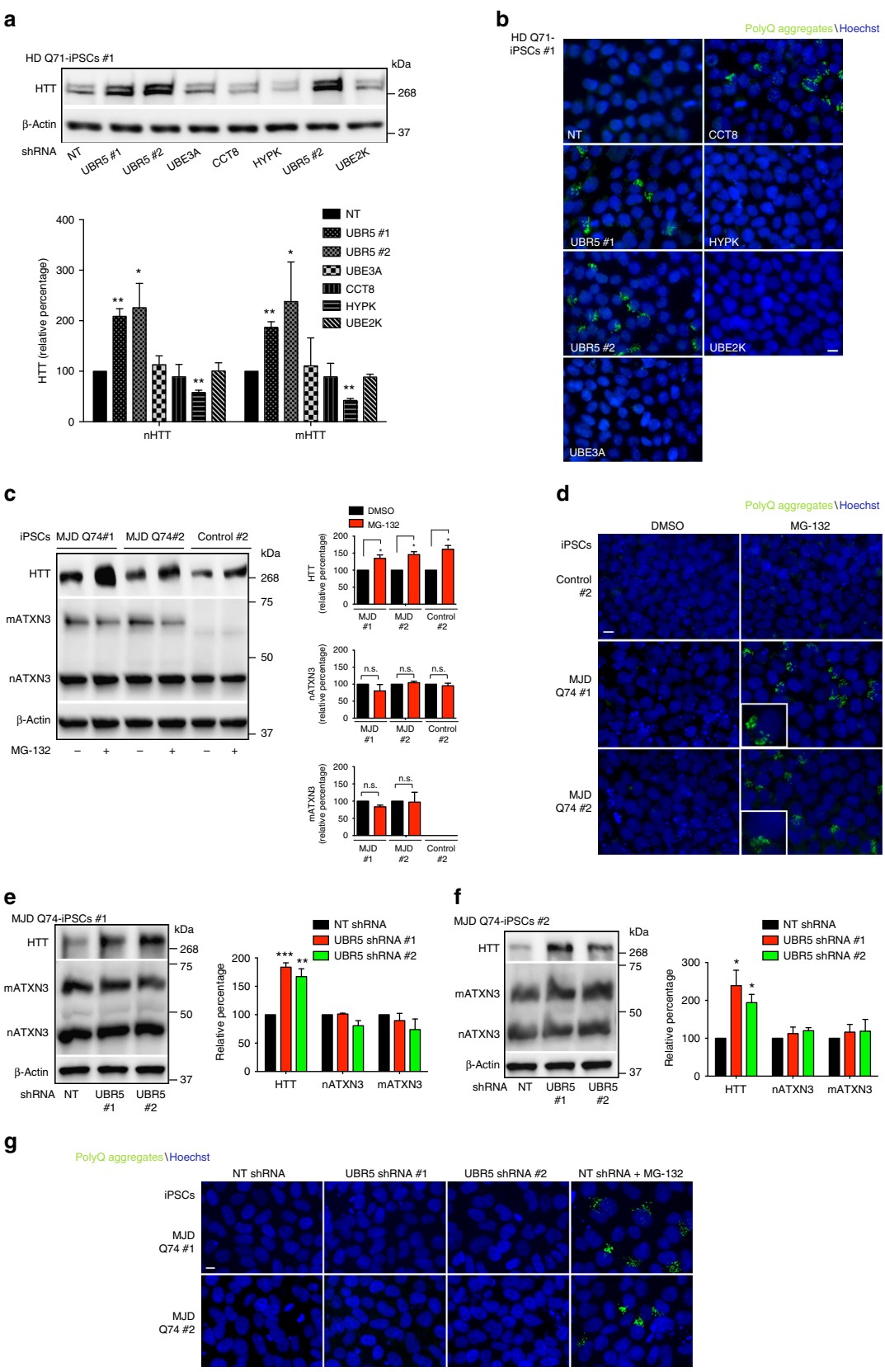

these inclusions alter iPSC identity. However, the accumulation of aggregates did not result in decreased expression of pluripotency markers in HD-iPSCs (Fig. 8a and Supplementary Fig. 24). Likewise, we did not observe increased levels of markers of the distinct germ layers (Fig. 8a), indicating that polyQ-expanded HTT aggregates do not induce differentiation. Accordingly, we did not find a decrease in the number of cells expressing the pluripotency marker OCT4 (Fig. 8b and Supplementary Fig. 25). Most importantly, UBR5 knockdown HD-iPSCs could differentiate into neural cells (Fig. 8c and Supplementary Fig. 26). Under neural induction treatment, naive HD-iPSCs differentiated into NPCs with no detectable amounts of polyQ-expanded-HTT aggregates (Fig. 8d and Supplementary Fig. 27). However, neural cells derived from HD-iPSCs with impaired expression of UBR5 exhibited increased levels of mutant HTT and accumulation of aggregates (Fig. 8d–h and Supplementary Figs. 27–29). Overall, our results indicate that intrinsic high levels of UBR5 are essential to suppress mutant HTT aggregation in iPSCs, contributing to their ability to generate neural progenitor cells with no detectable polyQ-expanded aggregates. One step further was to determine whether these cells are able to generate terminally differentiated neurons. Since GABAergic medium spiny neurons (MSNs) undergo the greatest neurodegeneration in HD[55], we differentiated iPSCs into striatal neurons[56]. Among those cells expressing the neuronal marker microtubule-associated protein-2 (MAP2), ~30–60% also expressed GABA depending on the cell line (Supplementary Fig. 30a, b). As previously reported[25], these differences were not associated with the expression of mutant HTT (Supplementary Fig. 30a, b). Knockdown of UBR5 at the iPSC stage did not reduce their ability to differentiate into striatal neurons (Supplementary Fig. 30a–g). In contrast to NPCs, terminally differentiated neurons derived from HD-iPSCs with downregulated UBR5 levels did not accumulate polyQ-expanded aggregates (Supplementary Fig. 31). Thus, the lack of defects in neurogenesis and aggregates in these cells indicate that other mechanisms activate during neuronal differentiation to facilitate proteostasis of mutant HTT.

**UBR5 loss hastens neurotoxicity in polyQ nematode models**. Although HD-iPSCs can terminally differentiate into MSNs, these cells do not exhibit mutant HTT aggregates even after the addition of proteasome and autophagy inhibitors or the induction of oxidative stress[12,20,25,27]. Thus, these findings support a rejuvenation process during cell reprogramming that prevents aberrant aggregation in differentiated neurons. In addition, the lack of polyQ-expanded aggregates in these cells could reflect the long period of time before aggregates accumulate in HD[25]. In these

lines, HD-MSNs derived from iPSCs do not accumulate detectable polyQ-expanded inclusions at 12 weeks after transplantation into HD rat models. However, they accumulate aggregates after 33 weeks of transplantation[27]. Proteasome inhibition did not induce upregulation of HTT protein levels or aggregation in HD-MSNs differentiated from iPSCs (Supplementary Fig. 32a–d). Likewise, knockdown of UBR5 in differentiated HD-MSNs was not sufficient to induce mutant HTT aggregation and upregulated HTT levels (Supplementary Fig. 33, 34). In addition, UBR5 knockdown did not reduce cell viability of HD-MSNs (Supplementary Fig. 35).

Given the challenges presented by MSNs derived from HD-iPSCs to study the role of UBR5 in polyQ-expanded aggregation and toxicity, we used a distinct model. Remarkably, a RNA interference (RNAi) screen against E3 ubiquitin ligases found that knockdown of the worm *UBR5* ortholog (*ubr-5*) accelerates paralysis in a *C. elegans* model expressing 35 polyQ repeats fused to yellow fluorescent protein (YFP) in body wall muscle cells[57]. To assess the requirement of *ubr-5* for resistance to polyQ neurotoxicity, we examined a *C. elegans* model that expresses polyQ-expanded YFP in the nervous system[58]. In these worms, polyQ aggregation and neurotoxicity correlates with the age and length of the polyQ repeat, with a pathogenic threshold of 40 glutamine repeats[58]. Notably, loss of *ubr-5* resulted in increased levels of polyQ67-expanded protein with concomitant aggregation (Fig. 9a, b). Knockdown of *ubr-5* had a strong effect in the aggregation propensity of head neurons, resulting in more aggregates in the circumpharyngeal nerve ring and chemosensory processes (Fig. 9c). We also observed more propensity aggregation in the neurons of the animal mid-body (Fig. 9c and Supplementary Fig. 36). To assess whether the polyQ67-YFP foci were immobile protein aggregates, we performed a quantitative fluorescence recovery after photobleaching analysis of head neurons. In both empty vector and *ubr-5* RNAi-treated worms, most of the polyQ67-YFP foci signal could not be recovered, indicating an immobile state (Supplementary Fig. 37a). However, *ubr-5* RNAi induced a faster incorporation of new polyQ67-YFP peptides into the aggregates (Supplementary Fig. 37b).

Although *ubr-5* RNAi was sufficient to induce a pronounced increase of polyQ-expanded aggregates (Fig. 9b), it is important to note that neurons are less sensitive to RNAi when compared with other tissues[59]. For this reason, we introduced a *rrf-3* mutation in the polyQ-YFP neuronal models, which confers hypersensitivity to RNAi in all the tissues, including neurons[60]. Accordingly, we found that *ubr-5* RNAi treatment induced a strong increase in polyQ67-YFP aggregation of *rrf-3* mutants (Fig. 9d). On the contrary, *ubr-5* knockdown did not induce

**Fig. 6** UBR5 maintains proteostasis of HTT but not ATXN3 in iPSCs. **a** Western blot analysis of HD Q71-iPSC line #1 with antibody to HTT. The graph represents the nHTT and mHTT relative percentage values (corrected for β-actin loading control) to NT shRNA iPSCs (mean ± s.e.m. of three independent experiments). **b** Immunocytochemistry of HD Q71-iPSC line #1 upon knockdown of the indicated proteostasis components. PolyQ-expanded and Hoechst 33342 staining were used as markers of aggregates and nuclei, respectively. Scale bar represents 10 μm. The images are representative of three independent experiments. **c** Western blot analysis with antibodies to HTT and ATXN3 of iPSCs derived from two individuals with MJD (MJD-iPSC lines #1 and #2) and control iPSCs. Anti-ATXN3 antibody detects both normal (nATXN3) and mutant ATXN3 (mATXN3). Proteasome inhibitor treatment: 5 μM MG-132 for 6 h. The graphs represent the HTT, nATXN3, mATXN3 relative percentage values to the respective DMSO-treated iPSCs corrected for β-actin loading control (mean ± s.e.m. of three independent experiments). **d** Immunocytochemistry of control iPSCs #2, MJD-iPSC lines #1 and #2 upon proteasome inhibition (5 μM MG-132 for 6 h). PolyQ-expanded and Hoechst 33342 staining were used as markers of aggregates and nuclei, respectively. Scale bar represents 10 μm. The images are representative of three independent experiments. **e, f** Western blot analysis with antibodies to HTT and ATXN3 of MJD-iPSC lines #1 and #2 upon UBR5 knockdown. The graphs represent the relative percentage values of HTT, nATXN3 and mATXN3 to the respective NT shRNA iPSCs corrected for β-actin loading control (mean ± s.e.m. of three independent experiments for each line). **g** Immunocytochemistry of MJD-iPSC lines #1 and #2 upon UBR5 knockdown. Proteasome inhibitor: 5 μM MG-132 for 6 h. PolyQ-expanded and Hoechst 33342 staining were used as markers of aggregates and nuclei, respectively. Scale bar represents 10 μm. The images are representative of three independent experiments. All the statistical comparisons were made by Student's *t*-test for unpaired samples. *$P < 0.05$, **$P < 0.01$, ***$P < 0.001$

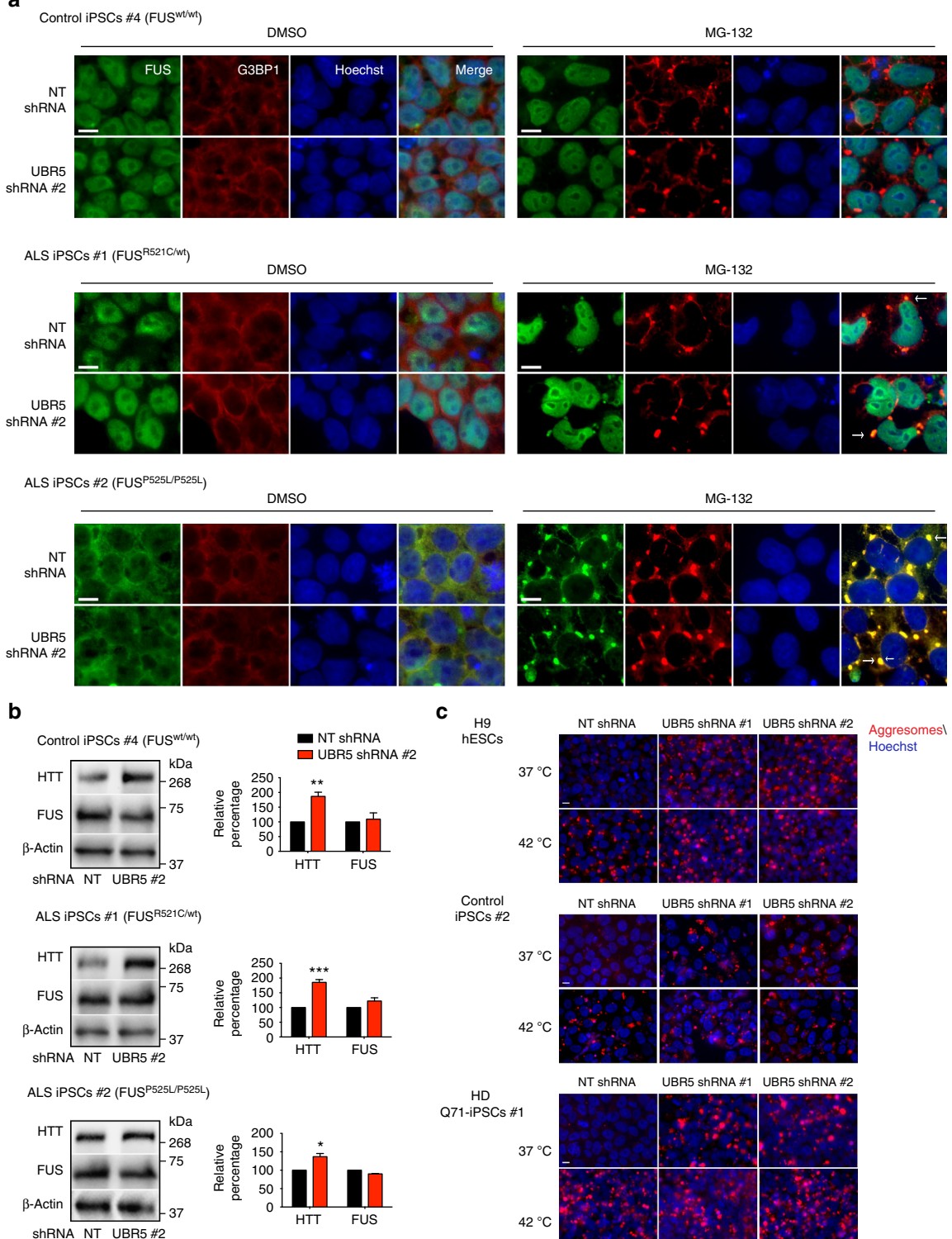

**Fig. 7** Loss of UBR5 triggers aggresome formation in pluripotent stem cells. **a** Immunocytochemistry of control iPSCs #4 (FUS^wt/wt), ALS-iPSCs #1 (FUS^R521C/wt) and ALS-iPSCs #2 (FUS^P525L/P525L) with anti-FUS antibody. G3BP1 and Hoechst 33342 staining were used as markers of stress granules (SGs) and nuclei, respectively. Proteasome inhibition: 5 μM MG-132 for 6 h. Scale bar represents 10 μm. Arrows indicate examples of co-localization of FUS with SGs. The images are representative of three independent experiments. **b** Western blot analysis with antibodies to HTT and FUS of ALS-iPSC lines upon UBR5 knockdown. The graphs represent the relative percentage values of HTT and FUS to the respective NT shRNA iPSCs corrected for β-actin loading control (mean ± s.e.m. of three independent experiments for each line). All the statistical comparisons were made by Student's t-test for unpaired samples. *$P < 0.05$, **$P < 0.01$, ***$P < 0.001$. **c** Staining of aggresomes in H9 hESCs, control iPSCs #2 and HD Q71-iPSCs #1. Hoechst 33342 staining was used as a marker of nuclei. Heat stress: 42 °C for 4 h. Scale bar represents 10 μm. The images are representative of two independent experiments for each line

aggregation of polyQ19-peptides, even in the RNAi-hypersensitive mutant strain (Fig. 9d).

Since the neurotoxic effects of polyQ-expanded aggregation correlate with impairment of coordinated movement[58], we performed motility assays to quantify the role of *ubr-5* in the resistance to polyQ-toxicity. Notably, loss of *ubr-5* hastened the

detrimental effects on motility induced by polyQ67 repeats (Fig. 9e), correlating with increased polyQ aggregation. On the contrary, knockdown of *ubr-5* did not affect the motility of polyQ19-expressing worms (Fig. 9e). Altogether, these results provide a direct link between UBR5 function and polyQ-expanded aggregation with age.

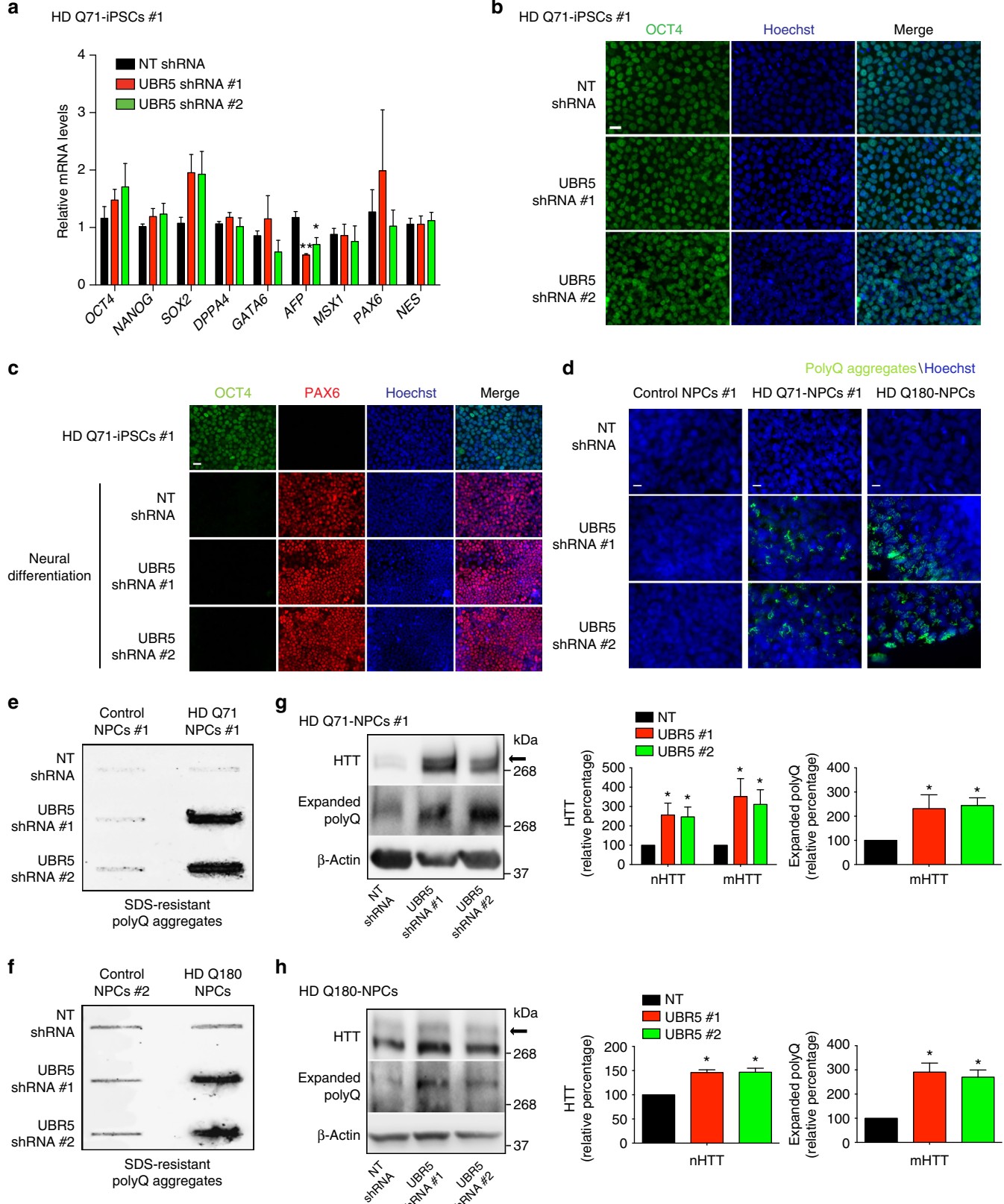

**Ectopic expression of UBR5 suppresses mutant HTT aggregation**. Since loss of UBR5 hastened polyQ-expanded aggregation, we asked whether ectopic expression of this HECT E3 enzyme is sufficient to ameliorate the accumulation of polyQ aggregates. To assess this hypothesis, we generated human cell models that express either control (Q23) or polyQ-expanded (Q100) HTT protein. In these cells, overexpression of mutant HTT resulted in the accumulation of polyQ-expanded aggregates, whereas control HTT did not form aggregates (Fig. 10a). Notably, ectopic expression of UBR5 ameliorated polyQ-expanded aggregation in Q100-HTT cells (Fig. 10a), a process blocked by proteasome inhibition (Fig. 10b). In contrast, UBR5 overexpression did not reduce the insolubility of aggregation-prone β-amyloid protein (Supplementary Fig. 38).

In support of a direct role of UBR5 in modulation of HTT, we found that UBR5 overexpression decreases the protein levels of mutant HTT (Fig. 10c). To further determine the role of UBR5, we overexpressed a UBR5 mutant with two point mutations that causes a change in an amino acid (C2769A) located in the HECT domain, resulting in ubiquitin ligase-dead UBR5[61]. Remarkably, similar overexpression levels of this catalytic inactive UBR5 mutant did not diminish polyQ-expanded HTT protein levels and aggregation (Fig. 10a–c). Given that our results in iPSCs indicate that UBR5 regulates HTT levels via the proteasome, we tested whether UBR5 overexpression promotes proteasomal degradation of Q100-HTT. Indeed, proteasome inhibition blocked the reduction of Q100-HTT levels induced by enhanced wild-type UBR5 expression (Fig. 10d). To further assess the link between UBR5 and HTT regulation, we performed immunoprecipitation experiments and examined polyubiquitination of HTT. Prior to immunoprecipitation, we treated the cells with proteasome inhibitor to block the degradation of HTT induced by UBR5. Under these conditions, we immunoprecipitated similar amounts of HTT in cells overexpressing wild-type UBR5 when compared with cells expressing empty vector or catalytic inactive UBR5 (Fig. 10e). Notably, we found that ectopic expression of UBR5 induces a dramatic increase in polyubiquitination of mutant HTT (Fig. 10e). In contrast, overexpression of the catalytic inactive UBR5 mutant did not promote polyubiquitination of Q100-HTT protein (Fig. 10e). In support of our findings, a recent study reported that UBR5 downregulation decreases the modification of overexpressed HTT protein with K11/K48-linked polyubiquitin chains in HeLa human cells[36]. Moreover, this study revealed K11/K48-linked polyubiquitin modifications at Lys337 of HTT by proteomics experiments[36]. We performed proteomics analysis of immunoprecipitated HTT in Q100-HTT-overexpressing cells to determine potential lysine sites modified by UBR5

(Supplementary Fig. 39a, b). Although we did not detect ubiquitination at Lys337 of HTT in our assays, we identified two other ubiquitinated lysine sites: Lys631 and Lys2097. However, only Lys631 shows a small but significant higher ubiquitination in cells overexpressing wild-type UBR5 when compared to catalytic inactive UBR5 (Supplementary Fig. 39b), suggesting that this site could be ubiquitinated by UBR5. Taken together, our data indicate that the ubiquitin ligase activity of UBR5 modulates proteasomal degradation of mutant HTT, a process that ameliorates polyQ-expanded aggregation.

## Discussion

While the transcriptional, epigenetic and signaling networks of pluripotency have been a primary focus of research efforts, emerging evidence indicates that pluripotent stem cells also exhibit intrinsic proteostasis mechanisms[8,12,14,17]. Thus, a comprehensive understanding of this proteostasis network could be necessary for pluripotent stem cells to hold a great promise for regenerative medicine. As an invaluable resource to generate terminally differentiated cells, pluripotent stem cells can facilitate the study of human diseases and drug screening. This is particularly fascinating in the context of proteostasis-related disorders. At least 30 different human diseases are directly associated with aberrant protein folding protein and aggregation[62]. Whereas a collapse in proteostasis of somatic cells could underlie these diseases, pluripotent stem cells exhibit a striking ability to correct and suppress proteostatic deficiencies. Thus, investigating pluripotent stem cells from patients could contribute to identifying super-vigilant proteostasis mechanisms that could be mimicked in somatic tissues to ameliorate disease.

Despite expressing significant amounts of mutant HTT[25], several studies have demonstrated that HD-iPSCs do not accumulate polyQ-expanded HTT aggregates even after multiple passages[12,25,27]. In addition, HD-iPSCs can terminally differentiate into neurons lacking the aggregation phenotype characteristic of HD[12,25,27], indicating a rejuvenation process during reprogramming that prevents polyQ-expanded aggregation in neurons prior to proteostasis collapse with age. Here we show that increased proteasome activity of iPSCs regulates the levels of both normal and mutant HTT, contributing to suppressing polyQ-expanded HTT aggregation in HD-iPSCs. Conversely, a dysfunction in proteasome activity results in impaired HTT levels, leading to aggregation of mutant HTT in HD-iPSCs. Importantly, we have uncovered that intrinsic high expression of UBR5 is a key component of the UPS to regulate HTT levels in iPSCs. This E3 enzyme interacts with HTT and promotes

**Fig. 8** UBR5 determines HD-iPSC differentiation into NPCs with no aggregates. **a** qPCR of pluripotency (*OCT4*, *NANOG*, *SOX2*, *DPPA4*), endodermal (*GATA6*, *AFP*), mesodermal (*MSX1*) and ectodermal markers (*PAX6*, *NES*) in HD Q71-iPSC line #1. The graph represents the relative expression to NT shRNA HD-iPSCs (mean ± s.e.m. (*n* = four independent experiments)). **b** Immunocytochemistry of HD Q71-iPSC line #1. OCT4 and Hoechst 33342 staining were used as markers of pluripotency and nuclei, respectively. Scale bar represents 20 μm. The images are representative of four independent experiments. **c** After neural induction of HD Q71-iPSC line #1 with downregulated levels of UBR5, we observed similar numbers of PAX6-positive cells compared to NT shRNA control. Scale bar represents 20 μm. **d** Upon neural induction of HD-iPSCs with downregulated levels of UBR5, NPCs accumulate mutant HTT aggregates. PolyQ-expanded and Hoechst 33342 staining were used as markers of aggregates and nuclei, respectively. Scale bar represents 10 μm. The images are representative of three independent experiments. **e** Filter trap analysis of polyQ-expanded aggregates in NPCs derived from HD Q71-iPSCs #1 with downregulated levels of UBR5. The images are representative of four independent experiments. **f** Filter trap of control NPCs #2 and HD Q180-NPCs derived from iPSCs with downregulated levels of UBR5. The images are representative of three independent experiments. **g** HD Q71-iPSCs #1 with downregulated UBR5 levels were differentiated into NPCs and analyzed by western blot. Graphs represent the relative percentage values to NT shRNA NPCs (corrected for β-actin) of nHTT and mHTT detected with antibodies to total HTT and polyQ-expanded proteins (mean ± s.e.m. of three independent experiments). **h** Western blot analysis of NPCs derived from HD Q180-iPSCs with downregulated levels of UBR5. Graphs represent the relative percentage values to NT shRNA NPCs (corrected for β-actin) of nHTT and mHTT detected with antibodies to total HTT and polyQ-expanded proteins (mean ± s.e.m. of three independent experiments), respectively. In **g**, **h** arrow indicates polyQ-expanded HTT detected with total HTT antibody. Statistical comparisons were made by Student's *t*-test for unpaired samples. *$P < 0.05$, **$P < 0.01$

proteasomal degradation of both normal and polyQ-expanded HTT. Since the clearance of misfolded and aggregation-prone proteins is key to their aggregation, the impairment of HTT levels could be particularly relevant in iPSCs expressing mutant HTT. Accordingly, UBR5 downregulation and concomitant dysregulation of HTT levels results in polyQ-expanded HTT aggregation in HD-iPSCs. Thus, our results suggest that UBR5 monitors HTT levels, facilitating the regulation of mutant HTT by other proteostasis nodes. For instance, pluripotent stem cells also exhibit increased activity of the TRiC/CCT chaperone complex to tightly suppress the aggregation of mutant HTT[12]. We found that the HECT domain containing the E3 ligase activity of UBR5 is required for HTT degradation, providing a further link between increased proteasome activity and stringent control of mutant HTT. It is important to note that pluripotent stem cells also express high levels of other E3 ligases. Although our results discard a direct role of several of these E3s (i.e., UBR7, UBE3A, RNF181) in modulation of HTT levels in

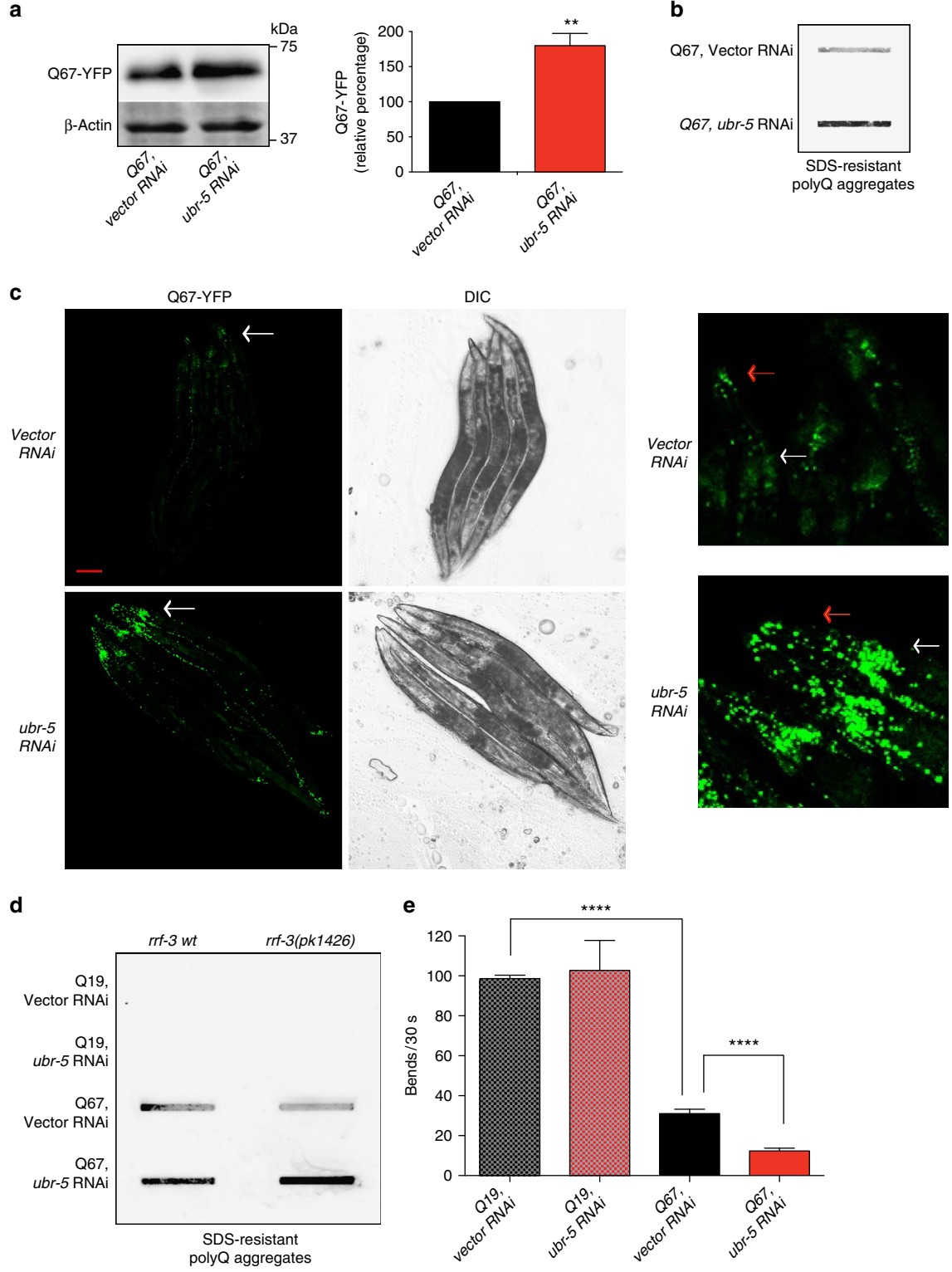

iPSCs, it will be fascinating to examine the impact of the other upregulated E3s.

Whereas our data highlight the importance of UBR5 in proteostasis of iPSCs, the intrinsic high levels of this enzyme could also suggest a role in pluripotency via the ubiquitination of endogenous substrates. Interestingly, a study has reported that Ubr5 knockdown results in significant loss of pluripotency markers in mouse ESCs[14]. However, a different study did not find impairment of pluripotency markers in murine ESCs upon Ubr5 RNAi treatment[63]. Likewise, we did not find changes in the expression of pluripotency or germ layer markers in control hESCs/iPSCs upon loss of UBR5. Interestingly, knockdown of Ubr5 reduces the induction of Sonic hedgehog (Shh) in murine pluripotent stem cells upon retinoic-acid treatment[63]. In these lines, a conditional Ubr5 mutant mouse presents decreased hedgehog signaling during embryogenesis[63]. Although this model exhibits shorter limbs, the differences were not significant and the mutants do not have other obvious morphological defects[63]. Since our striatal neuronal differentiation protocol is based on the induction of hedgehog signaling, we tested whether UBR5 is necessary for the generation of MSNs from human iPSCs. However, loss of UBR5 did not affect their differentiation into MSNs, indicating that these cells conserved their ability to induce the hedgehog signaling. The distinct impact of UBR5 in pluripotency and differentiation could be associated with genetic differences between mouse and human species. Moreover, it is important to note the distinct pluripotent states exhibited by murine ESCs and hESCs/iPSCs[3]. Mouse ESCs are cultured in the presence of serum and leukemia inhibitory factor (LIF), and exhibit a naive state resembling the pluripotent state observed in the inner cell mass of the pre-implantation embryo[3]. On the other hand, hESCs as well as human iPSCs do not require LIF signaling and exhibit a more primed state that resembles post-implantation embryonic configurations[3].

Although loss of UBR5 did not impair pluripotency markers in human control iPSCs, it induced the formation of misfolded protein aggregates (i.e., aggresomes). However, we found that the accumulation of polyQ-expanded aggregates in HD-iPSCs does not impair pluripotency markers or induce their differentiation. Most importantly, these cells retain the ability to differentiate into neural cells that also accumulate polyQ-expanded aggregates. With these results, we speculate that increased proteostasis of pluripotent stem cells may be required to avoid the generation of precursor cells that accumulate protein aggregates at early organismal stages, a process that could be detrimental to organismal survival and healthspan. However, it is important to remark that most of our studies were performed in iPSCs, and further evidence in hESCs as well as in vivo experiments using mouse models are necessary to assess this intriguing possibility. On the other hand, HD-iPSCs with decreased levels of UBR5 can be further differentiated into MSNs with no aggregates. These results

could imply the activation of additional proteostatic control mechanisms to scavenge aggregates during terminal neuronal differentiation. It will be fascinating to define these mechanisms due to its implications for development and organismal aging.

Notably, genetic variations in a region of the chromosome 8 that contains the UBR5 gene, among others (i.e., RRM2B, MIR5680, NCALD), have been associated with an early onset of HD by genome-wide association analysis[37]. Whereas our results demonstrate that UBR5 levels decrease during differentiation, an intriguing possibility is that a further dysfunction of UBR5 activity with age could contribute to the onset of diseases such as HD. However, modeling HD as well as other neurodegenerative disorders using patient-specific neurons is challenging, as neurons differentiated from iPSCs lack aggregates and strong cell death phenotype[12,20,25,27]. Moreover, iPSC-derived neurons do not exhibit HTT mutant aggregates even after the addition of cellular stressors such as proteasome inhibitors. Although the relevance of UBR5 in HD pathology remains unclear due to these limitations, we find that ectopic expression of UBR5 is sufficient to reduce the protein levels of polyQ-expanded HTT and its aggregation in mutant HTT-overexpressing cell models. Moreover, UBR5 knockdown hastens the deleterious changes induced by polyQ-expanded expression in C. elegans models, providing direct evidence of a link between UBR5 activity and polyQ-expanded aggregation with age. It is important to note that these C. elegans models express polyQ-expanded fused to YFP but not HTT protein. Thus, these results suggest that UBR5 could also have a role in the proteostasis of other polyQ-containing proteins related with disease. However, we found that UBR5 knockdown does not impair polyQ-expanded ATXN3 levels and aggregation in MJD-iPSCs. Thus, we conclude that not all polyQ-containing proteins are regulated via UBR5 activity. In these lines, we observed that UBR5 does not impinge upon cellular localization and aggregation of FUS variants linked with ALS. Although these results suggest specificity of UBR5 for HTT regulation, we cannot discard a role in the proteostasis of other aggregation-prone proteins associated with disease. Notably, UBR5 downregulation is sufficient to induce the accumulation of aggresomes in control hESCs/iPSCs. Since aggresomes form from the accumulation of misfolded proteins when proteolytic systems are overwhelmed, UBR5 may also be involved in the degradation of misfolded proteins ensued from normal metabolism. Taken together, our studies in immortal pluripotent stem cells identified UBR5 as a determinant of their super-vigilant proteostasis.

## Methods

**hESC/iPSC lines and culture.** The H9 (WA09) and H1 (WA01) hESC lines were obtained from the WiCell Research Institute. The H9 and H1 hESCs used in our study matches exactly the known short tandem repeat (STR) profile of these cells across the 8 STR loci analyzed (Supplementary Table 3a). No STR polymorphisms other than those corresponding to H9 and H1[12] were found in the respective cell

**Fig. 9** UBR5 loss hastens aggregation and neurotoxicity in HD C. elegans models. **a** Western blot analysis of C. elegans with antibodies to GFP and β-actin loading control. The graph represents the Q67-YFP relative percentage values to Q67, vector RNAi-C. elegans corrected for β-actin loading control (mean ± s.e.m. of three independent experiments). **b** Filter trap analysis indicates that ubr-5 knockdown increases polyQ aggregates (detected by anti-GFP antibody) in C. elegans. The images are representative of four independent experiments. **c** Representative images of Q67-YFP aggregation under ubr-5 knockdown in whole adult worms. On the right, a higher magnification of head neurons is presented. For higher magnification of C. elegans mid-body, please see Supplementary Fig. 36. The images are representative of five independent experiments. Scale bar represents 100 μm. White arrows and red arrows indicate nerve ring and chemosensory processes, respectively. DIC, differential interference contrast. **d** Filter trap analysis with anti-GFP antibody of polyQ-expressing neuronal models in wild-type and rrf-3(pk1426) background. The images are representative of five independent experiments. **e** Loss of ubr-5 hastens the motility defects of polyQ67 worms. Bar graphs represent average ( ± s.e.m.) thrashing movements over a 30 s period on day 3 of adulthood (Q19 fed vector RNAi (n = 62) versus Q19 fed ubr-5 RNAi (n = 64), P = 0.11; Q19 fed vector RNAi versus Q67 fed vector RNAi (n = 65), P < 0.0001; Q67 fed vector RNAi versus Q67 fed ubr-5 RNAi (n = 65), P < 0.0001). All the statistical comparisons were made by Student's t-test for unpaired samples. **P < 0.01, ****P < 0.0001

lines, indicating correct hESC identity and no contamination with any other human cell line. The control iPSCs #3 and HD Q71-iPSC lines #1 were a gift from G.Q. Daley. These cells were generated using retroviral induction of c-Myc, Klf4, Oct4 and Sox2 and fully characterized for pluripotency in refs.[26,64]. HD Q71-iPSC line #2 (ND42230) was obtained from NINDS Human Cell and Data Repository (NHCDR) through Coriell Institute. These cells were generated from the same parental fibroblast as HD Q71-iPSC line #1 via episomal expression of l-Myc, Klf4, Oct4, Sox2 and LIN28 reprogramming factors. Likewise, ND42242 (control iPSC

line #1, Q21), ND36997 (control iPSC line #2, Q33), ND41656 (HD Q57-iPSC) and ND36999 (HD Q180-iPSC) were also obtained from NHCDR through Coriell Institute. Corrected isogenic counterparts of Q180-iPSCs (i.e., HD-C#1 and HD-C#2) were a gift from M.A. Pouladi[30]. By STR analysis, we confirmed correct genetic identity of the control and HD-iPSCs used in our study with the corresponding parental fibroblast lines when fibroblasts were available (that is, control iPSCs #1, control iPSCs #2, HD Q71-iPSCs, HD Q57-iPSCs and HD Q180-iPSCs; Supplementary Table 4). Parental fibroblasts GM02183 (control #2), GM04281

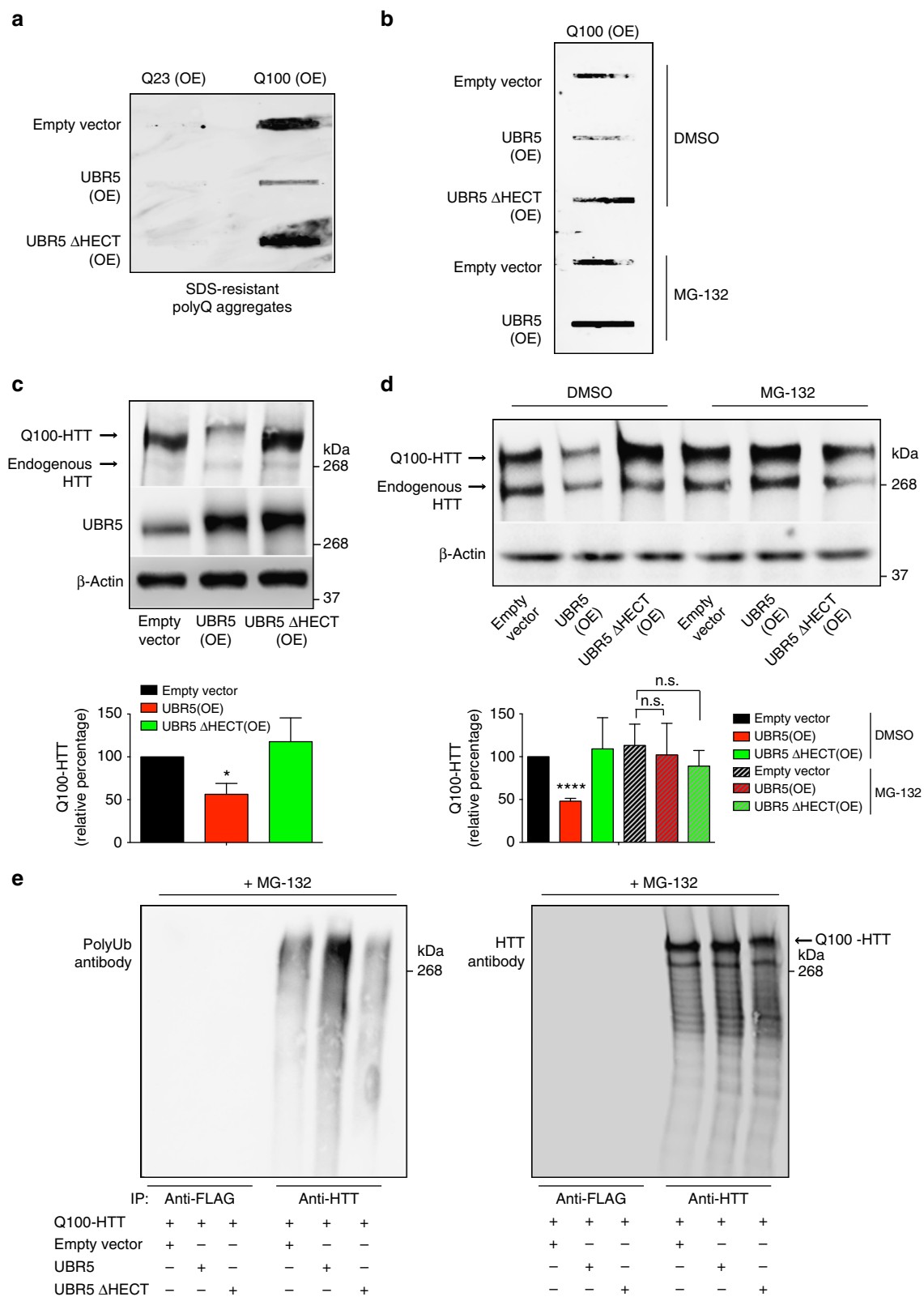

(HD Q71) and GM09197 (HD Q180) were obtained from Coriell Institute, whereas ND30014 (control #1) and ND33392 (HD Q57) fibroblasts were obtained from NHCDR through RUCDR Infinite Biologics at Rutgers University.

UKBi001-B (MJD Q74-iPSC line #1) and UKBi003-A (MJD Q74-iPSC line #2) were obtained from the European Bank for induced pluripotent Stem Cells (EBiSC). These MJD-iPSC lines were generated using retroviral expression of reprogramming factors (i.e., c-Myc, Klf4, Oct4 and Sox2) and fully characterized for pluripotency in ref.[48]. The MJD-iPSCs used in our study matches exactly the STR profile of the parental fibroblasts provided by the depositor of the lines (Supplementary Table 3b).

Control iPSCs #4 (FUS$^{wt/wt}$), ALS-iPSCs #1 (FUS$^{R521C/wt}$) and ALS-iPSCs #2 (FUS$^{P525L/P525L}$) were kindly provided by A. Rosa and I. Bozzoni. All of them were produced and characterized for pluripotency in ref.[49]. Briefly, control iPSCs #4 were derived from a control donor and checked for absence of mutation in FUS[49] whereas ALS-iPSCs #1 were generated from a patient affected by ALS in mid-late age[50]. Both iPSCs lines were established via lentiviral expression of c-Myc, Klf4, Oct4 and Sox2. ALS-iPSCs #2 were raised from control iPSCs #4 by TALEN (transcription activator-like effector nucleases)-directed mutagenesis and are homozygote for a FUS mutation (P525L) linked with severe and juvenile ALS[49]. We confirmed that the STR profile of the ALS-iPSCs #2 used in our experiments matches with the profile of control iPSCs #4 (Supplementary Table 4).

The hESCs and iPSCs were maintained on Geltrex (ThermoFisher Scientific) using mTeSR1 (Stem Cell Technologies). Undifferentiated hESC/iPSC colonies were passaged using a solution of dispase (2 mg ml$^{-1}$), and scraping the colonies with a glass pipette. Alternatively, cells were individualized with Accutase (1 unit ml$^{-1}$, Invitrogen) and seeded into Geltrex-coated plates for experiments with proteasome and autophagy inhibitors to facilitate homogenous treatment of cultures. All the hESC and iPSC lines used in our experiments had a normal diploid karyotype as assessed by single nucleotide polymorphism (SNP) genotyping (Supplementary Fig. 40). Moreover, all the cell lines used in this study were tested for mycoplasma contamination at least once every 3 weeks. No mycoplasma contamination was detected. Research involving hESC lines was performed with approval of the German Federal competent authority (Robert Koch Institute).

**SNP genotyping**. The molecular karyotype was analyzed by SNP genotyping with Illumina's HumanOmniExpressExome-8-v1.2 BeadArray (Illumina, Inc., San Diego, USA) at the Institute for Human Genetics (Department of Genomics, Life & Brain Center, University of Bonn, Germany). Processing was performed on genomic DNA following the manufacturer's procedures. Copy number regions were detected using the cnvPartition version 3.1.6.

**STR analysis**. STR analysis of H9 and H1 hESCs was conducted using the Promega PowerPlex 21 system (Promega Corporation) by Eurofins Genomics (Germany). We analyzed loci D5S818, D13S317, D7S820, D16S539, vWA, TH01, TP0X and CSF1P0 to compare with the known STR profile of these hESC lines[12]. The STR analysis of MJD-iPSC lines was also performed by Eurofins Genomics using the Promega PowerPlex 21 system.

Genotype comparison of control and HD-iPSCs lines with their parental fibroblasts was performed using the following microsatellite markers: D17S1303, D16S539, vWA, THO1, CSF1PO and TPOX. Fluorescently labeled PCR products were electrophoresed and detected on an automated 3730 DNA Analyzer and data were analyzed using Genemapper software version 3.0 to compare allele sizes between iPSCs and their parental fibroblasts (Applied Biosystems).

**Neural differentiation**. To obtain NPC cultures, we induced neural differentiation of iPSCs with STEMdiff Neural Induction Medium (Stem Cell Technologies) following the monolayer culture method[65]. Undifferentiated iPSCs were rinsed once with phosphate-buffered saline (PBS) and then we added 1 ml of Gentle Dissociation Reagent (Stem Cell Technologies) for 10 min. After the incubation period, we gently dislodged pluripotent cells and added 2 ml of Dulbecco's Modified Eagle Medium (DMEM)/F12 (ThermoFisher Scientific)+10 µM ROCK inhibitor (Abcam). Then, we centrifuged cells at 300 × g for 10 min. Cells were resuspended on STEMdiff Neural Induction Medium+10 µM ROCK inhibitor and plated on poly-ornithine (15 µg ml$^{-1}$)/laminin (10 µg ml$^{-1}$)-coated plates at a density of 200,000 cells cm$^{-2}$.

**Striatal neuron differentiation**. iPSCs were differentiated into striatal neurons by induction of hedgehog signaling pathway[56]. iPSCs were detached by incubating with dispase (1 mg ml$^{-1}$) for 20 min. The detached colonies were cultured in suspension as free-floating embryoid bodies (EBs) in the differentiation medium consisting of DMEM/F12, 20% knockout serum replacement, 100 µM β-Mercaptoethanol (Sigma), 1× minimum essential medium (MEM) non-essential amino acids and 2 mM L-glutamine. On day 4, the medium was replaced with a neural induction medium consisting of DMEM/F12, N2 supplement (ThermoFisher Scientific), 1× MEM non-essential amino acids, 2 mM glutamine and 2 µg ml$^{-1}$ heparin. On day 7, the EBs were attached to the laminin (ThermoFisher Scientific)-coated substrate in a 35 mm culture Petri dish and cultured in the neural induction medium. In the next week, the EBs flattened and columnar neuroepithelia organized into rosette appeared in the center of individual colonies. On day 12, 0.65 µM purmorphamine (Stem Cell Technologies) was added for 14 days (until day 25). From day 26, neuroepithelial spheres were dissociated with Accutase (1 unit ml$^{-1}$, Invitrogen) at 37 °C for 5 min and placed onto poly-ornithine/laminin-coated cover slips in Neurobasal medium containing a set of trophic factors, including brain-derived neurotrophic factor (20 ng ml$^{-1}$), glial-derived neurotrophic factor (10 ng ml$^{-1}$), insulin-like growth factor-1 (10 ng ml$^{-1}$), and cAMP (1 µM) (all from R&D Systems). DARPP32-expressing neurons appeared by day 32 as assessed by immunohistochemistry using Rabbit anti-DARPP32 (Abcam, ab40801, 1:50). We performed the experiments between days 32 and 35 of the differentiation protocol.

**Lentiviral infection of iPSCs**. Lentivirus (LV)-non-targeting small hairpin RNA (shRNA) control, LV-HTT shRNA (TRCN0000322961), LV-UBR5 shRNA #1 (TRCN0000003411), LV-UBR5 shRNA #2 (TRCN0000226458), LV-UBE3A shRNA (TRCN0000419838), LV-UBR7 shRNA (TRCN0000037025), LV-RNF181 shRNA #1 (TRCN0000364405), LV-RNF181 shRNA #2 (TRCN0000022389), LV-TRIM71 shRNA #1 (TRCN0000245956) and LV-TRIM71 shRNA #2 (TRCN0000245959) in pLKO.1-puro vector were obtained from Mission shRNA (Sigma). Transient infection experiments were performed as follows. iPSCs colonies growing on Geltrex were individualized using Accutase. Hundred thousand cells were plated on Geltrex plates and incubated with mTesR1 medium containing 10 µM ROCK inhibitor for 1 day. Then, cells were infected with 5 µl of concentrated lentivirus. Plates were centrifuged at 800 × g for 1 h at 30 °C. Cells were fed with fresh media the day after to remove the virus. After 1 day, cells were selected for lentiviral integration using 2 µg ml$^{-1}$ puromycin (ThermoFisher Scientific). Cells were split for further experiments and collected after 5–7 days of infection.

**Transfection of HEK293T cells**. HEK293T cells (ATCC) were plated on 0.1% gelatin-coated plates and grown in DMEM supplemented with 10% fetal bovine serum and 1% MEM non-essential amino acids (ThermoFisher Scientific) at 37 °C, 5% $CO_2$ conditions. Cells were transfected once they reached 80–90% confluency. Then, 1 µg GFP-UBR5 wild-type or GFP-UBR5 ΔHECT overexpression plasmid

---

**Fig. 10** Upregulation of UBR5 suppresses polyQ-expanded HTT aggregation. **a** Filter trap analysis with anti-polyQ-expansion diseases marker antibody of HEK293 human cells overexpressing (OE) Q23-HTT or Q100-HTT. Ectopic expression of UBR5 reduces the accumulation of aggregates in Q100-HTT(OE) cells. On the contrary, overexpression of a catalytic inactive UBR5 mutant (UBR5 ΔHECT) does not ameliorate polyQ-expanded aggregation. The images are representative of three independent experiments. **b** Filter trap analysis of Q100-HTT(OE) HEK293 cells. Proteasome inhibition with 0.5 µM MG-132 for 16 h blocks the reduction of polyQ-expanded aggregation induced by ectopic expression of wild-type UBR5. The images are representative of three independent experiments. **c** Western blot analysis of Q100-HTT(OE) HEK293 cells with antibodies to HTT and UBR5. Overexpression of wild-type UBR5 reduces the levels of Q100-HTT, whereas similar levels of UBR5 ΔHECT do not decrease mutant HTT levels. The graph represents the relative percentage values of Q100-HTT to empty vector cells corrected for β-actin loading control (mean ± s.e.m. of three independent experiments). **d** Western blot analysis of whole cell lysates from Q100-HTT(OE) HEK293 cultures with antibodies to HTT and β-actin loading control. Proteasome inhibition with 0.5 µM MG-132 for 16 h blocks proteasomal degradation of Q100-HTT levels induced by wild-type UBR5 overexpression. The graph represents the relative percentage values of Q100-HTT to DMSO-empty vector cells corrected for β-actin loading control (mean ± s.e.m. of three independent experiments). **e** Immunoprecipitation with anti-HTT and anti-FLAG antibodies in Q100-HTT(OE) HEK293. Immunoprecipitation was followed by western blot with antibodies to HTT and polyubiquitinated proteins (polyUb) to detect immunoprecipitated total HTT protein and polyUb-HTT, respectively. Prior to immunoprecipitation, cells were treated with proteasome inhibitor (0.5 µM MG-132, 16 h) to block the degradation of HTT induced by UBR5 so we could immunoprecipitate similar amounts of HTT for direct comparison of polyubiquitination among the distinct conditions. The images are representative of three independent experiments. All the statistical comparisons were made by Student's t-test for unpaired samples. *P < 0.05, ****P < 0.0001

and 0.5 µg pARIS-mCherry-httQ23-GFP or pARIS-mCherry-httQ100-GFP were used for transfection, using Fugene HD (Promega) following the manufacturer's instructions. After 36 h of incubation in normal medium, the cells were harvested for further experiments. GFP-UBR5 wild-type and GFP-UBR5 ΔHECT over-expression plasmids (Addgene plasmids #52050 and # 52051, respectively) were a gift from D. Saunders and were first published in ref.[61]. The pARIS-mCherry-httQ23-GFP and pARIS-mCherry-httQ100-GFP[66] plasmids were a gift from F. Saudou.

**Analysis of β-amyloid aggregation.** N-terminal c-Myc-tagged β23 protein construct was a gift from F.U. Hartl[67]. For mammalian expression, c-Myc-tagged β23 sequence was cloned into pCDH-CMV-MCS-EF1 from pcDNA3.1 using XbaI and BamHI enzymes. Cell fractionation was performed following the protocol described in ref.[67]. Briefly, cells were collected and resuspended in lysis buffer (50 mM Tris (pH 7.8), 150 mM NaCl, 1% (v/v) NP40, 0.25% sodium deoxycholate, 1 mM EDTA and 1 tablet protease inhibitor cocktail (Roche) per 10 ml). The cell lysate was then centrifuged at $20,000 \times g$ for 30 min at 4 °C. The supernatant was applied for analyzing soluble fraction and pellets were dissolved in SDT buffer (4% (w/v) sodium dodecyl sulfate (SDS), 100 mM Tris-HCl (pH 7.6) and 0.1 M dithiothreitol (DTT)). Anti-Myc antibody (Proteintech, #66004-1-Ig, 1:2000) was used for western blot to detect c-Myc-tagged β23 protein.

**Protein immunoprecipitation for interaction analysis.** iPSCs were lysed in RIPA buffer (50 mM Tris-HCl (pH 7.4), 150 mM NaCl, 1% Triton X-100, 1% sodium deoxycholate, 0.1% SDS, 1 mM EDTA, 1 mM phenylmethylsulfonyl fluoride (PMSF)) supplemented with protease inhibitor cocktail (Roche). Lysates were homogenized by passing 10 times through a 27-gauge (27 G) needle attached to a 1 ml syringe and centrifuged at $13,000 \times g$ for 15 min at 4 °C. After pre-clearing the supernatant with Protein A agarose beads (Pierce), the samples were incubated overnight with UBR5 antibody (Cell Signaling, #8755, 1:50) on the overhead shaker at 4 °C. As a control, the same amount of protein was incubated with anti-FLAG (SIGMA, F7425, 1:50) or anti-UBE3A antibody (Cell Signaling, #7526, 1:50) in parallel. Subsequently, samples were incubated with 30 µl of Protein A beads for 1 h at room temperature. After this incubation, samples were centrifuged for 5 min at $5,000 \times g$ and the pellet was washed three times with RIPA buffer. For elution of the proteins, the pellet was incubated with 2× Laemmli Buffer, boiled for 5 min and centrifuged for 5 min at maximum speed. The supernatant was taken and loaded onto a sodium dodecyl sulfate–polyacrylamide gel electrophoresis (SDS–PAGE) gel for western blot analysis.

**Protein immunoprecipitation for assessing polyubiquitination.** HEK293 cells were lysed in protein lysis buffer (50 mM Tris-HCl (pH 6.7), 150 mM NaCl, 1% NP40, 0.25% sodium deoxycholate, 1 mM EDTA, 1 mM PMSF, 1 mM Na$_3$VO, 1 mM NaF) supplemented with protease inhibitor cocktail. The samples were homogenized through a syringe needle (27 G) and centrifuged at $13,000 \times g$ for 15 min at 4 °C. The samples were incubated for 30 min with HTT antibody (Cell Signaling, ab#5656, 1:1000) on ice. As a negative control, the same amount of protein was incubated with anti-FLAG antibody (SIGMA, F7425) in parallel. Subsequently, samples were incubated with 50 µl of µMACS Micro Beads for 1 h at 4 °C with overhead shaking. After this incubation, samples were loaded to pre-cleared µMACS column (#130-042-701). Beads were washed three times with 50 mM Tris (pH 7.5) buffer containing 150 mM NaCl, 5% glycerol and 0.05% Triton and then washed five times with 50 mM Tris (pH 7.5) and 150 mM NaCl. For protein elution, the beads were incubated with 1× Laemmli Buffer for 5 min and collected into tubes. The samples were boiled for 5 min at 95 °C and loaded in a SDS–PAGE gel for western blot analysis.

**Proteomics analysis of HTT ubiquitination sites.** HEK293 cells were lysed in protein lysis buffer (50 mM Tris-HCl (pH 6.7), 150 mM NaCl, 1% NP40, 0.25% sodium deoxycholate, 1 mM EDTA, 1 mM NaF) supplemented with protease inhibitor cocktail. Lysates were homogenized through syringe needle (27 G) and centrifuged at $13,000 \times g$ for 15 min at 4 °C. The samples were incubated for 30 min with HTT antibody (Cell Signaling, ab#5656, 1:1000) on ice. Subsequently, samples were incubated with 50 µl of µMACS Micro Beads for 1 h at 4 °C with overhead shaking. After this incubation, samples were loaded to pre-cleared µMACS column (#130-042-701). Beads were washed three times with 50 mM Tris (pH 7.5) buffer containing 150 mM NaCl, 5% glycerol and 0.05% Triton and then washed five times with 50 mM Tris (pH 7.5) and 150 mM NaCl. Then, columns were subjected to in-column tryptic digestion containing 7.5 mM ammonium bicarbonate, 2 M urea, 1 mM DTT and 5 ng ml$^{-1}$ trypsin. Digested peptides were eluted using two times 50 µl of elution buffer 1 containing 2 M urea, 7.5 mM Ambic, and 5 mM iodoacetamide. Digests were incubated overnight at room temperature with mild shaking in the dark. Samples were stage-tipped the next day for label-free quantitative proteomics. All samples were analyzed on a Q-Exactive Plus (Thermo Scientific) mass spectrometer that was coupled to an EASY nLC 1200 UPLC (Thermo Scientific)[68]. Peptides were loaded with solvent A (0.1% formic acid in water) onto an in-house packed analytical column (50 cm × 75 µm I.D., filled with 2.7 µm Poroshell EC120 C18, Agilent). Peptides were chromatographically separated at a constant flow rate of 250 nl min$^{-1}$ using 150 min methods: 5–30%

solvent B (0.1% formic acid in 80% acetonitrile) within 119 min, 30–50% solvent B within 19 min, followed by washing and column equilibration. The mass spectrometer was operated in data-dependent acquisition mode. The MS1 survey scan was acquired from 300 to 1750 $m/z$ at a resolution of 70,000. The top 10 most abundant peptides were subjected to higher collisional dissociation fragmentation at a normalized collision energy of 27%. The AGC (automatic gain control) target was set to 5e5 charges. Product ions were detected in the Orbitrap at a resolution of 17,500.

All mass spectrometric raw data were processed with Maxquant (version 1.5.3.8) using default parameters. Briefly, MS2 spectra were searched against the human Uniprot database, including a list of common contaminants. False discovery rates (FDRs) on protein and peptide–spectrum match (PSM) level were estimated by the target-decoy approach to 0.01% (Protein FDR) and 0.01% (PSM FDR) respectively. The minimal peptide length was set to 7 amino acids and carbamidomethylation at cysteine residues was considered as a fixed modification. Oxidation (M), GlyGly (K) and Acetyl (Protein N-term) were included as variable modifications. The match-between runs option was enabled. Label-free quantification (LFQ) was enabled using default settings. The resulting output was processed using Perseus as follows: protein groups flagged as "reverse", "potential contaminant" or "only identified by site" were removed from the proteinGroups.txt. LFQ values were log2 transformed. Missing values were replaced using an imputation-based approach (random sampling from a normal distribution using a down shift of 1.8 and a width of 0.3). Significant differences between the groups were assessed using Student's $t$-test. A permutation-based FDR approach was applied to correct for multiple testing.

**Western blot.** Cells were scraped from tissue culture plates by cell scraping and lysed in protein cell lysis buffer (50 Mm Hepes pH 7.4, 150 Mm NaCl, 1 mM EDTA, 1% Triton X-100) supplemented with 2 mM sodium orthovanadate, 1 mM PMSF and protease inhibitor mix). Lysates were homogenized by syringe needle (27 G) followed by centrifugation at $8000 \times g$ for 5 min at 4 °C and then supernatants were collected (with the exception of the western blots showed in Fig. 6d, where we loaded whole cell lysates without centrifugation because proteasome inhibition induced high concentration of Q100-HTT (OE) in the pellet fraction). Protein concentrations were determined with a standard BCA protein assay (ThermoFisher Scientific). Approximately 30 µg of total protein was separated by SDS–PAGE, transferred to polyvinylidene difluoride membranes (Millipore) and subjected to immunoblotting. Western blot analysis was performed with anti-UBR5 (Stem Cell Technologies, #60094, 1:1000), anti-UBE3A (Cell Signaling, ab#7526, 1:1000), anti-RNF181 (ThermoFisher Scientific, PA5-31008, 1:2000), anti-UBR7 (ThermoFisher Scientific, PA5-31559, 1:1000), anti-ATXN3 (Merck, MAB5360, 1:500), anti-FUS (Abcam, #154141, 1:1000), anti-GFP (Immunokontakt, 210-PS-1GF, 1:5000), anti-polyubiquitinylated conjugates (Enzo, PW8805-0500, 1:1000), anti-p62 (Progen, GP62-C, 1:1000), anti-LC3 (Sigma, L7543, 1:1000), anti-DARPP32 (Abcam, #40801, 1:1000) and anti-β-actin (Abcam, #8226, 1:5000). To detect anti-HTT (Cell Signaling, ab#5656, 1:1000), a monoclonal antibody produced by immunizing animals with a synthetic peptide corresponding to residues surrounding Pro1220 of human HTT protein. To detect mutant HTT, we used anti-polyQ-expansion diseases marker (Millipore, MAB1574, 1:1000), a monoclonal antibody raised against TATA-binding protein that recognizes peptides overlapping the polyQ stretch of this protein. This antibody also recognizes polyQ-containing proteins such as HTT and ATXN3 with the remarkable property of detecting much better the polyQ-expanded pathological proteins than the wild-type proteins[12,31]. Uncropped versions of all important western blots are presented in Supplementary Fig. 41.

**Immunocytochemistry.** Cells were fixed with paraformaldehyde (4% in PBS) for 20 min, followed by permeabilization (0.2% Triton X-100 in PBS for 10 min) and blocking (3% bovine serum albumin in 0.2% Triton X-100 in PBS for 10 min). Human iPSCs/NPCs were incubated in primary antibody for 2 h at room temperature (Mouse anti-polyQ (Millipore, MAB1574, 1:50), Mouse anti-OCT4 (Stem Cell Technologies, #60093, 1:200), Rabbit anti-PAX6 (Stem Cell Technologies, #60094, 1:300), Mouse anti-FUS (Abcam, #154141, 1:500), Rabbit anti-G3BP1 (MBL, #RN048PW, 1:500), Rabbit anti-DARPP32 (Abcam, #40801, 1:50), Rabbit anti-GABA (Sigma, #A2052, 1:100) and Chicken anti-MAP2 (Abcam, #5392, 1:500). Then, cells were washed with 0.2% Triton-X/PBS and incubated with secondary antibody (Alexa Fluor 488 Goat anti-Mouse (ThermoFisher Scientific, #A-11029, 1:500), Alexa Fluor 568F(ab')2 Fragment of Goat Anti-Rabbit IgG (H+L) (ThermoFisher Scientific, #A-21069, 1:500)), Alexa Fluor 647 Donkey anti-Chicken (Jackson ImmunoResearch, #A-703-605-155, 1:500) and Hoechst 33342 (Life Technologies, #1656104) for 1 h at room temperature. PBS and distilled water wash were followed before the cover slips were mounted on Mowiol (Sigma, #324590).

**Aggresome detection.** Pluripotent stem cells were cultured at 37 °C or under heat stress conditions at 42 °C for 4 h. Aggresome formation was detected by PROTEOSTAT Aggresome Detection Kit (ENZO, #ENZ-51035), following the manufacturer's instructions. Briefly, the cells were fixed in 4% formaldehyde solution and permeabilized with 0.5% Triton X-100, 3 mM EDTA, pH 8.0, in 1× assay buffer from the detection kit. Then, aggresomes were stained with PROTEOSTAT

Aggresome Detection reagent, including Hoechst 33342 for nuclear staining. Imaging of aggresomes was performed using a standard rhodamine filter set.

**Propidium iodide staining**. Cells were incubated with 2 μg ml$^{-1}$ propidium iodide (Sigma, #81845) and Hoechst 33342 (Life Technologies, #1656104) for 1.5 h. Then, cover slips were mounted using FluorSave Reagent (Calbiochem, #345789).

**TUNEL staining**. TUNEL (terminal deoxynucleotidyl transferase dUTP nick end labeling) measurement was performed with the In situ BrdU-Red DNA Fragmentation (TUNEL) Assay Kit (Abcam, #66110) according to the manufacturer's protocol. Briefly, cells were washed twice and stained with TdT enzyme and BrdUTP for 1 h at 37 °C. A second incubation with an Anti-BrdU-Red antibody was performed at room temperature for 30 min. Nuclei were stained with Hoechst 33342 (Life Technologies, #1656104).

**RNA isolation and quantitative reverse transcription-PCR**. Total RNA was extracted using RNAbee (Tel-Test Inc.). Complementary DNA (cDNA) was generated using qScript Flex cDNA synthesis kit (Quantabio). SybrGreen real-time quantitative PCR (qPCR) experiments were performed with a 1:20 dilution of cDNA using a CFC384 Real-Time System (Bio-Rad) following the manufacturer's instructions. Data were analyzed with the comparative $2\Delta\Delta C_t$ method using the geometric mean of *ACTB* and *GAPDH* as housekeeping genes. See Supplementary Table 5 for details about the primers used for this assay.

**26S proteasome fluorogenic peptidase assays**. Cells were collected in proteasome activity assay buffer (50 mM Tris-HCl, pH 7.5, 250 mM sucrose, 5 mM MgCl$_2$, 0.5 mM EDTA, 2 mM ATP and 1 mM DTT) and lysed by passing 10 times through a 27 G needle attached to a 1 ml syringe needle. Lysates were centrifuged at 10,000 × *g* for 10 min at 4 °C. Then, 25 μg of total protein of cell lysates were transferred to a 96-well microtiter plate (BD Falcon) and incubated with the fluorogenic proteasome substrate. To measure the chymotrypsin-like activity of the proteasome we used either Z-Gly-Gly-Leu-AMC (Enzo) or Suc-Leu-Leu-Val-Tyr-AMC (Enzo). We used Z-Leu-Leu-Glu-AMC (Enzo) to measure the caspase-like activity of the proteasome, and Ac-Arg-Leu-Arg-AMC (Enzo) for the proteasome trypsin-like activity. Fluorescence (380 nm excitation, 460 nm emission) was monitored on a microplate fluorometer (EnSpire, Perkin Elmer) every 5 min for 1 h at 37 °C.

**Quantitative proteomics analysis of E3 enzymes**. For the characterization of protein expression differences in E3 enzymes comparing H9 hESCs with their neuronal counterparts, we analyzed quantitative proteomics data[35] available via ProteomeXchange with identifier PXD007738. Then, we intersected the annotated human E3 network from KEGG (Kyoto Encyclopedia of Genes and Genomes) database[69] with this proteomics dataset. Statistical comparisons were made by Student's *t*-test. FDR-adjusted *P* value (*q* value) was calculated using the Benjamini–Hochberg procedure.

**C. elegans strains and maintenance**. *C. elegans* strains were maintained at 20 °C using standard methods[17]. AM23 (*rmIs298[pF25B3.3::Q19::CFP]*) and AM716 (*rmIs284[pF25B3.3::Q67::YFP]*) strains were a gift from R. I. Morimoto. For the generation of the strains DVG144 (*rmIs298[F25B3.3p::Q19::CFP]; rrf-3(pk1426)*) and DVG145 (*rmIs284[pF25B3.3::Q67::YFP]; rrf-3(pk1426)*), NL2099 strain (*rrf-3 (pk1426)*) was crossed to AM23 and AM716, respectively. Screening of *rrf-3 (pk1426)* worms was done by PCR using the forward primer F: GTTTTGACGC CAAACGGTGA and two reverse primers: TGCAGCATGTCCAGACACAA, which outflanks the deleted region in *rrf-3(pk1426)*, and CCATTCTGTGCACGT TTCCA, which binds inside the deletion.

**RNAi experiments in C. elegans**. RNAi-treated strains were fed *Escherichia coli* (HT115) containing an empty control vector (L4440) or expressing double-stranded *ubr-5* RNAi. *ubr-5* RNAi construct was obtained from the Ahringer RNAi library and sequence verified.

**Imaging of polyQ aggregates in C. elegans**. AM716 and AM23 strains were grown at 20 °C until L4 stage and then grown at 25 °C on *E. coli* (HT115) containing either empty control vector or *ubr-5* RNAi until day 3. Day 3 adult worms were immobilized following the protocol described in ref.[70]. Briefly, worms were placed on 5% agarose-containing pads on a suspension of polystyrene beads (Polyscience, 2.5% by volume). For imaging, we used a Meta 710 confocal microscope (Zeiss) at the CECAD Imaging Facility.

**Quantitative fluorescence recovery after photobleaching**. Q67-YFP day 3 adult worms were immobilized on 5% agarose pads using 0.1% sodium azide. After three prebleaching scans, a constant region of interest (ROI) (44.29 × 30.09 μm) was bleached for 20 scans (860 ms per iteration) in a SP8 Confocal Microscope (Leica). Directly after bleaching, the fluorescence recovery was sampled once every 2 s for 90 times. Average fluorescence intensities within ROIs were measured under the

same condition for empty vector and *ubr-5* RNAi-treated worms using ImageJ software. The half-life of fluorescence recovery (t1/2) was determined by curve fitting of experimental data using the following exponential equation: $I(t) = a(1 - e^{(\tau \times t)})$.

**Motility assay**. Animals were grown on *E. coli* (OP50) bacteria at 20 °C until L4 stage and then transferred to 25 °C and fed with *E. coli* (HT115) bacteria containing empty control vector or *ubr-5* RNAi for the rest of the experiment. At day 3 of adulthood, worms were transferred to a drop of M9 buffer and after 30 s of adaptation the number of body bends was counted for 30 s. A body bend was defined as change in direction of the bend at the mid-body[58].

**Filter trap**. AM716, AM23, DVG144 and DVG145 *C. elegans* strains were grown at 20 °C until L4 stage and then grown at 25 °C on *E. coli* (HT115) bacteria containing either empty control vector or *ubr-5* RNAi for the rest of the experiment. Day 3 adult worms were collected with M9 buffer and worm pellets were frozen with liquid N2. Frozen worm pellets were thawed on ice and worm extracts were generated by glass bead disruption on ice in non-denaturing lysis buffer (50 mM Hepes pH 7.4, 150 mM NaCl, 1 mM EDTA, 1% Triton X-100) supplemented with EDTA-free protease inhibitor cocktail (Roche). Worm and cellular debris was removed with 8000 × *g* spin for 5 min. Approximately 100 μg of protein extract was supplemented with SDS at a final concentration of 0.5% and loaded onto a cellulose acetate membrane assembled in a slot blot apparatus (Bio-Rad). The membrane was washed with 0.2% SDS and retained Q67-GFP was assessed by immunoblotting for green fluorescent protein (GFP; ImmunoKontakt, 210-PS-1GFP, 1:5000). Extracts were also analyzed by SDS–PAGE with GFP antibody to determine protein expression levels.

iPSCs were collected in non-denaturing lysis buffer supplemented with EDTA-free protease inhibitor cocktail and lysed by passing 10 times through a 27 G needle attached to a 1 ml syringe. Then, we followed the filter trap protocol described above. Cell pellet lysates were loaded after solubilization with 2% SDS. The membrane was washed with 0.2% SDS and retained polyQ proteins were assessed by immunoblotting for anti-polyQ-expansion diseases marker antibody (Millipore, MAB1574, 1:5000). Extracts were also analyzed by SDS–PAGE to determine HTT protein expression levels.

**Data availability**. Proteomics data of HTT ubiquitination sites have been deposited to the ProteomeXchange Consortium via the PRIDE partner repository with the dataset identifier PXD009803. To define E3 enzymes significantly increased in hESC, we analyzed available quantitative proteomics data from PRIDE: PXD007738 [35]. All the other data are also available from the corresponding author upon reasonable request.

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

## Acknowledgements

This work was supported by the European Research Council (ERC Starting Grant-677427 StemProteostasis), Else Kröner-Fresenius-Stiftung (2015_A118), Alexander von Humboldt Stiftung and Deutsche Forschungsgemeinschaft (DFG) (CECAD and VI742/

1-1). This work was also supported by the European Research Council (ERC Consolidator Grant-616499) to T.H. In addition, this article is based upon work from COST Action (PROTEOSTASIS BM1307). W.P. thanks the Polish National Science Center (grant UMO-2016/23/B/NZ3/00753). We are grateful to Mahmoud A. Pouladi for corrected isogenic lines of HD Q180-iPSCs. We would also like to thank Alessandro Rosa and Irene Bozzoni for ALS-iPSC lines. We thank the CECAD Proteomics and Imaging Facilities for their advice and contribution to proteomics and imaging experiments, respectively.

## Author contributions

S.K. and I.S. performed most of the experiments, data analysis and interpretation through discussions with D.V., H.J.L. performed analysis of aggresomes/β-amyloid aggregation and helped with other experiments. R.G.-G. carried out *C. elegans* motility and imaging experiments. W.P. contributed to HTT ubiquitination assays and provided critical advice for other experiments. A.F. performed analysis of pluripotent stem cell lines and helped with neuronal differentiation experiments. T.H. contributed with his knowledge in proteostasis and provided critical advice for the project. D.V. planned and supervised the project. The manuscript was written by D.V. All the authors discussed the results and commented on the manuscript.

## Additional information

**Competing interests:** The authors declare no competing interests.

