## [Peer Review File · Nature Communications]

Reviewers' comments:

Reviewer #1 (Remarks to the Author):

iPSCs derived from controls and HD patients have been used widely as in vitro cell models to facilitate discovery of novel targets and treatments for human HD without the ethical limitations. In this manuscript, the authors used HD-iPSCs lines that carry different CAG repeat lengths (71 and 180 CAGs) in the Huntington disease gene for study and found that the increased proteasome activity of iPSCs can regulate HTT levels and polyQ expanded HTT aggregation in these cells. More importantly, they found the intrinsic high levels of the E3 ubiquitin ligase UBR5 are a key component to maintain proteostasis of HTT in HD-iPSCs thus provide some interesting evidence to explain why HD-iPSCs lack the aggregate phenotype characteristic of HD. Although the findings in the manuscript might be highly interesting to the field, the paper seems to suffer from some significant problems as listed below.

Major issues:

1. It is expected that proteostasis in iPSCs is different from that in differentiated neuronal cells. As shown by the paper, a number of proteins related to proteostasis are reduced when iPSCs are differentiated. It is also well-known that proteasome activity determines the level of mutant HTT. The more interesting issue is how specific is UBR5's regulation on HTT accumulation. The comparison between UBR5 and other related proteins on HTT accumulation is very limited. Also, whether UBR5 influences other protein accumulation remains unknown. Without the above information, the impact of this paper is quite minimal.

The authors claimed that UBR5 is a central modulator of mutant HTT aggregation in iPSCs. Does UBR5 also regulate the accumulation of other proteins, especially other misfolded proteins? Western blot results actually show that inhibiting UBR5 also increases actin expression on some blots. UBR5 and other proteins may increase intrinsic higher UPS activities in iPSCs, but how critical UBR5 is for this higher activity remains unknown.

2. It is known that the impairment of clearance of misfolded or mutant proteins is key to their accumulation. There is no evidence that UBR5 selectively regulates the level of mutant HTT, and western blotting in Fig. 1e did not show the selective increase of expanded Htt, as the upper band (mutant HTT) had the same changes as the lower band (normal HTT). Fig 2h does not show any specific precipitation of mutant HTT with UBR-5. This raises a significant issue of how UBR5 regulates both normal and mutant HTT. Without knowing this, the conclusion that that UBR5 is a central modulator of mutant HTT aggregation in iPSCs is weak.

Other issues:

1. It is interesting to see that loss of UBR5 hastened the aggregation phenotypes in HD neurons differentiated from HD-iPSCs. In Fig5b and Sup Fig 5, it seemed that Q71 neurons formed more aggregates than Q180 neurons, quantitative data should be added to make it more convincing and explain this phenomenon. Do the increased aggregates in HD neurons have toxicity and affect the cell line viability when knocking down UBR5? Does the polyQ length (71 Q and 180 Q) have relation with increased aggregates in HD neurons? More evidence should be provided in HD neurons since neuronal cells are more vulnerable than other cells in HD.

2. In Fig 5a and b, the HD neurons used are induced striatal neurons, but there is lack the DARPP32 staining or other markers that can prove the successful induction of HD iPSCs to striatal neurons. Moreover, the authors found that UBR5 knockdown did not influence the iPSCs differentiation. Lack of the convincing figures for neuronal induction weakens the conclusion of this manuscript.

3. Please provide more detail information for the HD iPSCs cell lines used in this manuscript, including the HD patient donors, onset ages, and symptoms.
4. The authors described in Fig (1e,1f) that the differences between the protein levels of normal and mHTT were more pronounced in HD-iPSCs that express longer polyQ repeats(Q180). It would be more convincingly shown if both Q71 and Q180 samples were on the same blot to compare the difference. It seemed that there are no significant changes with or without MG132 for the mHTT bands using HTT antibody in Q180 cell lines of Fig 1f.
5. In Fig1a, the nuclei Hoechst staining in control iPSCs seemed much less than HD iPSCs and the magnifications are different between the two groups. Please make the figures clearer and use the same magnification.
6. The entire blot of HTT needs to be presented to see how degraded HTT products are changed by UBR5.
7. Fig 2h does not show any specific precipitation of mutant HTT with UBR-5. Since the authors can detect both normal and mutant HTT on the same western blot and their sizes are distinguishable, the authors need to compare immunoprecipitation of both normal and mutant HTT with UBR5 on the same western blot.
8. Figure 3, UBR5 shRNA treatment did not show clearly the increase in HTT aggregates that are often localized in the nuclei. It seems that soluble cytoplasmic HTT, but not nuclear HTT aggregation, was increased in the presented images.
9. MG132 treatment for 12 h may reduce cell viability to increase HTT accumulation and aggregation. The authors need to rule out this possibility.

Reviewer #2 (Remarks to the Author):

The ubiquitin ligase UBR5 suppresses proteostasis collapse in immortal pluripotent stem cells and Huntington's disease models

Seda Koyuncu, Isabel Saez, Ricardo Gutierrez-Garcia, Wojciech Pokrzywa, Azra Fatima, Thorsten Hoppe, and David Vilchez

A. Summary of the key results

In this paper the authors have for the first time shown that UBR5 ubiquitin ligase suppresses a breakdown of protein homeostasis due to upregulation of UBR5 in iPS cells. They also show some evidence of how these enhanced proteostatic capabilities of PSCs may be useful creating better iPSC models of Huntington's Disease poly-Q repeat expansion disorder. They then complement their human PSC data using *C. elegans* model of HD. The advances described in this study are worthy of publication in Nature Communications after some additional data and revisions suggested below.

B. Originality and significance: if not novel, please include reference

This study is of great significance and novelty is medium. The same group has described similar the importance of similar mechanisms in Nature Communications 2016 article "Somatic increase of CCT8 mimics proteostasis of human pluripotent stem cells and extends *C. elegans* lifespan".

There are other studies that describe a similar important role of UBR5 in pluripotent stem cells.

Buckley et al. Cell Stem Cell 2012. Regulation of Pluripotency and Cellular Reprogramming by the Ubiquitin-Proteasome System

Sanchez et al. PNAS 2016. BMI1-UBR5 axis regulates transcriptional repression at damaged chromatin

C. Data & methodology: validity of approach, quality of data, quality of presentation

The overall quality and style of presenting the data is and approach taken is valid. Experiments have been carried out with precision and appropriate set of controls is included. The weakness in this study are highlighted below. At least all experiments should be conducted with the allelic series of lines in this study – Q21, Q33, Q71, Q180 to reach statistically sound conclusions. The authors could also bolster their conclusions by using multiple clonal HD iPSC lines is from the CAG repeat donor. This is a norm for iPSC disease modeling field now.

Most importantly, this paper has not demonstrated evaluation of these molecular mechanisms in relevant terminally differentiated striatal neurons that degenerate in human HD patients. There is lack of characterization of the striatal markers in the neuronal differentiation experiments.

Therefore, it is important to demonstrate relevance of disturbing the proteostasis and poly Q aggregates in striatal neurons.

There is also a lack of in-lab HD-iPSC pluripotency validation data described in supplementary figures. Even though iPSCs may be procured from another lab, basic iPSC QC within the lab is required to ensure stable cytogenetics. G-band karyotype (cytogenetic stability) and STR identity analysis is required on all the PSCs used in this study including the hESC, as well as the HD-iPSCs.

D. Suggested improvements: experiments, data for possible revision

1. The paper suggests that the discoveries for the role of UBR5 observed here in the iPSCs is also applicable to pluripotent stem cells in general, including hESCs. However, this could be exclusive to iPSCs and may be because of somatic cell reprogramming the associated stress during low efficiency reprogramming of somatic cells into clonal iPSCs.

2. A better description of the antibodies for the normal and expanded HTT is required in the methods section including the epitopes and validation of these antibodies.

3. Figure legend 2C. Knock down of CCT6A and CCT7 does not appear to induce expression of endoderm markers at all. Rather it should say ectoderm. Please correct this. Also, it is rather curious as to why the knockdown does not affect mesoderm and endoderm expression in 2C. The authors need to explain clearly the rationale behind this?

4. Utilize isogenic mutation expansion (Q71 and Q180 alleles corrected in iPSCs by CRISPR) corrected iPSCs to validate these observations.

5. In general, the ICC images of poly Q aggregates of are not very high quality: Please acquire higher resolution images with greater magnification.

6. Figure 1d: It is not clear why the Q33 control iPSC line has more SDS-resistant poly Q aggregates in DMSO condition compared to the Q71 and Q71 HD iPSC lines. There should be minimal or no aggregates in the control iPSC line or at least equal to the HD iPSC lines in DMSO condition, even if UBR5 levels are high in iPSC stage. Is the filter trap assay an artifact? Also, how is the loading and intensity of the bands normalized in this assay?

7. All Western blots, please quantify the bands by densitometry and normalize to b-actin loading.

8. Throughout the manuscript, the authors pick and choose which HD iPSC line and control iPSC line to use in experiments. The authors keep randomly switching between Q71 and Q180 as well as different clones of these iPSCs for different experiments and figures. Please be consistent and include data from the Q71 and Q180 lines in all experiments.

9. Figure 1e: Why does MG132 treatment increase levels of both normal and mHTT protein?

10. What was the rationale for ignoring the TRIM71 gene from Supplementary Table 1? This was the highest upregulated E3 ubiquitin ligase with 17-fold up regulation in hESCs compared to neurons. Why UBR5 chosen for the studies when it was relatively lower fold up-regulation.

11. It is a bit annoying that there was no consistency in the differentiated states of neural cell types evaluated in the manuscript. In some figures iPSCs and NPCs were evaluated, while in other neurons were included and then sometimes terminally differentiated iPSC-neurons were ignored for analysis

12. Figure 2b: Why is data from terminally differentiated neurons missing here? Please include.

13. Figure 2g: I don't think that the authors can directly conclude from this experiment that "UBR5 is involved in proteasomal degradation of HTT"

14. It would be nice see consistency in the cell types evaluated.

15. What time point of differentiation of iPSC-neurons was chosen for the analysis shown here?

16. What is the difference between HD-iPSC #1 and #2, both with Q71 repeats?

17. In the discussion, please state why do the authors think the role of UBR5 protein is specific to regulating homeostasis of HTT protein only and not any other protein with repeat poly-Q expansions?

18. Figure 3b: Filter trap experiment results are not consistent with Figure 1d. The NT shRNA results for the 3 lines are not consistent with the DMSO results Fig. 1D for the different iPSCs.

19. Clarify the statistics as stated below.

20. Figure 3e: Please label Figure 3e appropriately to clarify samples with MG132 treatment and those without.

21. Please incorporate allelic series of CAG repeat control and HD iPSC lines – Q21, Q33, Q71 and Q180 in all experiments.

22. Figure 4c: Please include data from differentiated striatal neurons as well.

23. Figure 4c: UBR shRNA#2 condition Merge panel is incorrect and has not merged channels. Please rectify.

24. Figure 4f: Please include staining with the expanded poly Q specific antibody as well.

25. Figure 5: Please include validation of striatal neuron differentiation and cultures using western blot and ICC co-staining with specific markers of striatal neurons such as DARPP32. In fact, throughout the manuscript the striatal neuron data is very important to the conclusions of the manuscript.

26. Figure 6b: Filter trap results are not convincing. Figure 6a is convincing. Again, there are inconsistencies between 6a and 6b.

27. Figure 6d: b-actin loading in MG132 treatment samples is not consistent with DMSO control. It is likely that those samples are over-loaded 2-3x over DMSO control based on band intensity. Thus, the treatments are not directly comparable.

28. Figure 6d: It appears that MG132 treatment appears to decrease Q100 HTT in UBR overexpressing samples compared to the empty vector. Please explain this observation?

29. Figure 6e: Results text does not match the figure. Should there be MG132 treatment as "+" in all the samples. Please explain this figure better in the text and figure legend. The HTT antibody blot does not appear to match the Figure 6a result.

30. There is also a lack of in-lab HD-iPSC pluripotency genetic integrity data over multiple passages, which should be included in supplementary figures. Even though iPSCs may be procured from another lab, basic iPSC QC within the lab is required to ensure stable cytogenetics. G-band karyotype (cytogenetic stability) and STR identity analysis is required on all the iPSC lines to ensure that the UBR5 observation is not because of an abnormal karyotype of the cell lines.

E. Appropriate use of statistics and treatment of uncertainties

This was one of the major weaknesses in this manuscript. Then n's are mentioned in each figure legend, however, it is not clearly described where the n's are derived from? Are these true biological and experimental replicates or only technical replicates? Is it independent experiments with multiple biological replicates? The statistics should be run on minimum of 3 exclusively performed independent experiments (each experiment containing multiple biological and technical replicates).

Minimum of three biological replicates are required in 3 independent PSC lines in 1 experiment.

Then the assay and phenotype data should be averaged for the group. Then each assay must be repeated a minimum of 3 times (for every iPSC line) in the same experiment. Then the data should be averaged and compared across the groups. Multiple wells in 1 experiment do not qualify as independent experiments.

Given the significant variability in differentiation of iPSC-derivatives from experiments a minimum of 3 experimental replicates are required and data averaged over those experiments.

F. References: appropriate credit to previous work?

References have been appropriately cited except for some of the citations mentioned in this review.

G. Clarity and context: lucidity of abstract/summary, appropriateness of abstract, introduction and conclusions

The paper is apprehensible and well-written including the abstract. Introduction is sound, although a clearer description of how proteostasis changes during somatic reprogramming to iPSCs should be included. The conclusions and their interpretations can be better understood after the

suggested corrections in figures and additional data, inclusion of isogenic mutant correct iPSC controls as well as other additional experiments requested.

Reviewer #3 (Remarks to the Author):

The manuscript by Koyuncu et al. examines the role of UBR5 in HTT and polyQ HTT degradation and aggregation in stem cells (iPSC) and differentiated NPCs.

Main findings of the study:

pluripotent iPSC has increased level of UBR5, which may modulate levels of mutant HTT.

Downregulation of UBR5 in *C. elegans* Huntington's Disease model increases the aggregation of HTT.

Overall, the findings regarding UBR5 are interesting. On one hand, the idea that it plays a role in heightened damage surveillance in stem cells is interesting and somewhat convincing. More experiments would be needed to convincingly demonstrate its quality control role in stem cells. On the other hand, the connection to HTT seems forced, and so far appears tenuous. 1. looking at UBR5 and HTT in complete isolation is biased and is reminiscent of dozens of other studies pitting any number of E3 ligases against HTT, without convincing physiological proof of a direct mechanistic relationship. 2. relatively little effort is made to show that UBR5 in fact ubiquitinates HTT, as opposed to exerting some kind of indirect effect in the KD experiment, whereby the KD significantly increases the misfolded protein load, thereby also affecting HTT.

Major concerns:

Proteasome activity determines the level of HTT and other proteins, inhibition of proteasome would result in protein accumulation that can indirectly trigger HTT aggregation. Or just be explained by increased concentration of HTT, not necessarily the effect of proteasome on HTT. The authors should compare it to autophagy inhibition.

There are many ways to verify a direct relationship between E3 ligase and substrate, from mass spec, to lysine mutation.

It would be interesting to see comparison of mRNA levels of HTT between iPSC and differentiated cells, since UBR5 plays role in the transcriptional repression.

Minor concerns:

Although the authors show that UBR5 KD doesn't prevent differentiations it is important to note that it may still affect differentiation pathways, for example Shh which was also used in this paper to differentiate striatal neurons (<http://journals.plos.org/plosone/article?id=10.1371/journal.pone.0157079>). Did the author find any difference between differentiated striatal neurons with or without UBR5?

It is not clear why the authors used HEK293 model for the ectopic expression, it would be interesting to demonstrate it using differentiated iPSC model.

it would be helpful to see additional, better resolved versions of fig6e

Reviewer #4 (Remarks to the Author):

The manuscript by Koyuncu et. al. titled, "The ubiquitin ligase UBR5 suppresses proteostasis collapse in immortal pluripotent stem cells and Huntington's disease models" asks an important scientific question of significant interest to the field. Namely, the authors ask whether iPSC cells rely on the E3 ubiquitin ligase UBR5 to regulate polyQ-expanded mutant Htt protein levels and aggregation. This is an especially enticing question because stem cells seem to be resistant to proteostasis decline (and polyQ length-dependent Htt aggregation) in a manner that was, until now, not well understood.

The authors found that UBR5 is naturally upregulated in iPSCs and that the knockdown of its expression significantly increased the amount of Htt protein aggregation, both in iPSCs and in *C. elegans* models of polyQ aggregation/toxicity. Ultimately, they nicely demonstrate that UBR5 is a "central modulator" of mHtt aggregation.

The authors' arguments would be even more convincing if they would address the following:

1. In figure 1e and f, the authors show a western blot using antibodies against Htt. They find that mutant Htt is expressed at lower levels than normal huntingtin. It would be helpful if the authors explained how they are certain that their antibody is equally immunoreactive to mutant Htt and to normal Htt protein.
2. In figure 1d, the authors show a filter trap assay to represent the amount of Htt aggregates present in different iPS cells lines in the presence or absence of the proteasome inhibitor MG132. It would be helpful if the authors provided an explanation for the relatively high number of Q33 aggregates observed in untreated cells.
3. In Figure 5C-E, the authors demonstrate the effect of UBR5 knockdown on *C. elegans* models of polyQ aggregation/toxicity. Specifically, they demonstrate that knockdown results in an increase in polyQ67 protein levels, aggregation, and toxicity.
 - a) The model that they used expresses a polyQ peptide fused to YFP for visualization. The authors should clarify that outside of the polyQ region, this model expresses no Htt protein. They should also discuss the implications of this. Do they think that UBR5 acts on polyQ alone, irrespective of the protein context in which the polyQ is found? Would this mean that it acts on all polyQ proteins, including other disease-associated proteins?
 - b) Because these animals express polyQ67 fused to YFP, the authors could easily show aggregation in fixed (or live) animals in addition to doing filter trap assays. In the original study describing this *C. elegans* model for aggregation/toxicity, differences in aggregation propensity were reported for different neuronal subtypes. By examining whole animals, the authors could demonstrate whether UBR5 depletion has a greater effect in certain neurons or if the effect on polyQ67 aggregation is universal.
 - c) The authors used thrashing assays as a measure of toxicity. Their n-number was relatively low (18-20) and I suggest increasing it. Nonetheless, the statistics indicate a significant increase in Q67 toxicity when UBR5 gene expression is knocked down. I am somewhat surprised that the effect was as strong as it was, given that *C. elegans* neurons are famously resistant to RNAi. In fact, mutant strains that are hypersensitive to RNAi are typically used in studies requiring RNAi-mediated knockdown of neuronal gene expression. If the authors introduced an *rrf-3* mutation, for example, they might see a stronger and more reliable effect of UBR5 RNAi in neurons. Perhaps they would then see more of an effect on Q19 as well.
 - d) Several RNAi screens have been conducted in *C. elegans* to identify genes whose knockdown causes a measurable increase or decrease in polyQ protein aggregation. The authors should mention whether UBR5 was previously uncovered as a proteostasis regulator in any of those

screens.

Overall, I feel that this is an important study that elevates UBR5 to a prominent position within the proteostasis network. In fact, at least in stem cells, UBR5 seems to play a central role in downregulating Htt protein levels and thus preventing (or delaying) Htt aggregation in differentiated cells. After the authors address the above comments, I would strongly recommend the manuscript for publication.

Reviewer #1:

iPSCs derived from controls and HD patients have been used widely as in vitro cell models to facilitate discovery of novel targets and treatments for human HD without the ethical limitations. In this manuscript, the authors used HD-iPSCs lines that carry different CAG repeat lengths (71 and 180 CAGs) in the Huntington disease gene for study and found that the increased proteasome activity of iPSCs can regulate HTT levels and polyQ expanded HTT aggregation in these cells. More importantly, they found the intrinsic high levels of the E3 ubiquitin ligase UBR5 are a key component to maintain proteostasis of HTT in HD-iPSCs thus provide some interesting evidence to explain why HD-iPSCs lack the aggregate phenotype characteristic of HD. Although the findings in the manuscript might be highly interesting to the field, the paper seems to suffer from some significant problems as listed below.

Major issues:

1. It is expected that proteostasis in iPSCs is different from that in differentiated neuronal cells. As shown by the paper, a number of proteins related to proteostasis are reduced when iPSCs are differentiated. It is also well-known that proteasome activity determines the level of mutant HTT. The more interesting issue is how specific is UBR5's regulation on HTT accumulation. The comparison between UBR5 and other related proteins on HTT accumulation is very limited. Also, whether UBR5 influences other protein accumulation remains unknown. Without the above information, the impact of this paper is quite minimal.

The authors claimed that UBR5 is a central modulator of mutant HTT aggregation in iPSCs. Does UBR5 also regulate the accumulation of other proteins, especially other misfolded proteins? Western blot results actually show that inhibiting UBR5 also increases actin expression on some blots. UBR5 and other proteins may increase intrinsic higher UPS activities in iPSCs, but how critical UBR5 is for this higher activity remains unknown.

We agree with Reviewer #1 that all of these important points need to be addressed.

- *We have now compared the requirements of UBR5 for the regulation of HTT levels and aggregation in iPSCs with other related proteins on HTT accumulation, including distinct components of the UPS network (i.e., UBE3A, UBE2K) as well as chaperones (i.e., TRiC/CCT, HYPK). Please see **Fig. 6a, b** and **Supplementary Fig. 20**. These results support an important role of UBR5 on monitoring HTT levels, a process that could facilitate mutant HTT regulation by other proteostasis nodes such as enhanced TRiC/CCT complex in HD-iPSCs.*
- *We have now examined whether UBR5 regulates other aggregation-prone proteins. In particular, we have performed the following experiments:*
 - a) *Analysis of ATXN3 levels and aggregation upon proteasome inhibition and UBR5 knockdown in iPSCs derived from two distinct patients diagnosed with Machado-Joseph disease (MJD). Please see **Fig. 6c-g** and **Supplementary Fig. 21**. Although loss of UBR5 increased HTT levels in MJD-iPSCs, we did not find changes in either normal or mutant ATXN3. Moreover, UBR5 downregulation did not induce aggregation of polyQ-expanded ATXN3. This phenotype differed from HD-iPSCs lines, where*

up-regulation of mHTT levels upon UBR5 knockdown was sufficient to trigger polyQ-expanded HTT aggregation. Thus, our data indicate that not all polyQ-containing proteins are modulated by UBR5.

- b) We have now examined whether UBR5 regulates protein levels or aggregation of wild-type FUS as well as two distinct mutant FUS variants linked with amyotrophic lateral sclerosis (ALS). Remarkably, loss of UBR5 did not stimulate RNA-stress granules formation or FUS intracellular delocalization in control or ALS-iPSC lines (**Fig. 7a**). In addition, UBR5 knockdown did not impair wild-type or mutant FUS levels whereas the levels of HTT were up-regulated in these cells (**Fig. 7b**). Thus, our results support a specific role of UBR5 on HTT regulation. Although UBR5 was dispensable for ATXN3 and FUS proteostasis, we have indicated that we cannot discard a role of UBR5 in the control of other aggregation-prone proteins associated with disease.
- c) Besides its role in HTT modulation, we asked whether UBR5 also determines the global proteostasis ability of human pluripotent cells. The text now says: "Under normal conditions, misfolded proteins are refolded by chaperones or terminated via proteolytic systems¹³. Metabolic and environmental conditions (e.g., heat stress) challenge the structure of proteins, increasing the load of misfolded and damaged proteins. When proteolytic systems are overwhelmed, misfolded proteins accumulate into aggresomes. In these lines, we observed that heat stress induces the accumulation of aggresomes in pluripotent stem cells despite their increased proteolytic ability (**Fig. 7c**). Notably, loss of UBR5 was sufficient to induce the accumulation of aggresomes in these cells (**Fig. 7c**). Thus, our data suggest that UBR5 not only regulates the levels of specific proteins such as HTT, but it is also involved in the degradation of misfolded proteins ensued from normal metabolism".
- d) Since UBR5 overexpression was sufficient to ameliorate aggregation of polyQ100-HTT in human cell models, we have now examined whether ectopic expression of this E3 can reduce aggregation of other misfolded proteins. For this purpose, we generated a human cell model that expresses an aggregation-prone β -amyloid protein (i.e., beta23 construct provided by F.U Hartl (Olzscha et al, Cell, 2011)). We found that ectopic expression of UBR5 does not reduce aggregation of these β -amyloid fibrils.
- As Reviewer #1 indicates, actin expression was increased in samples with UBR5 knockdown on some blots presented in our first submission. However, this is not consistent among the multiple independent experiments performed for the different lines. In fact, we did not observe actin differences upon UBR5 knockdown in most of the experiments performed. For instance, please see the experiments now presented in **Fig. 4b-e, 6a, 6-e-f, 7b** and **Supplementary Fig. 13a-e**. Moreover, we have now quantified HTT, ATXN3 and FUS levels by densitometry and normalize to β -actin loading. These quantifications showed a significant up-regulation of HTT levels upon UBR5 knockdown, whereas ATXN3 and FUS levels remained similar.
 - As mentioned by Reviewer #1, analysis of the role of UBR5 in the intrinsic high proteasome activity of iPSCs is of high importance for our conclusions. We have now assessed whether UBR5 regulates chymotrypsin-like, caspase-like and trypsin-like proteasome activities in iPSC lines. Our results indicate that loss of UBR5 does not

affect these activities in control and HD-iPSC lines (Please see **Fig. 4f and Supplementary Fig. 14**).

2. It is known that the impairment of clearance of misfolded or mutant proteins is key to their accumulation. There is no evidence that UBR5 selectively regulates the level of mutant HTT, and western blotting in Fig. 1e did not show the selective increase of expanded Htt, as the upper band (mutant HTT) had the same changes as the lower band (normal HTT). Fig 2h does not show any specific precipitation of mutant HTT with UBR-5. This raises a significant issue of how UBR5 regulates both normal and mutant HTT. Without knowing this, the conclusion that that UBR5 is a central modulator of mutant HTT aggregation in iPSCs is weak.

*Reviewer #1 is absolutely right. The results presented in our first submission indicated that the UPS determines the levels of both wild-type and mutant HTT in iPSCs. Likewise, our data suggested that UBR5 is key for the proteostatic regulation of normal and aggregation-prone HTT. Although we discussed the role of UBR5 in the modulation of both normal and mutant HTT, we apologize if we did not make clear this point in our first submission. We have now presented several experiments that strengthen our conclusions. For instance, we have now included western blots of three independent control iPSCs (please see **Fig. 2b and Supplementary Fig. 5a, b**), indicating a strong increase in wild-type HTT upon proteasome inhibition. Likewise, we have now included experiments in an additional control iPSC line as well as two independent human embryonic stem cell lines (hESCs), showing an increase in wild-type HTT upon UBR5 knockdown (**Supplementary Fig. 13a-c**). Moreover, our experiments in HD-iPSCs lines (including new analysis of additional independent HD-iPSC lines) support a strong increase of both wild-type and mutant HTT upon proteasome inhibition and UBR5 knockdown (i.e., **Fig. 4 and Supplementary Fig. 5, 13**). In addition, we have now presented quantifications of all these western blot experiments that clearly show a similar up-regulation of normal HTT when compared with mutant HTT in HD-iPSCs (e.g., **Fig. 2c, d and Fig. 4d, e**). Finally, we have performed co-immunoprecipitation experiments with anti-UBR5 antibody in two control iPSCs and one additional HD-iPSC line (**Supplementary Fig. 16**). These experiments further validated the interaction of UBR5 with both normal and mutant HTT.*

*As Reviewer #1 indicates, it is well-known that the clearance of misfolded or mutant proteins is key to their aggregation. Since UBR5 not only monitors the levels of normal HTT but also mutant HTT, we hypothesize that this process facilitates regulation of polyQ-expanded HTT aggregation by other enhanced proteostasis nodes such as enhanced TRiC/CCT complex. Besides normal HTT, loss of UBR5 in HD-iPSCs also increases mutant HTT expression, a process that could overwhelm the proteostasis regulation of this mutant variant. In these lines, we observed that downregulation of UBR5 and concomitant dysregulation of HTT levels results in polyQ-expanded HTT aggregation in HD-iPSCs, as they express mutant HTT (e.g. **Fig. 5 and Supplementary Fig. 17-18**). Thus, we believe that our results support a key role of UBR5 in the regulation of polyQ-expanded HTT aggregation. This is further supported by UBR5 overexpression experiments, which strengthen its potential role to reduce mutant HTT aggregation (**Fig. 10**). We have now made clear this in the abstract as well as further discussed it in the Results and Discussion sections.*

Other issues:

1. It is interesting to see that loss of UBR5 hastened the aggregation phenotypes in HD neurons differentiated from HD-iPSCs. In Fig5b and Sup Fig 5, it seemed that Q71 neurons formed more aggregates than Q180 neurons, quantitative data should be added to make it more convincing and explain this phenomenon. Do the increased aggregates in HD neurons have toxicity and affect the cell line viability when knocking down UBR5? Does the polyQ length (71 Q and 180 Q) have relation with increased aggregates in HD neurons? More evidence should be provided in HD neurons since neuronal cells are more vulnerable than other cells in HD.

In our first submission, we did not include extensive analysis on striatal neurons from HD-iPSCs because our main focus was to study proteostasis of pluripotent stem cells. However, Reviewer #1 raises a very important point: Since neurons derived from HD-iPSCs lack HD phenotypes (i.e., neurodegeneration and accumulation of polyQ inclusions), it is interesting to examine in detail the relevance of disturbing proteostasis in these cells.

*In our first version of the manuscript, we only presented experiments in one HD-Q71 line as well as HD-180 cells upon striatal neuronal differentiation. As indicated by Reviewer #1, the conclusions from these experiments were weak with the presented data (i.e., only two lines analysed, lack of neuronal markers to confirm neuronal identity). Although we were very excited about the data presented in our first submission, thanks to the Reviewer's comments we have now obtained strong evidence that indicates that striatal neurons derived from HD-iPSCs do not accumulate polyQ-expanded aggregates upon UBR5 knockdown. We have now performed new neuronal differentiation experiments for control iPSCs #1 and #2 as well as Q57, Q71 (lines #1 and #2) and Q180 iPSCs. Upon UBR5 knockdown, we were not able to find polyQ-aggregates in cells which were positive for either pan-neuronal (i.e., MAP2) or striatal neuronal (i.e., DARPP32) markers in the multiple experiments analysed (please see **Supplementary Fig. 32**). These results are in line with previous studies that reported lack of HTT aggregates even after the addition of proteasome and autophagy inhibitors or the induction of oxidative stress in neurons (from our laboratory and other laboratories: Noormohammadi et al, Nat. Comm., 2016; HD-iPSC Consortium, Cell Stem Cell, 2012; Jeon et al, Stem Cells, 2012; Victor et al, Nat. Neuroscience, 2018). The lack of polyQ-expanded aggregates in these cells could reflect the long period of time before aggregates accumulate in HD. We have confirmed that proteasome inhibition does not induce polyQ-expanded aggregation in our differentiated neuronal models from iPSCs (**Supplementary Fig. 31a, b**). More importantly, we found that proteasome inhibition does not increase HTT protein levels in these cells (**Supplementary Fig. 31c, d**). In these lines, we have observed that loss of UBR5 does not induce an increase in HTT levels in neurons (**Supplementary Fig. 33**). Moreover, we found that UBR5 knockdown does not trigger neuronal death in control or HD-neurons derived from iPSCs (**Supplementary Fig. 34**). The text now says: "Although HD-iPSCs can terminally differentiate into MSNs, these cells do not exhibit mutant HTT aggregates even after the addition of proteasome and autophagy inhibitors or the induction of oxidative stress^{16,31,33,66}. Thus, these findings support a rejuvenation process during cell reprogramming that prevents aberrant aggregation in differentiated neurons. In addition, the lack of polyQ-expanded aggregates in these cells could reflect the long period of time before aggregates accumulate in HD³¹. In these lines, HD-MSNs derived from iPSCs do not accumulate detectable polyQ-expanded inclusions at 12 weeks after transplantation into HD rat models. However, they accumulate aggregates after 33 weeks of transplantation³³. As previously*

reported, proteasome inhibition did not induce polyQ-HTT aggregation in HD-MSNs differentiated from iPSCs (**Supplementary Fig. 31a, b**). Moreover, we did not observe a significant increase in HTT levels upon proteasome inhibition (**Supplementary Fig. 31c, d**). Likewise, knockdown of UBR5 in differentiated HD-MSNs was not sufficient to induce mutant HTT aggregation and up-regulated HTT levels (**Supplementary Fig. 32, 33**). In addition, UBR5 knockdown did not reduce cell viability of HD-MSNs (**Supplementary Fig. 34**)”.

2. In Fig 5a and b, the HD neurons used are induced striatal neurons, but there is lack the DARPP32 staining or other markers that can prove the successful induction of HD iPSCs to striatal neurons. Moreover, the authors found that UBR5 knockdown did not influence the iPSCs differentiation. Lack of the convincing figures for neuronal induction weakens the conclusion of this manuscript.

As mentioned above, we apologize for not including these controls in our first submission and we thank the Reviewer for raising this issue. As we have explained in detail above, knockdown of UBR5 in differentiated neurons from naïve control and HD-iPSCs do not trigger polyQ-aggregation in MAP2 or DARPP32-positive cells.

*Regarding the induction of HD-iPSCs to neurons, we have now included quantification of striatal neuronal differentiation efficiency in the different lines (please see **Supplementary 29a, b**). We observed that among those cells expressing the neuronal marker MAP2, ~30-60% also expressed GABA depending on the cell lines (similar to Ma et al, Cell Stem Cell, 2012). We have now examined whether loss of UBR5 at the iPSC level affects neuronal induction of these cells (both control and HD iPSCs) by assessing percentage of GABA-positive cells and the expression levels of distinct striatal neuronal markers. We did not observe differences in their ability to differentiate into striatal neurons (**Supplementary Fig. 29a-g**). The text now says: “One step further was to determine whether these cells are able to generate terminally differentiated neurons. Since GABAergic medium spiny neurons (MSNs) undergo the greatest neurodegeneration in HD⁶⁴, we differentiated iPSCs into striatal neurons⁶⁵. Among those cells expressing the neuronal marker MAP2, ~30-60% also expressed GABA depending on the cell line (**Supplementary Fig. 29a, b**). As previously reported³¹, these differences were not associated with the expression of mutant HTT (**Supplementary Fig. 29a, b**). Knockdown of UBR5 at the iPSC stage did not reduce their ability to differentiated into striatal neurons (**Supplementary Fig. 29a-g**). In contrast to NPCs, terminally differentiated neurons derived from HD-iPSCs with downregulated UBR5 levels did not accumulate polyQ-expanded aggregates (**Supplementary Fig. 30**). Thus, the lack of defects in neurogenesis and aggregates in these cells indicate that other mechanisms activate during neuronal differentiation to facilitate proteostasis of mutant HTT”.*

3. Please provide more detail information for the HD iPSCs cell lines used in this manuscript, including the HD patient donors, onset ages, and symptoms.

We have now added more detail information for the patient-derived iPSCs used in our manuscript:

- 1) *In the main text, we have indicated onset ages. For instance, the text now says: “we assessed whether this activity is required to prevent mutant HTT aggregation in iPSCs derived from two individuals with juvenile onset HD (i.e., Q71 and Q180)³¹*

(Supplementary Table 1)". (...) "proteasome inhibition also induced a high percentage of mutant HTT aggregation in iPSCs derived from an individual with adult onset HD (Q57)". "Likewise, FUS was predominantly detected in the nucleus of iPSCs ($FUS^{R521C/wt}$) derived from a patient affected by ALS in mid-late age^{56,57} (**Fig. 7a**). In iPSCs expressing a FUS variant linked with severe and juvenile ALS ($FUS^{P525L/P525L}$), the protein was mostly located in the cytoplasm⁵⁶ (**Fig. 7a**)".

- 2) We have now added a Table with information of age at sampling, age of onset, gender and symptoms for all the disease-iPSCs lines used in our study (please see **Supplementary Table 1**).
- 3) We have now included detail information in the Methods section regarding the fibroblast lines, methods and transcription factors used for reprogramming in the original publications.

4. The authors described in Fig (1e,1f) that the differences between the protein levels of normal and mHTT were more pronounced in HD-iPSCs that express longer polyQ repeats(Q180). It would be more convincingly shown if both Q71 and Q180 samples were on the same blot to compare the difference. It seemed that there are no significant changes with or without MG132 for the mHTT bands using HTT antibody in Q180 cell lines of Fig 1f.

*This is an excellent suggestion which made more make clear our results. We have now presented a western blot with the different control and iPSC lines to compare differences in the signal observed with anti-HTT and anti-expanded polyQ antibodies (please see **Fig. 2a**). In these lines, Reviewer #4 raised an important point (please see below): the differences detected with anti-HTT antibody could be due to a decrease immunoreactivity to mutant HTT when compared to normal HTT. We have now performed extensive validation of the antibodies that clearly shows that anti-HTT antibody is less immunoreactive to mutant HTT (**Supplementary Fig. 4**). On the other hand, the antibody to polyQ-expanded proteins only recognizes mutant HTT on western blots and the intensity of the signal correlates with the length of the polyQ expansion (**Fig. 2a** and **Supplementary Fig. 4**). The text now says: "For this purpose, we characterized in our model two antibodies that recognize HTT and polyQ-expanded proteins, respectively^{16,38}. First, we validated that these antibodies detect endogenous levels of HTT in iPSCs (**Supplementary Fig. 4a-c**). The HD-iPSCs used in this study express one mutant allele of HTT but also one normal copy (**Supplementary Table 1**)^{16,31,39-41}. Since the length of the polyQ stretch diminishes the electrophoretic mobility of proteins³⁸, we could discriminate normal HTT and mutant HTT in both HD Q71 and Q180-iPSC lines by western blot using anti-HTT antibody (**Fig. 2a** and **Supplementary Fig. 4a**). In HD Q57-iPSCs, the differences in the electrophoretic motilities of both alleles were marginal and they were not efficiently separated on western blot assays (**Fig. 2a**). We also observed that anti HTT-antibody was less immunoreactive to mutant HTT (**Fig. 2a** and **Supplementary Fig. 4d**). These differences were more pronounced in HD-iPSCs that express longer polyQ repeats (Q180), as mutant HTT was only detected after high exposure times in these cells (**Fig. 2a**). Thus, we used the antibody to polyQ-expanded proteins to examine the expression of mutant HTT in these cells (**Fig. 2a**). We confirmed that this antibody only recognizes mutant HTT on western blots and the intensity of the signal correlates with the length of the polyQ expansion (**Fig. 2a** and **Supplementary Fig. 4d**), as previously reported³⁸. Although Q57 expansion was not detected with anti-polyQ-expanded proteins by western blot, this antibody strongly detected mutant HTT in Q71 and Q180 lines (**Fig. 2a**)".*

Since anti-HTT is less immunoreactive to mutant HTT and these differences are very pronounced in HD-iPSCs that express longer polyQ repeats (Q180), mutant HTT was only detected after high exposure times in these cells. In fact, this signal was barely detectable in some replicate experiments. In **Fig. 2a**, we have included a higher exposure time to show mutant HTT detected by anti-HTT antibody. Regarding former Fig. 1f, we have now presented a new replicate experiment (please see **Fig. 2d**) and a higher exposure of the entire western blot (**Supplementary Fig. 6**). With this exposure, the differences in mutant HTT detected by anti-HTT mutant were more visibly upon proteasome inhibition treatment (**Supplementary Fig. 6**). Nevertheless, due to the weak signal (or even undetectable in some replicates) in HD Q180-iPSCs, we based our conclusions for this line on the changes observed in mutant HTT detected (and quantified) with anti-expanded polyQ antibody.

5. In Fig1a, the nuclei Hoechst staining in control iPSCs seemed much less than HD iPSCs and the magnifications are different between the two groups. Please make the figures clearer and use the same magnification.

We apologize for the low magnification of the immunocytochemistry experiments presented in our first submission. We have now repeated all the experiments and presented higher magnifications. In the Fig. 1a presented in our first submission, we used the same magnification for all the images. The differences in the nuclei size may be due to imaging of areas with different cell confluency. We have now taken images of areas with similar confluency (e.g., **Fig. 1a**).

6. The entire blot of HTT needs to be presented to see how degraded HTT products are changed by UBR5.

We have now presented uncropped blots for all the experiments in **Supplementary Fig. 39**. We could not fit the entire blots in the main figures due to the limited space.

7. Fig 2h does not show any specific precipitation of mutant HTT with UBR-5. Since the authors can detect both normal and mutant HTT on the same western blot and their sizes are distinguishable, the authors need to compare immunoprecipitation of both normal and mutant HTT with UBR5 on the same western blot.

Indeed, the indicated figure (now presented in **Fig. 4g**) does not show specific precipitation of mutant HTT with UBR5. Anti-HTT antibody shows co-immunoprecipitation of both mutant and normal HTT with UBR5. The interaction with mutant HTT was confirmed by using anti-polyQ-expanded protein antibody. As mentioned above, we have now performed co-immunoprecipitation experiments with anti-UBR5 antibody in two control iPSCs and one additional HD-iPSC line (**Supplementary Fig. 16**). These experiments further validated the interaction of UBR5 with both normal and mutant HTT. We have now made more clear this point in the text: "Moreover, UBR5 interacted with both normal and polyQ-expanded HTT in HD-iPSCs whereas we were not able to detect this interaction with a distinct up-regulated E3 enzyme (**Fig. 4g and Supplementary Fig. 16**)".

8. Figure 3, UBR5 shRNA treatment did not show clearly the increase in HTT aggregates that are often localized in the nuclei. It seems that soluble cytoplasmic HTT, but not nuclear HTT aggregation, was increased in the presented images.

*As mentioned above, we have now performed new replicate experiments and presented higher magnification images. For instance, please see **Fig. 1a and 5a**. These images now clearly show cytoplasmic accumulation of polyQ-HTT aggregates in the distinct HD-iPSCs upon proteasome inhibition or UBR5 knockdown. Filter trap experiments confirm aggregation of mutant HTT under these conditions. Reviewer #1 is right and the aggregates are mostly cytoplasmic (as we also validated by confocal imaging). We have now made clear this phenotype in the main text.*

9. MG132 treatment for 12 h may reduce cell viability to increase HTT accumulation and aggregation. The authors need to rule out this possibility.

*We agree with Reviewer #1 and we have now assessed this possibility using different concentrations of proteasome inhibitor as well as different time points (please see **Fig. 1 d, e**). Our results indicate aggregation of mutant HTT upon proteasome inhibition in the absence of cell death. Thus, this phenotype cannot only be explained by impairment of cell viability. The text now says: "To further examine the link between proteasome activity and polyQ-expanded aggregation in iPSCs, we tested different concentrations of proteasome inhibitor. Lower concentrations reduced approximately 50% of the proteasome activity, which was sufficient to induce mutant HTT aggregation as assessed by both filter trap and immunofluorescence experiments (**Fig. 1b-d and Supplementary Fig. 3**). Since these analyses were performed in iPSCs treated with proteasome inhibitor for 12 h, we examined whether this treatment reduces cell viability, a process that could trigger proteostasis collapse and, in turn, dysregulation of protein aggregation. Whereas different concentrations of proteasome inhibitor resulted in polyQ-expanded aggregation (**Fig. 1c, d**), only higher concentrations induced a mild increase (~8%) in cell death (**Fig. 1e**). In addition, we found accumulation of mutant HTT aggregates at an earlier time point (4 h) of the proteasome inhibition treatment when higher concentrations did not trigger cell death (**Fig. 1d, e**). Thus, these results indicate that a decline in proteasome activity promotes mutant HTT aggregation in HD-iPSCs, a process that cannot only be explained by impairment of cell viability. Remarkably, control and HD-iPSCs exhibited similar sensitivity to proteasome inhibition (**Fig. 1e**), suggesting that mutant HTT aggregation does not induce cell death in iPSCs".*

Reviewer #2

The ubiquitin ligase UBR5 suppresses proteostasis collapse in immortal pluripotent stem cells and Huntington's disease models. Seda Koyuncu, Isabel Saez, Ricardo Gutierrez-Garcia, Wojciech Pokrzywa, Azra Fatima, Thorsten Hoppe, and David Vilchez

A. Summary of the key results.

In this paper the authors have for the first time shown that UBR5 ubiquitin ligase suppresses a breakdown of protein homeostasis due to upregulation of UBR5 in iPS cells. They also show

some evidence of how these enhanced proteostatic capabilities of PSCs may be useful creating better iPSC models of Huntington's Disease poly-Q repeat expansion disorder. They then complement their human PSC data using *C. elegans* model of HD. The advances described in this study are worthy of publication in Nature Communications after some additional data and revisions suggested below.

B. Originality and significance: if not novel, please include reference

This study is of great significance and novelty is medium. The same group has described similar the importance of similar mechanisms in Nature Communications 2016 article "Somatic increase of CCT8 mimics proteostasis of human pluripotent stem cells and extends *C. elegans* lifespan".

There are other studies that describe a similar important role of UBR5 in pluripotent stem cells.

Buckley et al. Cell Stem Cell 2012. Regulation of Pluripotency and Cellular Reprogramming by the Ubiquitin-Proteasome System.

In this manuscript, the authors performed a systematic study on how components of the UPS regulate mouse ESC pluripotency and cell reprogramming. This relevant work has important implications for the understanding of pluripotency and reprogramming. As such, we have cited the manuscript several times. Moreover, we have now discussed in more detail their findings on UBR5 and pluripotency, in comparison with our work and other studies. The discussion section now says: "Whereas our results highlight the importance of UBR5 in proteostasis of iPSCs, the intrinsic high levels of this enzyme could also suggest a role in maintenance of pluripotency via the ubiquitination of endogenous substrates. Interestingly, a study has reported that RNAi-mediated knockdown of Ubr5 results in a significant loss of pluripotency markers in control mouse ESCs²⁰. However, a different study did not find impairment of pluripotency markers in murine ESCs upon UBR5 RNAi-treatment⁷⁴. Likewise, we did not find changes in the expression of pluripotency or germ layer markers in control hESCs/iPSCs upon loss of UBR5. Interestingly, knockdown of Ubr5 reduces the induction of Sonic hedgehog (Shh) in murine pluripotent stem cells upon retinoic-acid treatment⁷⁴. In these lines, a conditional Ubr5 mutant mouse presents decreased hedgehog signaling around embryonic day 13.5⁷⁴. Although this model exhibits shorter limbs when compared to controls, the differences were not significant and the mutants do not have other obvious morphological defects⁷⁴. Since our striatal neuronal differentiation protocol is based on the induction of hedgehog signaling pathway, we tested whether UBR5 is necessary for the generation of MSNs from human iPSCs. However, loss of UBR5 did not affect their differentiation into MSNs, indicating that these cells conserved their ability to induce the hedgehog signaling. The distinct impact of UBR5 in pluripotency and differentiation could be associated to genetic differences between mouse and human species. Moreover, it is important to note the distinct pluripotent states exhibited by murine ESCs and hESCs/iPSCs⁵. Mouse ESCs are cultured in the presence of serum and leukaemia inhibitory factor (LIF), and exhibit a naive state resembling the pluripotent state observed in the inner cell mass of the pre-implantation embryo⁵. On the other hand, hESCs as well as human iPSCs obtained by direct in vitro reprogramming do not require LIF signaling and exhibit a more primed state that resembles post-implantation embryonic configurations⁵".

Sanchez et al. PNAS 2016. BMI1–UBR5 axis regulates transcriptional repression at damaged chromatin.

In this study, the authors found UBR5 as a downstream factor of BMI1, a component of Polycomb Repressive Complex (PRC1). By using HeLa, 293T, and U2OS cell models, they discovered that BMI1 and UBR5 repress polymerase II-mediated transcription at damaged chromatin. It is important to note that the authors did not assess the role of UBR5 in stem cells or its impact in development. However, since PRC1 maintains epigenetic silencing during development, UBR5 may also be involved in this process. We have now cited this manuscript. The text now says: “Since UBR5 promotes microRNA-mediated transcript destabilization in mouse ESCs⁴⁵ and may be involved in transcriptional repression during development⁴⁶, we examined whether UBR5 downregulation during differentiation correlates with increased HTT mRNA levels”.

C. Data & methodology: validity of approach, quality of data, quality of presentation
The overall quality and style of presenting the data is and approach taken is valid. Experiments have been carried out with precision and appropriate set of controls is included. The weakness in this study are highlighted below. At least all experiments should be conducted with the allelic series of lines in this study – Q21, Q33, Q71, Q180 to reach statistically sound conclusions. The authors could also bolster their conclusions by using multiple clonal HD iPSC lines is from the CAG repeat donor. This is a norm for iPSC disease modeling field now.

Reviewer #2 is absolutely right, and all the experiments should be conducted at least with Q21, Q33, Q71 and Q180 to reach solid conclusions. We have now examined all these lines in each analysis. Moreover, the analysis include a second HD Q71 iPSC line generated by an independent laboratory (as explained in detail in the methods section). For the most important experiments, we have now added a third independent control iPSC line as well as HD Q57-iPSCs. Moreover, we have now examined how UBR5 affects HTT levels in two hESC lines as well as iPSCs from Machado-Joseph disease (MJD) and amyotrophic lateral sclerosis (ALS) patients.

Most importantly, this paper has not demonstrated evaluation of these molecular mechanisms in relevant terminally differentiated striatal neurons that degenerate in human HD patients. There is lack of characterization of the striatal markers in the neuronal differentiation experiments. Therefore, it is important to demonstrate relevance of disturbing the proteostasis and poly Q aggregates in striatal neurons.

*We have now included quantification of striatal neuronal differentiation efficiency in the different lines (please see **Supplementary 29a, b**). In our first submission, we did not include extensive analysis on striatal neurons from HD-iPSCs because our main focus was to study proteostasis of pluripotent stem cells. However, both Reviewer #1 and #2 raise a very important point: Since neurons derived from HD-iPSCs lack HD phenotypes (i.e., neurodegeneration and accumulation of polyQ inclusions), it is interesting to examine in detail the relevance of disturbing proteostasis in these cells.*

In our first version of the manuscript, we presented experiments in one HD-Q71 line as well as HD-180 cells upon striatal neuronal differentiation. We completely agree with Reviewer #2 that the conclusions from these experiments were weak with the presented data (i.e., only two lines analysed, lack of neuronal markers to confirm neuronal identity). Although we were

very excited about the data presented in our first submission, thanks to the Reviewers' comments we have now obtained strong evidence that indicates that striatal neurons derived from HD-iPSCs do not accumulate polyQ-expanded aggregates upon UBR5 knockdown. We have now performed new neuronal differentiation experiments for control iPSCs #1 and #2 as well as Q57, Q71 (lines #1 and #2) and Q180 iPSCs. Upon UBR5 knockdown, we were not able to find polyQ-aggregates in cells which were positive for either pan-neuronal (i.e., MAP2) or striatal neuronal (i.e., DARPP32) markers in the multiple experiments analysed (please see **Supplementary Fig. 32**). These results are in line with previous studies that reported lack of HTT aggregates even after the addition of proteasome and autophagy inhibitors or the induction of oxidative stress in neurons (from our laboratory and other laboratories: Noormohammadi et al, Nat. Comm., 2016; HD-iPSC Consortium, Cell Stem Cell, 2012; Jeon et al, Stem Cells, 2012; Victor et al, Nat. Neuroscience, 2018). The lack of polyQ-expanded aggregates in these cells could reflect the long period of time before aggregates accumulate in HD. We have confirmed that proteasome inhibition does not induce polyQ-expanded aggregation in our differentiated neuronal models from iPSCs (**Supplementary Fig. 31a, b**). More importantly, we found that proteasome inhibition does not increase HTT protein levels in these cells (**Supplementary Fig. 31c, d**). In these lines, we have observed that loss of UBR5 does not induce an increase in HTT levels in neurons (**Supplementary Fig. 33**). Moreover, we found that UBR5 knockdown does not trigger neuronal death in control or HD-neurons derived from iPSCs (**Supplementary Fig. 34**). The text now says: "Although HD-iPSCs can terminally differentiate into MSNs, these cells do not exhibit mutant HTT aggregates even after the addition of proteasome and autophagy inhibitors or the induction of oxidative stress^{16,31,33,66}. Thus, these findings support a rejuvenation process during cell reprogramming that prevents aberrant aggregation in differentiated neurons. In addition, the lack of polyQ-expanded aggregates in these cells could reflect the long period of time before aggregates accumulate in HD³¹. In these lines, HD-MSNs derived from iPSCs do not accumulate detectable polyQ-expanded inclusions at 12 weeks after transplantation into HD rat models. However, they accumulate aggregates after 33 weeks of transplantation³³. As previously reported, proteasome inhibition did not induce polyQ-HTT aggregation in HD-MSNs differentiated from iPSCs (**Supplementary Fig. 31a, b**). Moreover, we did not observe a significant increase in HTT levels upon proteasome inhibition (**Supplementary Fig. 31c, d**). Likewise, knockdown of UBR5 in differentiated HD-MSNs was not sufficient to induce mutant HTT aggregation and up-regulated HTT levels (**Supplementary Fig. 32, 33**). In addition, UBR5 knockdown did not reduce cell viability of HD-MSNs (**Supplementary Fig. 34**)".

Moreover, we have now examined whether loss of UBR5 at the iPSC level affects neuronal induction of these cells (both control and HD iPSCs) by assessing percentage of GABA-positive cells and the expression levels of distinct striatal neuronal markers. We did not observe differences in their ability to differentiate into striatal neurons (**Supplementary Fig. 29a-g**). The text now says: "One step further was to determine whether these cells are able to generate terminally differentiated neurons. Since GABAergic medium spiny neurons (MSNs) undergo the greatest neurodegeneration in HD⁶⁴, we differentiated iPSCs into striatal neurons⁶⁵. Among those cells expressing the neuronal marker MAP2, ~30-60% also expressed GABA depending on the cell line (**Supplementary Fig. 29a, b**). As previously reported³¹, these differences were not associated with the expression of mutant HTT (**Supplementary Fig. 29a, b**). Knockdown of UBR5 at the iPSC stage did not reduce their ability to differentiate into striatal neurons (**Supplementary Fig. 29a-g**). In contrast to NPCs, terminally differentiated neurons derived from HD-iPSCs with downregulated UBR5 levels did not accumulate polyQ-expanded aggregates (**Supplementary Fig. 30**). Thus, the lack of defects in neurogenesis and aggregates in these cells indicate that other

mechanisms activate during neuronal differentiation to facilitate proteostasis of mutant HTT”.

There is also a lack of in-lab HD-iPSC pluripotency validation data described in supplementary figures. Even though iPSCs may be procured from another lab, basic iPSC QC within the lab is required to ensure stable cytogenetics. G-band karyotype (cytogenetic stability) and STR identity analysis is required on all the PSCs used in this study including the hESC, as well as the HD-iPSCs.

*We have now confirmed that all the hESC and iPSC lines used in our experiments had a normal diploid karyotype as assessed by single nucleotide polymorphism (SNP) genotyping (please see **Supplementary Fig. 38**). We have indicated this quality control in the Methods section and described how we performed the SNP analysis.*

*We have now performed STR analysis to confirm the identity of the hESCs and iPSCs used in our study. We have discussed all these data in the Methods section. STR analysis across 8 different loci indicated that the H9 and H1 hESCs used in our study matched the reported STR profile of these cells (please see **Supplementary Table 3a**). These analyses also indicated no contamination with any other human cell lines. We also confirmed genetic identity of all the HD-iPSCs and two control iPSCs (#1 and #2) with the corresponding parental fibroblasts by STR analysis (please see **Supplementary Table 4**). Unfortunately, we did not have the parental fibroblasts of control iPSC line #3 (hFIB2-iPS4) for STR profile comparison. However, we believe that the use of this line as an additional control is justified because it has similar characteristics regarding proteostasis of HTT when compared with the other two control iPSC lines. For instance, we confirmed that control iPSCs #3 do not express polyQ-expanded HTT (**Supplementary Fig. 4a, 5b and 13c**) and do not accumulate polyQ aggregates upon proteasome inhibition or UBR5 knockdown (**Supplementary Fig. 1e and 17b**). As mentioned above, the control iPSCs #3 used in our experiments had a normal, diploid, male, chromosomal content as assessed by SNP analysis*

*We also confirmed that the MJD-iPSCs used in our study matches exactly the STR profile of the parental fibroblasts provided by the depositor of the lines. The STR profile is not presented in full to protect the donor's identity as requested by the provider of the cells (EBiSC consortium) (please see **Supplementary Table 3b**).*

*Since ALS-iPSCs #2 were raised from control iPSCs #4 by TALEN-directed mutagenesis, we confirmed that the STR profile of the ALS-iPSCs #2 used in our experiments matches with the profile of control iPSCs #4 (please see **Supplementary Table 4**).*

D. Suggested improvements: experiments, data for possible revision

1. The paper suggests that the discoveries for the role of UBR5 observed here in the iPSCs is also applicable to pluripotent stem cells in general, including hESCs. However, this could be exclusive to iPSCs and may be because of somatic cell reprogramming the associated stress during low efficiency reprogramming of somatic cells into clonal iPSCs.

Reviewer #2 is right and we cannot discard that some of our discoveries regarding the role of UBR5 are exclusive to iPSCs. Since most of our results were obtained from iPSCs, we focused our conclusions on these cells. We apologize if we did not make clear this point in our first submission. The discussion section now says: “With these results, we speculate that increased proteostasis of pluripotent stem cells may be required to avoid the generation of precursor cells that accumulate protein aggregates at early organismal stages, a process that

could be detrimental to organismal survival and healthspan. However, it is important to note that most of our studies were performed in iPSCs, and further evidence in hESCs as well as in vivo experiments using mouse models are necessary to assess this intriguing possibility”.

In addition, we have now included experiments in hESCs as a further control pluripotent stem cell line. We observed similar results when compared to control iPSCs. These data support our hypothesis that UBR5 could have a similar role in hESCs. Specifically, we have now included the following findings:

- UBR5 mRNA and protein levels decrease during differentiation of hESCs (**Supplementary Fig. 9**).
- Knockdown of UBR5 does not impair the mRNA levels of HTT in hESCs (**Supplementary Fig. 12a, b**).
- Knockdown of UBR5 results in increased protein levels of HTT in hESCs (**Supplementary Fig. 13a, b**).
- Knockdown of UBR5 results in accumulation of aggregates in hESCs (**Figure 7c**).
- Knockdown of UBR5 does not change the expression of pluripotency markers (**Supplementary Fig. 22c**).

2. A better description of the antibodies for the normal and expanded HTT is required in the methods section including the epitopes and validation of these antibodies.

We have now provided more information about total-HTT and polyQ-expanded protein antibodies in the methods section: “To detect HTT protein we used anti-HTT (Cell Signaling, ab#5656, 1:1000), a monoclonal antibody produced by immunizing animals with a synthetic peptide corresponding to residues surrounding Pro1220 of human HTT protein. To detect mutant HTT, we used anti-polyQ-expansion diseases marker (Millipore, MAB1574, 1:1000), a monoclonal antibody raised against TATA-binding protein that recognizes peptides overlapping the polyQ stretch of this protein. This antibody also recognizes polyQ-containing proteins such as HTT and ATXN3 with the remarkable property of detecting much better the polyQ-expanded pathological proteins than the wild type proteins^{16,38}”.

We have now performed validation of these antibodies in iPSCs (e.g., HTT knockdown and overexpression experiments, comparison of the distinct lines in the same blot). We have discussed these validation experiments in detail. The text now says: “With the strong link between proteasome dysfunction and mutant HTT aggregation, we asked whether HTT levels are regulated by the proteasome in HD-iPSCs. For this purpose, we characterized in our model two antibodies that recognize HTT and polyQ-expanded proteins, respectively^{16,38}. First, we validated that these antibodies detect endogenous levels of HTT in iPSCs (**Supplementary Fig. 4a-c**). The HD-iPSCs used in this study express one mutant allele of HTT but also one normal copy (**Supplementary Table 1**)^{16,31,39-41}. Since the length of the polyQ stretch diminishes the electrophoretic mobility of proteins³⁸, we could discriminate normal HTT and mutant HTT in both HD Q71 and Q180-iPSC lines by western blot using anti-HTT antibody (**Fig. 2a and Supplementary Fig. 4a**). In HD Q57-iPSCs, the differences in the electrophoretic motilities of both alleles were marginal and they were not efficiently separated on western blot assays (**Fig. 2a**). We also observed that anti-HTT-antibody was less immunoreactive to mutant HTT (**Fig. 2a and Supplementary Fig. 4d**). These differences were more pronounced in HD-iPSCs that express longer polyQ repeats (Q180), as mutant HTT was only detected after high exposure times in these cells (**Fig. 2a**). Thus, we used the antibody to polyQ-expanded proteins to examine the expression of mutant HTT in these cells

(Fig. 2a). We confirmed that this antibody only recognizes mutant HTT on western blots and the intensity of the signal correlates with the length of the polyQ expansion (Fig. 2a and Supplementary Fig. 4d), as previously reported³⁸. Although Q57 expansion was not detected with anti-polyQ-expanded proteins by western blot, this antibody strongly detected mutant HTT in Q71 and Q180 lines (Fig. 2a)”.

3. Figure legend 2C. Knock down of CCT6A and CCT7 does not appear to induce expression of endoderm markers at all. Rather it should say ectoderm. Please correct this. Also, it is rather curious as to why the knockdown does not affect mesoderm and endoderm expression in 2C. The authors need to explain clearly the rationale behind this?

Since we have not presented results on CCT6A and CCT7 in this manuscript, we believe that this comment must refer to our manuscript entitled “Somatic increase of CCT8 mimics proteostasis of human pluripotent stem cells and extends C. elegans lifespan” (Nature Communications, 2016). As such, it was already addressed in the aforementioned article.

4. Utilize isogenic mutation expansion (Q71 and Q180 alleles corrected in iPSCs by CRISPR) corrected iPSCs to validate these observations.

*We have now included experiments in two independent isogenic counterparts of the Q180-iPSC line (i.e., HD-C#1 and HD-C#2), in which the 180 CAG expansion was corrected to a non-pathological repeat length. These lines were a gift from M.A. Pouladi (Xu et al, Stem Cell Reports, 2017). We found that proteasome inhibition does not induce polyQ-expanded HTT aggregation in these cells (please see **Supplementary Fig. 2**). Moreover, we did not observe accumulation of polyQ aggregates in the corrected isogenic lines upon UBR5 knockdown (**Fig. 5b**). By western blot analysis, we assessed that these cells do not express polyQ-expanded HTT (**Supplementary Fig. 18**). Similar to control iPSC lines (#1-#3), loss of UBR5 induces the upregulation of normal HTT protein levels in corrected isogenic lines (**Supplementary Fig. 18**). Thus, these experiments validate the main conclusions of our manuscript (i.e., UBR5 regulates HTT levels in iPSCs, leading to polyQ-expanded HTT aggregation only in those iPSCs that express mutant HTT).*

Unfortunately, we could not obtain a corrected isogenic line of HD Q71-iPSCs. Since we have now included numerous controls to strengthen our conclusions, we think that these lines are not necessary for the main conclusions of our manuscript. As comprehensively reviewed by Tousley & Kegel-Gleason (Journal of Huntington’s disease, 2016), most of the current knowledge using iPSCs for HD modelling is based on the comparison of distinct HD lines with lines derived from healthy donors, lacking the use of control isogenic corrected lines. Indeed, in vitro phenotypes are predicted to be subtle and susceptible to effects of genetic background variations in the study of late age-onset disorders. Ultimately, the generation and use of isogenic control lines will facilitate the discovery of new phenotypes which have not been previously associated with disease and authenticate mutant HTT-specific effects.

However, it is important to note that here we focused on defining novel modulators of well-documented hallmarks of HD such as the accumulation of HTT aggregates induced by polyQ-expanded mutations. In other words, we did not focus on phenotypes induced by mutant HTT but rather on identifying modifiers of polyQ-expanded mutant HTT aggregation, one of the hallmarks of the disease. Using different methods (filter trap, immunofluorescence), we observed that proteasome inhibition and loss of UBR5 triggers the accumulation of polyQ-expanded HTT aggregates only in the patient-derived iPSCs that express polyQ-expanded

HTT. Besides corrected isogenic lines from HD Q180-iPSCs, we have now strengthened these conclusions by adding a third control iPSC line (control iPSC #3) as well as HD Q57-iPSCs.

We also show that proteasome inhibition or loss of UBR5 increases the levels of both normal HTT and polyQ-expanded HTT. The effects on normal HTT are now validated in 4 control iPSCs, 2 corrected isogenic lines of Q180, 2 hESC lines, 2 MJD-iPSCs lines, 2 ALS-iPSCs as well as HD-iPSCs. The effects on the protein levels of polyQ-expanded HTT can only be shown in HD patient-derived iPSCs and, thus, the corrected isogenic Q71 line is not essential for this conclusion.

Finally, we also report here that loss of UBR5 does not affect neural and neuronal differentiation from both control iPSCs and HD-iPSCs. Since we did not observe differences in this phenotype between control and HD-iPSCs, we believe that the analysis in 5 independent lines provides strong evidence that loss of UBR does not affect differentiation.

5. In general, the ICC images of poly Q aggregates of are not very high quality: Please acquire higher resolution images with greater magnification.

We apologize for the poor quality and low magnification of the immunocytochemistry experiments presented in our first submission. We have now repeated all the experiments and presented higher magnifications. For instance, please see **Fig. 1a and 5a**. These images now clearly show accumulation of polyQ-HTT aggregates in the distinct HD-iPSCs upon proteasome inhibition or UBR5 knockdown.

6. Figure 1d: It is not clear why the Q33 control iPSC line has more SDS-resistant poly Q aggregates in DMSO condition compared to the Q71 and Q71 HD iPSC lines. There should be minimal or no aggregates in the control iPSC line or at least equal to the HD iPSC lines in DMSO condition, even if UBR5 levels are high in iPSC stage. Is the filter trap assay an artifact? Also, how is the loading and intensity of the bands normalized in this assay?

Filter trap is a reliable method to examine protein aggregation of misfolded proteins, particularly polyQ-expanded HTT. As such, this method is often used by laboratories with a strong expertise in proteostasis (e.g., Behrends et al, *Molecular Cell*, 2006). We calculate protein concentration and load the same amount of protein for each sample included in a specific blot to allow for direct comparison between samples. Although it is not possible to add loading controls such as B-actin in western blots, our filter trap assays support the conclusions from immunocytochemistry experiments. In HD-iPSCs, proteasome inhibition and UBR5 knockdown trigger the accumulation of polyQ-expanded aggregates. In contrast, these treatments do not result in polyQ-expanded aggregation in control iPSCs.

We apologize because the replicate presented in Fig. 1d in our first submission made difficult to interpret the results. We have now presented data to show that the signal detected in control Q33- iPSCs corresponds to background signal (please see **Supplementary Fig. 3a**). We did not observe differences in the signal when we loaded higher amounts of total protein from the Q33 cell lysate, indicating that the weak signal detected for Q33 corresponds to background signal. Likewise, we did not observe an increase in the signal when we treated these cells with proteasome inhibition.

We also loaded different amounts of total protein for HD-iPSCs without proteasome inhibition and observe no changes, indicating that these cells do not accumulate aggregates under normal conditions as we confirmed by immunocytochemistry experiments (**Supplementary Fig. 3b**). In contrast, proteasome inhibition induces accumulation of polyQ-expanded aggregates in these cells. In MG-132-treated HD-iPSC samples, the levels of

detected polyQ-expanded aggregates correlate with the amount of total protein loaded, demonstrating that this signal is specific (**Supplementary Fig. 3b**). We have now provided this explanation in the legend of **Supplementary Figure 3**.

It is important to note that we are assessing aggregation of endogenous levels of HTT and this requires to load high amounts of total protein and relatively high exposure times to detect the specific signal under proteasome inhibition or UBR5 knockdown. Since we load the same high amounts of total protein for control iPSCs and naive HD-iPSCs in the same blot, the unspecific signal background is high under the exposure times used for these assays. In contrast, this issue is not present in the models that overexpress polyQ-expanded proteins (for instance, see **Fig. 9d and 10a**).

As the signal observed in Q33 corresponds to background and the unspecific signal varies between experiments, we have replaced former Fig. 1d with a new replicate (please see **Fig. 1c**), including more iPSC lines in the same blot for direct comparison.

7. All Western blots, please quantify the bands by densitometry and normalize to b-actin loading.

We really appreciate this suggestion. We have now quantified the western blots presented in our manuscript. These quantifications were really helpful to present our data in a more clear manner and strengthen our conclusions.

8. Throughout the manuscript, the authors pick and choose which HD iPSC line and control iPSC line to use in experiments. The authors keep randomly switching between Q71 and Q180 as well as different clones of these iPSCs for different experiments and figures. Please be consistent and include data from the Q71 and Q180 lines in all experiments.

We apologize for this. To reach solid conclusions, we have now conducted all the experiments with control iPSCs #1 (Q21), control iPSCs #2 (Q33), HD Q71-iPSCs #1, HD Q71-iPSCs #2 and HD Q180-iPSCs. For the most important experiments, we have now added a third independent control iPSC line as well as HD Q57-iPSCs.

9. Figure 1e: Why does MG132 treatment increase levels of both normal and mHTT protein?

The results presented in our first submission indicated that the proteasome determines the levels of both wild-type and mutant HTT in iPSCs. Likewise, our data suggested that UBR5 is key for the proteostatic regulation of normal and aggregation-prone HTT. Although we discussed the role of the proteasome/UBR5 in the modulation of both normal and mutant HTT, we apologize if we did not make clear this point in our first submission. We have now presented several experiments that strengthen our conclusions. For instance, we have now included western blots of three independent control iPSCs (please see **Fig. 2b** and **Supplementary Fig. 5a, b**), indicating a strong increase in wild-type HTT upon proteasome inhibition. Likewise, we have now included experiments in an additional control iPSC line as well as two independent human embryonic stem cell lines (hESCs), showing an increase in wild-type HTT upon UBR5 knockdown (**Supplementary Fig. 13a-c**). Moreover, our experiments in HD-iPSCs lines (including new analysis of additional independent HD-iPSC lines) support a strong increase of both wild-type and mutant HTT upon proteasome inhibition and UBR5 knockdown (i.e., **Fig. 4** and **Supplementary Fig. 5, 13**). In addition, we have now presented quantifications of all these western blot experiments that clearly show a similar up-regulation of normal HTT when compared with mutant HTT in HD-iPSCs (e.g., **Fig. 2c, d** and **Fig. 4d, e**). Finally, we

have performed co-immunoprecipitation experiments with anti-UBR5 antibody in two control iPSCs and one additional HD-iPSC line (**Supplementary Fig. 16**). These experiments further validated the interaction of UBR5 with both normal and mutant HTT. However, only the up-regulation of mutant HTT results in polyQ-expanded aggregates.

10. What was the rationale for ignoring the TRIM71 gene from Supplementary Table 1? This was the highest upregulated E3 ubiquitin ligase with 17-fold up regulation in hESCs compared to neurons. Why UBR5 chosen for the studies when it was relatively lower fold up-regulation.

*We have now explained in more detail why we focused on UBR5. The text now says: “Notably, UBR5 was one of the most up-regulated E3 enzymes (**Supplementary Table 2**). UBR5 shows a striking preference for Lys48 linkages of ubiquitin⁴³, which is the primary signal for proteasomal degradation²⁴. Under proteotoxic stress, UBR5 cooperates with Lys11-specific ligases to produce K11/K48 heterotypic chains, promoting proteasomal clearance of misfolded nascent polypeptides⁴³. Recently, a genome-wide association analysis has identified that genetic variations in a chromosome 8 region containing UBR5 gene hasten the clinical onset of HD⁴⁴. Remarkably, loss of UBR5 reduces the modification of overexpressed HTT protein with K11/K48-linked ubiquitin chains in a human cell line⁴³. Thus, increased endogenous expression of UBR5 could provide a link between proteostasis and regulation of HTT levels in pluripotent stem cells”.*

*Reviewer #2 is right and the 17-fold increase in the levels of TRIM71 compared to neurons cannot be ignored. We have now examined whether TRIM71 increases HTT levels using two independent shRNAs (please see **Supplementary Fig. 15f, g**). It is important to note that we had already tested the levels of HTT protein upon knockdown of other up-regulated E3 enzymes in our first submission. The text now says: “Besides UBR5, other E3 ligases are also increased in pluripotent stem cells (**Supplementary Table 2 and Supplementary Fig. 15a**). We knocked-down four of these up-regulated enzymes (i.e., UBE3A, RNF181, UBR7, TRIM71) and found no differences in the mutant and normal HTT levels of HD-iPSCs (**Supplementary Fig. 15b-g**)”.*

11. It is a bit annoying that there was no consistency in the differentiated states of neural cell types evaluated in the manuscript. In some figures iPSCs and NPCs were evaluated, while in other neurons were included and then sometimes terminally differentiated iPSC-neurons were ignored for analysis.

*We apologize for not including terminally differentiated neurons in these assays in our first submission and we have now corrected this. We confirmed that UBR5 protein levels decrease during neuronal differentiation of control iPSCs #1 (Q21), control iPSCs #2 (Q33), HD Q71-iPSCs #1, HD Q71-iPSCs #2 and HD Q180-iPSCs (**Fig. 3a-c and Supplementary Fig. 10**). We have also added the comparison with terminally differentiated neurons of UBR5 mRNA levels (please see **Fig. 3d-f and Supplementary Fig. 9b**). Moreover, we have now examined whether loss of UBR5 at the iPSC level affects neuronal induction of these cells (both control and HD iPSCs) by assessing percentage of GABA-positive cells and the expression levels of distinct striatal neuronal markers (**Supplementary Fig. 29a-g**). We also examined whether terminally differentiated neurons derived from HD-iPSCs with downregulated UBR5 levels accumulate polyQ-expanded aggregates (**Supplementary Fig. 30**). Moreover, we tested whether knockdown of UBR5 in differentiated neurons impairs HTT levels and polyQ-expanded HTT aggregation (**Supplementary Fig. 32-33**). Finally, we*

determined whether UBR5 downregulation results in cell death of neuronal cultures (**Supplementary Fig. 34**).

12. Figure 2b: Why is data from terminally differentiated neurons missing here? Please include.

*We have now added the comparison with terminally differentiated neurons of UBR5 mRNA levels (please see **Fig. 3d-f and Supplementary Fig. 9b**).*

13. Figure 2g: I don't think that the authors can directly conclude from this experiment that "UBR5 is involved in proteasomal degradation of HTT".

*Former Fig. 2g containing analysis of HD Q71-iPSCs is now presented as **Fig. 4d**. To further examine the potential link between proteasome and UBR5 on HTT regulation, we have now performed this experiment in control iPSCs #1 (Q21), control iPSCs #2 (Q33) and HD Q180-iPSCs (please see **Fig. 4b-e**). The text now says: "Since loss of UBR5 did not impair HTT mRNA levels (**Fig. 4a and Supplementary Fig. 12**), our data supported a role of this E3 enzyme in post-translational regulation of HTT. To examine whether UBR5 modulates the levels of HTT in a proteasome-dependent manner, we blocked proteasomal degradation in iPSCs. Notably, UBR5-knockdown and MG-132-treated cells exhibited similar levels of HTT (**Fig. 4b-e**). Most importantly, UBR5 downregulation did not further increase the levels of HTT in both control and HD-iPSCs with reduced proteasome activity (**Fig. 4b-e**), indicating that UBR5 could be involved in proteasomal degradation of HTT". Although these results strongly support that UBR5 is involved in proteasome degradation of UBR5, please note that we have now toned down our conclusions in this section. We have also replaced the former heading of this section 'Loss of UBR5 impairs proteasomal degradation of HTT in iPSCs' by 'UBR5 determines HTT levels preventing its aggregation in HD-iPSCs'.*

*The direct role of UBR5 in proteasomal degradation of HTT is later demonstrated in **Fig. 10**. In these experiments, we found that UBR5 overexpression is sufficient to reduce Q100-HTT levels (**Fig. 10c**). However, the treatment with proteasome inhibitor blocked the reduction in Q100-HTT levels induced by UBR5 overexpression (**Fig. 10d**). These results are strong evidence to conclude that UBR5 regulates proteasomal degradation of HTT. Moreover, we showed that UBR5 overexpression induces a strong Lys48-polyubiquitination of HTT. Lys48 linkages of ubiquitin are the primary signal for proteasomal degradation overexpression (**Fig. 10e**).*

14. It would be nice see consistency in the cell types evaluated.

As explained in point #11, we have now presented experiment with consistency in the cell types evaluated.

15. What time point of differentiation of iPSC-neurons was chosen for the analysis shown here?

We performed the experiments between day 32-35 of the neuronal differentiation protocol. We have now provided this important information in the methods section (striatal neuron differentiation).

16. What is the difference between HD-iPSC #1 and #2, both with Q71 repeats?

The HD Q71-iPSC line #1 was generated using retroviral induction of cMyc, Klf4, Oct4 and Sox2 by Prof. Daley's laboratory. From the same parental fibroblast, the HD-iPSCs consortium generated the HD Q71-iPSC line #2 via episomal expression of lMyc, Klf4, Oct4, Sox2 and LIN28 reprogramming factors. We have now indicated these differences in the Methods section.

17. In the discussion, please state why do the authors think the role of UBR5 protein is specific to regulating homeostasis of HTT protein only and not any other protein with repeat poly-Q expansions?

We have now examined whether UBR5 regulates other aggregation-prone proteins. In particular, we have performed the following experiments:

- *Analysis of ATXN3 levels and aggregation upon proteasome inhibition and UBR5 knockdown in iPSCs derived from two distinct patients diagnosed with Machado-Joseph disease (MJD). Please see **Fig. 6c-g** and **Supplementary Fig. 21**. Although loss of UBR5 increased HTT levels in MJD-iPSCs, we did not find changes in either normal or mutant ATXN3. Moreover, UBR5 downregulation did not induce aggregation of polyQ-expanded ATXN3. This phenotype differed from HD-iPSCs lines, where up-regulation of mHTT levels upon UBR5 knockdown was sufficient to trigger polyQ-expanded HTT aggregation. Thus, our data indicate that not all polyQ-containing proteins are modulated by UBR5.*
- *We have now examined whether UBR5 regulates protein levels or aggregation of wild-type FUS as well as two distinct mutant FUS variants linked with amyotrophic lateral sclerosis (ALS). Remarkably, loss of UBR5 did not stimulate RNA-stress granules formation or FUS intracellular delocalization in control or ALS-iPSC lines (**Fig. 7a**). In addition, UBR5 knockdown did not impair wild-type or mutant FUS levels whereas the levels of HTT were up-regulated in these cells (**Fig. 7b**). Thus, our results support a specific role of UBR5 on HTT regulation. Although UBR5 was dispensable for ATXN3 and FUS proteostasis, we have indicated that we cannot discard a role of UBR5 in the control of other aggregation-prone proteins associated with disease.*
- *Besides its role in HTT modulation, we asked whether UBR5 also determines the global proteostasis ability of human pluripotent cells. The text now says: "Under normal conditions, misfolded proteins are refolded by chaperones or terminated via proteolytic systems¹³. Metabolic and environmental conditions (e.g., heat stress) challenge the structure of proteins, increasing the load of misfolded and damaged proteins. When proteolytic systems are overwhelmed, misfolded proteins accumulate into aggresomes. In these lines, we observed that heat stress induces the accumulation of aggresomes in pluripotent stem cells despite their increased proteolytic ability (**Fig. 7c**). Notably, loss of UBR5 was sufficient to induce the accumulation of aggresomes in these cells (**Fig. 7c**). Thus, our data suggest that UBR5 not only regulates the levels of specific proteins such as HTT, but it is also involved in the degradation of misfolded proteins ensued from normal metabolism".*
- *Since UBR5 overexpression was sufficient to ameliorate aggregation of polyQ100-HTT in human cell models, we have now examined whether ectopic expression of*

this E3 can reduce aggregation of other misfolded proteins. For this purpose, we generated a human cell model that expresses an aggregation-prone β -amyloid protein (i.e., beta23 construct provided by F.U Hartl (Olzscha et al, Cell, 2011)). We found that ectopic expression of UBR5 does not reduce aggregation of these β -amyloid fibrils.

*The discussion section now says: “Notably, genetic variations in a region of the chromosome 8 that contains the UBR5 gene, among others (i.e., RRM2B, MIR5680, NCALD), have been associated with an early onset of HD by genome-wide association analysis⁴⁴. Here we show that knockdown of UBR5 hastens the deleterious changes induced by polyQ-expanded expression in *C. elegans* models, providing direct evidence of a link between UBR5 activity and polyQ-expanded aggregation with age. It is important to note that these *C. elegans* models express polyQ-expanded fused to YFP but not HTT protein. Thus, these results suggest that UBR5 could also have a role in the proteostasis of other polyQ-containing proteins related with disease. However, we found that UBR5 knockdown does not impair polyQ-expanded ATXN3 levels and its aggregation in MJD-iPSCs. Thus, we conclude that not all polyQ-containing proteins are regulated via UBR5 activity. In these lines, we have observed that UBR5 does not impinge upon cellular localization and aggregation of FUS variants linked with ALS. Although these results suggest specificity of UBR5 for HTT regulation, we cannot discard a role in the proteostasis of other aggregation-prone proteins associated with disease. Notably, UBR5 downregulation is sufficient to induce the accumulation of aggresomes in control hESCs/iPSCs. Since aggresomes form from the accumulation of misfolded proteins when proteolytic systems are overwhelmed, UBR5 may also be involved in the degradation of misfolded proteins ensued from normal metabolism”.*

18. Figure 3b: Filter trap experiment results are not consistent with Figure 1d. The NT shRNA results for the 3 lines are not consistent with the DMSO results Fig. 1D for the different iPSCs.

*As we explained in point #6, the signal observed in control iPSC lines corresponds to background signal. Likewise, the signal observed in HD-iPSCs treated with DMSO or NT shRNA is background. Thus, we cannot draw conclusions from this signal specially to compare between blots of independent experiments. Figure 3b is now presented in **Fig. 5**. Since we loaded an independent blot for each line (the experiments of the distinct lines were collected at different days), we have now made this clear by separating the figure in different panels. Results from HD Q71-iPSCs #1 and HD-Q180 iPSCs are shown in **Fig. 5c** and **Fig. 5e**, respectively. Data from control iPSCs #1 (Q21) are presented in **Fig. 5f**. We have now included data from HD Q71-iPSCs #2 (**Fig. 5d**) and control iPSCs #2 (Q33) (**Fig. 5g**). These results show that loss of UBR5 induces aggregation of mutant HTT in HD-iPSCs lines, as we also confirmed by immunocytochemistry experiments. In contrast, we did not observe specific signal corresponding to polyQ-expanded aggregates in control iPSCs lines upon UBR5 knockdown. We have also added a filter trap assay in **Fig. 5k**, which presents different control and HD-iPSCs under proteasome inhibition and UBR5 knockdown loaded in the same blot for direct comparison.*

19. Clarify the statistics as stated below.

We have now clarified the statistics as explained below in Section E.

20. Figure 3e: Please label Figure 3e appropriately to clarify samples with MG132 treatment and those without.

*We apologize for this mistake. The label indicating MG-132 treatment was somehow lost when we first submitted our manuscript and we have now corrected this (please see **Fig. 5k, l**).*

21. Please incorporate allelic series of CAG repeat control and HD iPSC lines – Q21, Q33, Q71 and Q180 in all experiments.

As explained in Section C, we have now examined control iPSCs #1 (Q21), control iPSCs #2 (Q33), HD Q71-iPSCs and HD Q180-iPSCs in each analysis. For the most important experiments, we have now added a third independent control iPSC line as well as HD Q57-iPSCs.

22. Figure 4c: Please include data from differentiated striatal neurons as well.

*We have now examined whether loss of UBR5 at the iPSC level affects neuronal induction of both control and HD iPSCs by assessing percentage of GABA-positive cells and the expression levels of distinct striatal neuronal markers (**Supplementary Fig. 29a-g**).*

23. Figure 4c: UBR shRNA#2 condition Merge panel is incorrect and has not merged channels. Please rectify.

*We apologize for this mistake and we have now included the Merge panel for UBR5 shRNA#2 condition (please see **Supplementary Fig. 25**).*

24. Figure 4f: Please include staining with the expanded poly Q specific antibody as well.

*This figure is now presented as **Fig. 8g** (HD Q71-NPCs #1). We have now included staining with the expanded-polyQ specific antibody. We also present data of other lines using both anti-HTT and expanded-polyQ-antibody: Control NPCs #1 (**Supplementary Fig. 28a**), Control NPCs #2 (**Supplementary Fig. 28b**), HD Q71-NPCs #1 (**Supplementary Fig. 28c**) and HD Q180-NPCs (**Fig. 8h**)*

25. Figure 5: Please include validation of striatal neuron differentiation and cultures using western blot and ICC co-staining with specific markers of striatal neurons such as DARPP32. In fact, throughout the manuscript the striatal neuron data is very important to the conclusions of the manuscript.

*As mentioned above, we have now included specific markers of striatal neurons for the different experiments showed in this revision. **Fig. 29** presents analysis of striatal markers by immunocytochemistry and qPCR of neurons derived from UBR5 shRNA-iPSCs. **Fig. 31** presents co-staining with MAP2, DARPP32 and expanded-polyQ antibody in neurons treated with proteasome inhibitor. Likewise, **Fig. 32** presents co-staining with MAP2, DARPP32 and*

expanded-polyQ antibody in neurons upon UBR5 knockdown. **Fig. 33** shows western blots with HTT, expanded-polyQ and DARPP32 antibodies.

26. Figure 6b: Filter trap results are not convincing. Figure 6a is convincing. Again, there are inconsistencies between 6a and 6b.

*The indicated Figure 6a and 6b are now presented as **Fig. 10a** and **10b**, respectively. **Fig. 10a** shows that wild-type UBR5 overexpression reduces polyQ-expanded HTT aggregation. In contrast, overexpression of a catalytic inactive UBR5 mutant did not diminish polyQ-expanded HTT aggregation (**Fig. 10a**). Although the replicate presented in **Fig. 10b** may not be as striking as **Fig. 10a**, both blots show the same phenotype (i.e., wild-type UBR5 overexpression reduces polyQ-expanded HTT aggregation whereas catalytic inactive UBR5 mutant does not decrease mutant HTT aggregation). Moreover, **Fig. 10b** convincingly shows that proteasome inhibition blocks the amelioration of polyQ-expanded aggregation induced by wild-type UBR5 overexpression.*

27. Figure 6d: β -actin loading in MG132 treatment samples is not consistent with DMSO control. It is likely that those samples are over-loaded 2-3x over DMSO control based on band intensity. Thus, the treatments are not directly comparable.

*Reviewer #2 is right and the β -actin loading of the replicates shown in former Figure 6d was uneven, making difficult to draw solid conclusions from these data. We have now performed additional replicate experiments (please see **Fig. 10d**). In these experiments, the levels of β -actin among samples were similar, allowing for better comparison. In addition, our results are now further strengthened by quantification of HTT signal normalized to β -actin loading control.*

28. Figure 6d: It appears that MG132 treatment appears to decrease Q100 HTT in UBR overexpressing samples compared to the empty vector. Please explain this observation?

*As explained above, we have now performed additional replicate experiments with equal levels of β -actin among samples, allowing for better comparison (**Fig. 10d**). We have also added quantification of HTT signal normalized to β -actin loading control, leading to the following conclusions: 1) overexpression of wild-type UBR5 reduces Q100-HTT levels, 2) overexpression of catalytic inactive UBR5 mutant does not decrease Q100-HTT levels and 3) proteasome inhibition blocks the degradation of Q100-HTT induced by overexpression of wild-type UBR5.*

29. Figure 6e: Results text does not match the figure. Should there be MG132 treatment as "+" in all the samples. Please explain this figure better in the text and figure legend. The HTT antibody blot does not appear to match the Figure 6a result.

*This figure is now presented as **Fig. 10e**. We have now made more clear in the figure that all the samples were treated with MG-132 proteasome inhibitor. We apologize for not making clear these results in the main text and figure legend in our first submission. The main text now says: "To further assess the link between UBR5 and HTT regulation, we performed*

immunoprecipitation experiments and examined polyubiquitination of HTT. Prior immunoprecipitation, we treated the cells with proteasome inhibitor to block the degradation of HTT induced by UBR5. Under these conditions, we immunoprecipitated similar amounts of HTT in cells overexpressing wild-type UBR5 when compared with cells expressing empty vector or catalytic inactive UBR5 (**Fig. 10e**). Notably, we found that ectopic expression of UBR5 induces a dramatic increase in polyubiquitination of mutant HTT (**Fig. 10e**). In contrast, overexpression of the catalytic inactive UBR5 mutant did not promote polyubiquitination of Q100-HTT protein (**Fig. 10e**). The figure legend now says: “**e**, Immunoprecipitation with anti-HTT and anti-FLAG antibodies in Q100-HTT(OE) HEK293. Immunoprecipitation was followed by western blot with antibodies to HTT and polyubiquitinated proteins (polyUb) to detect immunoprecipitated total HTT protein and polyUb-HTT, respectively. Prior immunoprecipitation, cells were treated with proteasome inhibitor (0.5 μ M MG-132, 16 h) to block the degradation of HTT induced by UBR5 so we could immunoprecipitate similar amounts of HTT for direct comparison of polyubiquitination among the distinct conditions. The images are representative of three independent experiments”.

In former Figure 6a (now **Fig. 10a**), we presented filter trap results in cells without proteasome inhibitor treatment. In support of a role of UBR5 in the degradation of HTT, we observed that UBR5 overexpression reduces polyQ-expanded HTT aggregation. In **Fig. 10b**, we presented filter trap assays of cells treated with proteasome inhibition. In these cells, HTT degradation induced by UBR5 overexpression is blocked by the proteasome inhibitor treatment (**Fig. 10d**) and the accumulation of aggregates is higher when compared with UBR5(OE) cells without MG132 treatment (**Fig. 10b**). Accordingly, the HTT antibody blot presented in **Fig. 10e** matches with the results observed in cells treated with proteasome inhibitor in **Fig. 10b** and **10d**.

30. There is also a lack of in-lab HD-iPSC pluripotency genetic integrity data over multiple passages, which should be included in supplementary figures. Even though iPSCs may be procured from another lab, basic iPSC QC within the lab is required to ensure stable cytogenetics. G-band karyotype (cytogenetic stability) and STR identity analysis is required on all the iPSC lines to ensure that the UBR5 observation is not because of an abnormal karyotype of the cell lines.

As explained in more detail above, we have now confirmed that all the hESC and iPSC lines used in our experiments had a normal diploid karyotype as assessed by single nucleotide polymorphism (SNP) genotyping (please see **Supplementary Fig. 38**). We have also performed STR analysis to confirm the identity of the hESCs and iPSCs used in our study (**Supplementary Tables 3 and 4**). These quality control assays indicate that the UBR5 observation is not because of an abnormal karyotype of the cell lines.

E. Appropriate use of statistics and treatment of uncertainties

This was one of the major weaknesses in this manuscript. Then n's are mentioned in each figure legend, however, it is not clearly described where the n's are derived from? Are these true biological and experimental replicates or only technical replicates? Is it independent experiments with multiple biological replicates? The statistics should be run on minimum of 3 exclusively performed independent experiments (each experiment containing multiple biological and technical replicates). Minimum of three biological replicates are required in 3

independent iPSC lines in 1 experiment. Then the assay and phenotype data should be averaged for the group. Then each assay must be repeated a minimum of 3 times (for every iPSC line) in the same experiment. Then the data should be averaged and compared across the groups. Multiple wells in 1 experiment do not qualify as independent experiments. Given the significant variability in differentiation of iPSC-derivatives from experiments a minimum of 3 experimental replicates are required and data averaged over those experiments.

We apologize for not clearly describing where the n's are derived from in our first submission. In the corresponding figure legends, we have now made clear that mean \pm s.e.m and statistical analyses are calculated from n independent experiments. For each figure, we performed at least three independent experiments. Each independent experiment contains multiple biological and technical replicates. In each independent experiment, biological replicates/wells were averaged for every condition. Then, the data of different independent experiments (at least 3) were averaged. Finally, we compared this average across conditions/groups. As mentioned above, we have now used iPSCs from different HD patient donors (polyQ57, polyQ71 and polyQ180) and three different control iPSCs from different donors.

F. References: appropriate credit to previous work?

References have been appropriately cited except for some of the citations mentioned in this review.

As explained in section B, we have now discussed this relevant work in our manuscript.

G. Clarity and context: lucidity of abstract/summary, appropriateness of abstract, introduction and conclusions

The paper is apprehensible and well-written including the abstract. Introduction is sound, although a clearer description of how proteostasis changes during somatic reprogramming to iPSCs should be included. The conclusions and their interpretations can be better understood after the suggested corrections in figures and additional data, inclusion of isogenic mutant correct iPSC controls as well as other additional experiments requested.

We have now added a clearer description of how proteostasis changes during somatic reprogramming to iPSCs. We have now addressed all of the reviewer's comments and experimental suggestions. The manuscript is now more clear and the conclusions are supported with the additional experiments and controls requested by the reviewers.

Reviewer #3:

The manuscript by Koyuncu et al. examines the role of UBR5 in HTT and polyQ HTT degradation and aggregation in stem cells (iPSC) and differentiated NPCs.

Main findings of the study:

pluripotent iPSC has increased level of UBR5, which may modulate levels of mutant HTT.

Downregulation of UBR5 in *C. elegans* Huntington's Disease model increases the aggregation of HTT.

Overall, the findings regarding UBR5 are interesting. On one hand, the idea that it plays a role in heightened damage surveillance in stem cells is interesting and somewhat convincing. More experiments would be needed to convincingly demonstrate its quality control role in stem cells. On the other hand, the connection to HTT seems forced, and so far appears tenuous. 1. looking at UBR5 and HTT in complete isolation is biased and is reminiscent of dozens of other studies pitting any number of E3 ligases against HTT, without convincing physiological proof of a direct mechanistic relationship. 2. relatively little effort is made to show that UBR5 in fact ubiquitinates HTT, as opposed to exerting some kind of indirect effect in the KD experiment, whereby the KD significantly increases the misfolded protein load, thereby also affecting HTT.

Major concerns:

Proteasome activity determines the level of HTT and other proteins, inhibition of proteasome would result in protein accumulation that can indirectly trigger HTT aggregation. Or just be explained by increased concentration of HTT, not necessarily the effect of proteasome on HTT. The authors should compare it to autophagy inhibition.

We agree with Reviewer #3 that we had to provide further evidence to discard indirect effects of proteasome inhibition and UBR5 knockdown in the proteostasis of HTT. As Reviewer #3 indicates, proteostasis collapse induced by proteasome inhibition could induce mutant HTT aggregation. Alternatively, the proteostasis collapse induced from proteasome inhibition could indirectly result in high levels of mutant HTT that eventually triggers its aggregation. To assess these hypothesis, we have first performed autophagy inhibition experiments as suggested by Reviewer #3. Although autophagy inhibition induced mutant HTT aggregation in HD Q71-iPSCs, these aggregates were less compact and the percentage of aggregate-containing cells were lower when compared to proteasome inhibition treatment (Fig. 2e, f and Supplementary Fig. 7a, b). In HD Q180-iPSCs, autophagy inhibition only induced aggregation in a low percentage of cells in contrast to the high population of cells containing aggregates observed upon proteasome inhibition (Fig. 2e, f). Whereas proteasome inhibition increased HTT levels, autophagy downregulation resulted in decreased amounts of HTT protein (Fig. 2g and Supplementary Fig. 8a-c). This decreased in HTT levels was not associated to potential changes in cell viability or a compensatory up-regulation of proteasome, as we did not find significant changes in these parameters upon autophagy inhibition (Supplementary Fig. 8d-e). With these differences between autophagy and proteasome inhibition, our data indicate a direct role of the proteasome in HTT degradation in iPSCs. In these lines, proteasome dysfunction results in increased levels of both normal HTT and aggregation-prone HTT. Since it is known that the impairment of clearance of misfolded or mutant proteins is key to their accumulation, the increase in mutant HTT levels upon proteasome inhibition could contribute to diminish the ability of HD-iPSCs to suppress mutant

HTT aggregation. However, these data cannot discard that the proteostasis collapse induced by proteasome inhibition also contributes to polyQ-expanded HTT aggregation.

Besides the role of the proteasome in mutant HTT aggregation, we also tested whether this proteolytic system regulates proteostasis of polyQ-expanded ATXN3 in iPSCs derived from two distinct patients diagnosed with Machado-Joseph disease (MJD). As previously reported, these MJD-iPSCs did not accumulate polyQ-expanded aggregates (**Fig. 6d**). To examine whether the UPS regulates proteostasis of ATXN3 in iPSCs, we downregulated proteasome activity by using MG-132 proteasome inhibitor. In contrast with HD-iPSC lines, the treatment with proteasome inhibitor for 12 h induced acute cell death and detachment of MJD-iPSCs. To reduce these effects, we performed our analysis at an earlier time point of the treatment (6 h) (**Supplementary Fig. 21**). Notably, proteasome inhibition did not change the levels of normal and mutant ATXN3, whereas the amounts of HTT were up-regulated in these cells (**Fig. 6c**). Thus, our results suggest that not all polyQ-containing proteins are modulated by the proteasome in iPSCs, providing further evidence of specific effects on HTT degradation. On the contrary, proteasome inhibition triggered polyQ-expanded ATXN3 aggregation (**Fig. 6d**), a process that could be linked with the global proteostasis collapse induced by this treatment. These results indicate that both the specific impairment of HTT levels as well as indirect effects induced by proteasome inhibition could contribute to the accumulation of polyQ-expanded HTT aggregates in HD-iPSCs. Nevertheless, the specific requirements of UBR5 for proteostasis of HTT are demonstrated by our experiments in MJD-iPSC lines. Although loss of UBR5 increased HTT levels in MJD-iPSCs, we did not find changes in either normal or mutant ATXN3 (**Fig. 6e, f**). In contrast to global proteasome inhibition, UBR5 downregulation did not induce aggregation of polyQ-expanded ATXN3 (**Fig. 6g**). This phenotype differed from HD-iPSCs lines, where up-regulation of mHTT levels upon UBR5 knockdown was sufficient to trigger polyQ-expanded HTT aggregation (**Fig. 5**).

To further assess the specific effects of UBR5 in the modulation of HTT levels, we have now examined whether UBR5 regulates protein levels or aggregation of wild-type FUS as well as two distinct mutant FUS variants linked with amyotrophic lateral sclerosis (ALS). Remarkably, loss of UBR5 did not stimulate RNA-stress granules formation or FUS intracellular delocalization in control or ALS-iPSC lines (**Fig. 7a**). In addition, UBR5 knockdown did not impair wild-type or mutant FUS levels whereas the levels of HTT were up-regulated in these cells (**Fig. 7b**). Since UBR5 overexpression was sufficient to ameliorate aggregation of polyQ100-HTT in human cell models, we have now examined whether ectopic expression of this E3 can reduce aggregation of other misfolded proteins. For this purpose, we generated a human cell model that expresses an aggregation-prone β -amyloid protein (i.e., beta23 construct provided by F.U Hartl (Olzscha et al, Cell, 2011)). We found that ectopic expression of UBR5 does not reduce aggregation of these β -amyloid fibrils. Thus, we conclude that not all aggregation-prone proteins are modulated by UBR5, suggesting specific effects on HTT degradation. Although UBR5 was dispensable for the proteostasis of ATXN3, FUS and aggregation-prone β -amyloid polypeptide, we have indicated in the main text that we cannot discard a role of UBR5 in the control of other aggregation-prone proteins associated with disease.

Besides its role in the proteostasis of aggregation-prone proteins, we asked whether UBR5 also determines the global proteostasis ability of human pluripotent cells. The text now says: "Under normal conditions, misfolded proteins are refolded by chaperones or terminated via proteolytic systems¹³. Metabolic and environmental conditions (e.g., heat stress) challenge the structure of proteins, increasing the load of misfolded and damaged proteins. When proteolytic systems are overwhelmed, misfolded proteins accumulate into aggresomes. In

these lines, we observed that heat stress induces the accumulation of aggresomes in pluripotent stem cells despite their increased proteolytic ability (**Fig. 7c**). Notably, loss of UBR5 was sufficient to induce the accumulation of aggresomes in these cells (**Fig. 7c**). Thus, our data suggest that UBR5 not only regulates the levels of specific proteins such as HTT, but it is also involved in the degradation of misfolded proteins ensued from normal metabolism”.

Finally, it is important to note that during the revision of this manuscript a collaboration between the laboratories of Matsumoto, Dixit and Rape (Yau et al, Cell, 2017) reported that under proteotoxic stress, UBR5 cooperates with Lys11-specific ligases to produce K11/K48 heterotypic chains, promoting proteasomal clearance of misfolded nascent polypeptides. In support of our work, they also found that loss of UBR5 reduces the modification of overexpressed HTT protein with K11/K48-linked ubiquitin chains in HeLa cell line. We have now discussed these important findings in our manuscript.

There are many ways to verify a direct relationship between E3 ligase and substrate, from mass spec, to lysine mutation.

We have now performed mass spec experiments to identify potential lysine sites ubiquitinated by UBR5. We have also discussed the findings regarding polyubiquitinated sites of HTT from Yau et al, Cell, 2017. The text now says: “In support of our findings, a recent study reported that UBR5 downregulation decreases the modification of overexpressed HTT protein with K11/K48-linked polyubiquitin chains in HeLa human cells⁴⁵. Moreover, this study revealed K11/K48-linked polyubiquitin modifications at Lys337 of HTT by proteomics experiments⁴⁵. We performed proteomics analysis of immunoprecipitated HTT in Q100-HTT overexpressing cells to determine potential lysine sites modified by UBR5 (**Supplementary Fig. 37a, b**). Although we did not detect ubiquitination at Lys337 of HTT in our assays, we identified two other ubiquitinated lysine sites: Lys631 and Lys2097. However, only Lys631 shows a small but significant higher ubiquitination in cells overexpressing wild-type UBR5 when compared to catalytic inactive UBR5 (**Supplementary Fig. 37**), suggesting that this site could be ubiquitinated by UBR5”.

It would be interesting to see comparison of mRNA levels of HTT between iPSC and differentiated cells, since UBR5 plays role in the transcriptional repression.

We have now performed this comparison of mRNA levels of HTT between iPSCs and differentiated cells (please see **Supplementary Fig. 11**). The text now says: “Since UBR5 promotes microRNA-mediated transcript destabilization in mouse ESCs⁴⁵ and may be involved in transcriptional repression during development⁴⁶, we examined whether UBR5 downregulation during differentiation correlates with increased HTT mRNA levels. However, the amounts of HTT mRNA were either downregulated or not significantly changed with iPSC differentiation (**Supplementary Fig. 11**). As a more formal test, we assessed whether knockdown of UBR5 alters the transcript levels of HTT in distinct iPSC/hESC lines and found no differences (**Fig. 4a and Supplementary Fig. 12**)”.

Minor concerns:

Although the authors show that UBR5 KD doesn't prevent differentiations it is important to note that it may still affect differentiation pathways, for example Shh which was also used in this

paper to differentiate striatal neurons (<http://journals.plos.org/plosone/article?id=10.1371/journal.pone.0157079>). Did the author find any difference between differentiated striatal neurons with or without UBR5?

We agree with Reviewer #3 that we need to discuss this paper as well as Buckley et al (Cell Stem Cell, 2012), another relevant paper for the role of UBR5 in pluripotency. The discussion section now says: "Whereas our results highlight the importance of UBR5 in proteostasis of iPSCs, the intrinsic high levels of this enzyme could also suggest a role in maintenance of pluripotency via the ubiquitination of endogenous substrates. Interestingly, a study has reported that RNAi-mediated knockdown of Ubr5 results in a significant loss of pluripotency markers in control mouse ESCs²⁰. However, a different study did not find impairment of pluripotency markers in murine ESCs upon UBR5 RNAi-treatment⁷⁴. Likewise, we did not find changes in the expression of pluripotency or germ layer markers in control hESCs/iPSCs upon loss of UBR5. Interestingly, knockdown of Ubr5 reduces the induction of Sonic hedgehog (Shh) in murine pluripotent stem cells upon retinoic-acid treatment⁷⁴. In these lines, a conditional Ubr5 mutant mouse presents decreased hedgehog signaling around embryonic day 13.5⁷⁴. Although this model exhibits shorter limbs when compared to controls, the differences were not significant and the mutants do not have other obvious morphological defects⁷⁴. Since our striatal neuronal differentiation protocol is based on the induction of hedgehog signaling pathway, we tested whether UBR5 is necessary for the generation of MSNs from human iPSCs. However, loss of UBR5 did not affect their differentiation into MSNs, indicating that these cells conserved their ability to induce the hedgehog signaling. The distinct impact of UBR5 in pluripotency and differentiation could be associated to genetic differences between mouse and human species. Moreover, it is important to note the distinct pluripotent states exhibited by murine ESCs and hESCs/iPSCs⁵. Mouse ESCs are cultured in the presence of serum and leukaemia inhibitory factor (LIF), and exhibit a naive state resembling the pluripotent state observed in the inner cell mass of the pre-implantation embryo⁵. On the other hand, hESCs as well as human iPSCs obtained by direct in vitro reprogramming do not require LIF signaling and exhibit a more primed state that resembles post-implantation embryonic configurations⁵".

*We have now examined whether loss of UBR5 at the iPSC level affects neuronal induction of these cells (both control and HD iPSCs) by assessing percentage of GABA-positive cells and the expression levels of distinct striatal neuronal markers. We did not observe differences in their ability to differentiate into striatal neurons (**Supplementary Fig. 29a-g**). The text now says: "One step further was to determine whether these cells are able to generate terminally differentiated neurons. Since GABAergic medium spiny neurons (MSNs) undergo the greatest neurodegeneration in HD⁶⁴, we differentiated iPSCs into striatal neurons⁶⁵. Among those cells expressing the neuronal marker MAP2, ~30-60% also expressed GABA depending on the cell line (**Supplementary Fig. 29a, b**). As previously reported³¹, these differences were not associated with the expression of mutant HTT (**Supplementary Fig. 29a, b**). Knockdown of UBR5 at the iPSC stage did not reduce their ability to differentiate into striatal neurons (**Supplementary Fig. 29a-g**). In contrast to NPCs, terminally differentiated neurons derived from HD-iPSCs with downregulated UBR5 levels did not accumulate polyQ-expanded aggregates (**Supplementary Fig. 30**). Thus, the lack of defects in neurogenesis and aggregates in these cells indicate that other mechanisms activate during neuronal differentiation to facilitate proteostasis of mutant HTT".*

It is not clear why the authors used HEK293 model for the ectopic expression, it would be interesting to demonstrate it using differentiated iPSC model.

We completely agree with Reviewer #3 and we have put great efforts to perform these analysis. However, these experiments were not possible due to technical reasons. Neurons derived from HD-iPSC do not accumulate polyQ-expanded HTT aggregates even after the addition of proteasome and autophagy inhibitors or the induction of oxidative stress in neurons (as demonstrated in this study and other publications: Noormohammadi et al, Nat. Comm., 2016; HD-iPSC Consortium, Cell Stem Cell, 2012; Jeon et al, Stem Cells, 2012; Victor et al, Nat. Neuroscience, 2018). Thus, we could not assess the impact of UBR5 overexpression in reducing mutant HTT aggregates in these cells. We hypothesized that we could force the accumulation of polyQ-expanded HTT aggregates by overexpressing Q100-HTT protein in neurons. However, HTT is a large protein of over 300 kDa. Since lentiviral systems only allow for the insertion of small and mid-size proteins, we were not able to clone Q100-HTT in lentiviral plasmids for strong expression in neurons. Then, we tried to directly transfect Q100-HTT constructs using distinct methods such as Fugene liposomal-based transfection system and Nucleofector electroporation-based transfection system, which are also very efficient in post-mitotic cells. Unfortunately, we were not able to overexpress Q100-HTT protein due to the low transfection efficiency of this large construct. Nevertheless, we also tried to overexpress UBR5 in neurons derived from HD-iPSCs. As HTT, UBR5 is a large protein (309 kDa) and all our attempts to overexpress the protein failed.

Alternatively, we tried to perform UBR5 overexpression experiments in fibroblasts from HD patients using Fugene and Nucleofector systems, as these cells proliferate and we expected to obtain better transfection efficiency. However, the large size of the constructs also resulted in failed transfection and overexpression of UBR5 in these primary cells.

Thus, we focused on the HEK293 models for these assays, as they showed high efficiency of transfections allowing for simultaneous overexpression of Q100-HTT and UBR5 proteins.

it would be helpful to see additional, better resolved versions of fig6e.

*We apologize because the resolution of this figure was somehow lost in our first submission. We have now added an image with better resolution and lower exposure time for better comparison of differences. These data in now presented in **Fig. 10e**. We have also included an additional replicate in **Supplementary Fig. 37a** that validates increased polyubiquitination of HTT upon wild-type UBR5 overexpression when compared to catalytic inactive UBR5 mutant of the samples analyzed by proteomics for identification of ubiquitination sites.*

Reviewer #4:

The manuscript by Koyuncu et. al. titled, "The ubiquitin ligase UBR5 suppresses proteostasis collapse in immortal pluripotent stem cells and Huntington's disease models" asks an important scientific question of significant interest to the field. Namely, the authors ask whether iPS cells

rely on the E3 ubiquitin ligase UBR5 to regulate polyQ-expanded mutant Htt protein levels and aggregation. This is an especially enticing question because stem cells seem to be resistant to proteostasis decline (and polyQ length-dependent Htt aggregation) in a manner that was, until now, not well understood.

The authors found that UBR5 is naturally upregulated in iPSCs and that the knockdown of its expression significantly increased the amount of Htt protein aggregation, both in iPSCs and in *C. elegans* models of polyQ aggregation/toxicity. Ultimately, they nicely demonstrate that UBR5 is a “central modulator” of mHtt aggregation.

The authors’ arguments would be even more convincing if they would address the following:

1. In figure 1e and f, the authors show a western blot using antibodies against Htt. They find that mutant Htt is expressed at lower levels than normal huntingtin. It would be helpful if the authors explained how they are certain that their antibody is equally immunoreactive to mutant Htt and to normal Htt protein.

We are really thankful to Reviewer #4 for this concern, as we were mistaken in our conclusion regarding the expression of mutant HTT. We have now performed extensive validation of the antibodies that clearly shows that anti-HTT antibody is less immunoreactive to mutant HTT (Supplementary Fig. 4). On the other hand, the antibody to polyQ-expanded proteins only recognizes mutant HTT on western blots and the intensity of the signal correlates with the length of the polyQ expansion (Fig. 2a and Supplementary Fig. 4). The text now says: “For this purpose, we characterized in our model two antibodies that recognize HTT and polyQ-expanded proteins, respectively^{16,38}. First, we validated that these antibodies detect endogenous levels of HTT in iPSCs (Supplementary Fig. 4a-c). The HD-iPSCs used in this study express one mutant allele of HTT but also one normal copy (Supplementary Table 1)^{16,31,39-41}. Since the length of the polyQ stretch diminishes the electrophoretic mobility of proteins³⁸, we could discriminate normal HTT and mutant HTT in both HD Q71 and Q180-iPSC lines by western blot using anti-HTT antibody (Fig. 2a and Supplementary Fig. 4a). In HD Q57-iPSCs, the differences in the electrophoretic motilities of both alleles were marginal and they were not efficiently separated on western blot assays (Fig. 2a). We also observed that anti-HTT-antibody was less immunoreactive to mutant HTT (Fig. 2a and Supplementary Fig. 4d). These differences were more pronounced in HD-iPSCs that express longer polyQ repeats (Q180), as mutant HTT was only detected after high exposure times in these cells (Fig. 2a). Thus, we used the antibody to polyQ-expanded proteins to examine the expression of mutant HTT in these cells (Fig. 2a). We confirmed that this antibody only recognizes mutant HTT on western blots and the intensity of the signal correlates with the length of the polyQ expansion (Fig. 2a and Supplementary Fig. 4d), as previously reported³⁸. Although Q57 expansion was not detected with anti-polyQ-expanded proteins by western blot, this antibody strongly detected mutant HTT in Q71 and Q180 lines (Fig. 2a)”.

2. In figure 1d, the authors show a filter trap assay to represent the amount of Htt aggregates present in different iPS cells lines in the presence or absence of the proteasome inhibitor MG132. It would be helpful if the authors provided an explanation for the relatively high number of Q33 aggregates observed in untreated cells.

We apologize because the replicate presented in Fig. 1d in our first submission made difficult to interpret the results. We have now presented data to show that the signal detected in control Q33- iPSCs corresponds to background signal (please see **Supplementary Fig. 3a**). We did not observe differences in the signal when we loaded higher amounts of total protein from the Q33 cell lysate, indicating that the weak signal detected for Q33 corresponds to background signal. Likewise, we did not observe an increase in the signal when we treated these cells with proteasome inhibition.

We also loaded different amounts of total protein for HD-iPSCs without proteasome inhibition and observe no changes, indicating that these cells do not accumulate aggregates under normal conditions as we confirmed by immunocytochemistry experiments (**Supplementary Fig. 3b**). In contrast, proteasome inhibition induces accumulation of polyQ-expanded aggregates in these cells. In MG-132-treated HD-iPSC samples, the levels of detected polyQ-expanded aggregates correlate with the amount of total protein loaded, demonstrating that this signal is specific (**Supplementary Fig. 3b**). We have now provided this explanation in the legend of **Supplementary Figure 3**.

It is important to note that we are assessing aggregation of endogenous levels of HTT and this requires to load high amounts of total protein and relatively high exposure times to detect the specific signal under proteasome inhibition or UBR5 knockdown. Since we load the same high amounts of total protein for control iPSCs and naive HD-iPSCs in the same blot, the unspecific signal background is high under the exposure times used for these assays. In contrast, this issue is not present in the models that overexpress polyQ-expanded proteins (for instance, see **Fig. 9d and 10a**).

As the signal observed in Q33 corresponds to background and the unspecific signal varies between experiments, we have replaced former Fig. 1d with a new replicate (please see **Fig. 1c**), including more iPSC lines in the same blot for direct comparison.

3. In Figure 5C-E, the authors demonstrate the effect of UBR5 knockdown on *C. elegans* models of polyQ aggregation/toxicity. Specifically, they demonstrate that knockdown results in an increase in polyQ67 protein levels, aggregation, and toxicity.

a) The model that they used expresses a polyQ peptide fused to YFP for visualization. The authors should clarify that outside of the polyQ region, this model expresses no Htt protein. They should also discuss the implications of this. Do they think that UBR5 acts on polyQ alone, irrespective of the protein context in which the polyQ is found? Would this mean that it acts on all polyQ proteins, including other disease-associated proteins?

We completely agree with Reviewer #4 and we have now clarified in the main text that outside the polyQ region, this model expresses no HTT protein. The text now says: "Remarkably, a RNAi screen against E3 ubiquitin ligases found that knockdown of the worm UBR5 orthologue (*ubr-5*) accelerates paralysis in a *Caenorhabditis elegans* model expressing 35 polyQ-repeats fused to yellow fluorescent protein (YFP) in body wall muscle cells⁶⁷. To assess the requirement of *ubr-5* for resistance to polyQ neurotoxicity, we examined a *C. elegans* model that expresses polyQ-expanded YFP in the nervous system⁶⁸".

We have also discussed the implications of this. First, it is important to note that we have now performed analysis of polyQ-expanded ATXN3 levels and aggregation upon proteasome inhibition and UBR5 knockdown in iPSCs derived from two distinct patients diagnosed with Machado-Joseph disease (MJD). Please see **Fig. 6c-g** and **Supplementary**

Fig. 21. *Although loss of UBR5 increased HTT levels in MJD-iPSCs, we did not find changes in either normal or mutant ATXN3. Moreover, UBR5 downregulation did not induce aggregation of polyQ-expanded ATXN3. This phenotype differed from HD-iPSCs lines, where up-regulation of mHTT levels upon UBR5 knockdown was sufficient to trigger polyQ-expanded HTT aggregation. Thus, our data indicate that not all polyQ-containing proteins are modulated by UBR5. The Discussion section now says: “Here we show that knockdown of UBR5 hastens the deleterious changes induced by polyQ-expanded expression in C. elegans models, providing direct evidence of a link between UBR5 activity and polyQ-expanded aggregation with age. It is important to note that these C. elegans models express polyQ-expanded fused to YFP but not HTT protein. Thus, these results suggest that UBR5 could also have a role in the proteostasis of other polyQ-containing proteins related with disease. However, we found that UBR5 knockdown does not impair polyQ-expanded ATXN3 levels and its aggregation in MJD-iPSCs. Thus, we conclude that not all polyQ-containing proteins are regulated via UBR5 activity”.*

b) Because these animals express polyQ67 fused to YFP, the authors could easily show aggregation in fixed (or live) animals in addition to doing filter trap assays. In the original study describing this C. elegans model for aggregation/toxicity, differences in aggregation propensity were reported for different neuronal subtypes. By examining whole animals, the authors could demonstrate whether UBR5 depletion has a greater effect in certain neurons or if the effect on polyQ67 aggregation is universal.

*We have now included images of Q67 worms upon *ubr-5* knockdown (please see **Fig. 9c and Supplementary Fig. 35**). The text now says: “Knockdown of *ubr-5* had a strong effect in the aggregation propensity of head neurons, particularly in the circumpharyngeal nerve ring although chemosensory processes also presented increased aggregation (**Fig. 9c**). However, we did not observe dramatic differences in propensity aggregation of commissural neurons of the animal mid-body (**Fig. 9c and Supplementary Fig. 35**)”.*

c) The authors used thrashing assays as a measure of toxicity. Their n-number was relatively low (18-20) and I suggest increasing it. Nonetheless, the statistics indicate a significant increase in Q67 toxicity when UBR5 gene expression is knocked down. I am somewhat surprised that the effect was as strong as it was, given that C. elegans neurons are famously resistant to RNAi. In fact, mutant strains that are hypersensitive to RNAi are typically used in studies requiring RNAi-mediated knockdown of neuronal gene expression. If the authors introduced an *rrf-3* mutation, for example, they might see a stronger and more reliable effect of UBR5 RNAi in neurons. Perhaps they would then see more of an effect on Q19 as well.

*Reviewer #4 is absolutely right, and a higher n-number is needed to reach conclusions. We have now repeated the experiment to analyse more worms (over 60 worms per condition) and obtained similar results (please see **Fig. 9e**).*

*As reviewer #4 indicates, neurons are less sensitive to RNAi (and even completely refractory for RNAis to specific genes). We have now introduced the *rrf-3* mutation in Q19 and Q67 worm models. The text now says: “Although *ubr-5* RNAi was sufficient to induce a pronounced increase of polyQ-expanded aggregates (**Fig. 9b**), it is important to note that neurons are less sensitive to RNAi when compared with other tissues⁶⁹. For this reason, we introduced a *rrf-3* mutation in the polyQ-YFP neuronal models, which confers hypersensitivity*

to RNAi in all the tissues, including neurons⁷⁰. Accordingly, we found that *ubr-5* RNAi treatment induced a strong increase in polyQ67-YFP aggregation of *rrf-3* mutants (**Fig. 9d**). On the contrary, *ubr-5* knockdown did not induce aggregation of polyQ19-peptides, even in the RNAi-hypersensitive mutant strain (**Fig. 9d**)”.

d) Several RNAi screens have been conducted in *C. elegans* to identify genes whose knockdown causes a measureable increase or decrease in polyQ protein aggregation. The authors should mention whether UBR5 was previously uncovered as a proteostasis regulator in any of those screens.

We have now discussed in the main text the following study: “Remarkably, a RNAi screen against E3 ubiquitin ligases found that knockdown of the worm UBR5 orthologue (ubr-5) accelerates paralysis in a Caenorhabditis elegans model expressing 35 polyQ-repeats fused to yellow fluorescent protein (YFP) in body wall muscle cells⁶⁷”.

Overall, I feel that this is an important study that elevates UBR5 to a prominent position within the proteostasis network. In fact, at least in stem cells, UBR5 seems to play a central role in downregulating Htt protein levels and thus preventing (or delaying) Htt aggregation in differentiated cells. After the authors address the above comments, I would strongly recommend the manuscript for publication.

Reviewers' comments:

Reviewer #1 (Remarks to the Author):

The revision has added more data to support the idea that UBR5 is a central modulator of mutant HTT aggregation in iPSCs. However, the key issue of whether UBR5 is associated with HD pathology remains unclear. Although the expression level of UBR5 is dramatically reduced in differentiated neurons, there is no solid association between UBR5 and huntingtin accumulation in differentiated stem cells. The authors cited a few papers for the fact that inhibition of the UPS did not elicit huntingtin toxicity and accumulation in differentiated cells. Thus, the cultured cell system does not mimic the accumulation of mutant huntingtin in the brain, raising a significant concern about the relevance of using iPSCs to study the toxicity and accumulation of mutant huntingtin.

Indeed, by performing additional experiments, the authors could not find that suppression of UBR5 is able to increase the level of mutant huntingtin or its aggregates. It is well known that mutant huntingtin accumulates and forms aggregates in mature neurons in the brain. Although the data that inhibition of UBR5 increases huntingtin accumulation in HD-iPSCs is strong, it does not necessarily mean that UBR5 plays a key role in this event. Any interference with the UPS activity could trigger huntingtin accumulation in cultured cells, which has been demonstrated by many studies using different cell lines. A more important issue is whether UBR5 plays a critical role in huntingtin accumulation in mature neurons. This issue has not been addressed in the manuscript. Also, why UBR5 is more likely to influence mutant huntingtin, but not other misfolded proteins, has not been addressed. These unclear issues have weakened the conclusion of this paper that UBR5 is a central component to suppress huntingtin accumulation.

Reviewer #2 (Remarks to the Author):

I was glad that the authors acted upon the reviewer suggestions to improve the manuscript substantially. I am really impressed by the detailed author response to the reviewer's comments along with the additional experiments and the inclusion of new data. I am satisfied with all the responses and new data. I recommend accepting the manuscript for publication.

Reviewer #3 (Remarks to the Author):

I am impressed by the additional experiments that the authors performed. I think that they substantially improved the manuscript.

Reviewer #4 (Remarks to the Author):

Manuscript #NCOMMS-17-24191A is a revised manuscript by Koyuncu et. al. titled, "The ubiquitin ligase UBR5 suppresses proteostasis collapse in immortal pluripotent stem cells and Huntington's disease models." It asks an important scientific question of significant interest to the field. Namely, the authors ask whether iPS cells rely on the E3 ubiquitin ligase UBR5 to regulate polyQ-expanded mutant Htt protein levels and aggregation. This is an especially enticing question because stem cells seem to be resistant to proteostasis decline (and polyQ length-dependent Htt aggregation) in a manner that was, until now, not well understood.

The authors found that UBR5 is naturally upregulated in iPSCs and that the knockdown of its expression significantly increased the amount of Htt protein aggregation, both in iPSCs and in *C. elegans* models of polyQ aggregation/toxicity. Ultimately, they nicely demonstrate that UBR5 is a "central modulator" of mHtt aggregation.

The authors have done a very nice job of addressing the comments of four peer reviewers. The result is a much-improved manuscript that I look forward to seeing in print. However, I would first like to see the authors address the following with respect to Fig. 9:

- Fig. 9C. The fluorescent micrographs of polyQ-YFP *C. elegans* were obtained on a compound fluorescence microscope. These images would be greatly improved with the use of confocal microscopy. Image-quality would be improved, but more than that, confocal microscopy would allow the authors to utilize FRAP to determine whether the observed foci are immobile protein aggregates. The authors show aggregation with filter trap assays, but a live-imaging/biophysical approach such as FRAP would be a nice parallel experiment to validate those findings.
- Fig. 9C. The DIC images indicate that the imaged animals are caught in an air bubble and the bodies are not intact. This is not optimal and thus should be repeated with healthy animals in an aqueous environment.
- Fig. 9D. These filter trap assays seem to show that there is less Q67 aggregation in *rrf-3* mutant animals grown on control RNAi as compared to wild type animals grown on control RNAi. The authors should indicate why this is the case. Is it consistent or was in an artifact of uneven loading?

Reviewer #1:

The revision has added more data to support the idea that UBR5 is a central modulator of mutant HTT aggregation in iPSCs. However, the key issue of whether UBR5 is associated with HD pathology remains unclear. Although the expression level of UBR5 is dramatically reduced in differentiated neurons, there is no solid association between UBR5 and huntingtin accumulation in differentiated stem cells. The authors cited a few papers for the fact that inhibition of the UPS did not elicit huntingtin toxicity and accumulation in differentiated cells. Thus, the cultured cell system does not mimic the accumulation of mutant huntingtin in the brain, raising a significant concern about the relevance of using iPSCs to study the toxicity and accumulation of mutant huntingtin.

Indeed, by performing additional experiments, the authors could not find that suppression of UBR5 is able to increase the level of mutant huntingtin or its aggregates. It is well known that mutant huntingtin accumulates and forms aggregates in mature neurons in the brain. Although the data that inhibition of UBR5 increases huntingtin accumulation in HD-iPSCs is strong, it does not necessarily mean that UBR5 plays a key role in this event. Any interference with the UPS activity could trigger huntingtin accumulation in cultured cells, which has been demonstrated by many studies using different cell lines. A more important issue is whether UBR5 plays a critical role in huntingtin accumulation in mature neurons. This issue has not been addressed in the manuscript. Also, why UBR5 is more likely to influence mutant huntingtin, but not other misfolded proteins, has not been addressed. These unclear issues have weakened the conclusion of this paper that UBR5 is a central component to suppress huntingtin accumulation.

We agree with Reviewer #1 that modeling Huntington's disease as well as other neurodegenerative disorders using patient-specific neurons is challenging, as neurons differentiated from induced pluripotent stem cells lack aggregates and strong cell death phenotype. Moreover, HD iPSCs-derived neurons do not exhibit mutant HTT aggregates even after the addition of cellular stressors. Of particular relevance for our study, proteasome inhibitors do not induce accumulation of aggregates in these cells. Besides citing the most relevant papers for the lack of aggregates upon proteasome inhibition, we also analysed changes in HTT protein levels in differentiated neurons and found no differences. Likewise, loss of UBR5 did not impair HTT levels and aggregation in iPSCs-derived neurons. Although our data in C. elegans models and HTT-overexpressing human cell lines (Fig. 9-10) support a role of UBR5 in the suppression of polyQ-expanded aggregation, we agree with Reviewer #1 that the relevance of UBR5 in HD pathology remains unclear due to the lack of HD phenotypes in iPSCs-derived neurons. We have now discussed this in more detail in the introduction and results sections.

In these lines, we have now made more clear in the text that our main focus is to study the regulation of mutant HTT in HD-iPSCs and this may not have relevance to neurons in a disease context. The title and abstract have been adapted accordingly. In this regard, as Reviewer #1 indicates, we provided more experiments in the first revision supporting a role of UBR5 as a central modulator of mutant HTT aggregation in iPSCs. First, we added data strengthening a direct role of the proteasome in HTT degradation in iPSCs to regulate mutant HTT aggregation (e.g., differences observed between autophagy and proteasome inhibition treatment showed that alterations in HTT levels could not only be explained by a global

proteostasis collapse). Most importantly, our data support a specific role of UBR5 in the degradation of mutant HTT via the UPS without affecting proteasome activity levels (**Fig. 4f and Supplementary Fig. 14**). This conclusion was strengthened by multiple experiments in our first revision. For instance, we knock-downed other components of the UPS (distinct E3 and E2 enzymes) and found no changes in HTT levels or aggregation (**Supplementary Fig. 15b-g, Fig. 6a, b and Supplementary Fig. 20**). Moreover, we observed that UBR5 does not affect the levels and aggregation of mutant ATXN3 or FUS (**Fig. 6c-g and Fig. 7a-b**). In addition, we presented data showing the interaction of UBR5 with HTT in multiple iPSC lines (**Fig. 4g and Supplementary Fig. 16**). Importantly, we could also observe changes in the ubiquitination of HTT via UBR5 modulation (**Fig. 10e and Supplementary Fig. 39a, b**). We have now performed co-immunoprecipitation experiments that indicate that UBR5 does not interact with mutant ATXN3 or FUS proteins in MJD and ALS-iPSCs (**Supplementary Fig. 22a-c**). However, UBR5 interacts with HTT in these cells (**Supplementary Fig. 22a-c**). These results provide a potential explanation of why UBR5 is more likely to influence mutant HTT, but not other misfolded proteins. Taken together, we believe our results are important to define super-vigilant proteostasis of pluripotent stem cells and are relevant for stem cell and proteostasis research.

Reviewer #2:

I was glad that the authors acted upon the reviewer suggestions to improve the manuscript substantially. I am really impressed by the detailed author response to the reviewer's comments along with the additional experiments and the inclusion of new data. I am satisfied with all the responses and new data. I recommend accepting the manuscript for publication.

We thank Reviewer #2 for recommending publication and for the thoughtful comments that led us to substantially improve our manuscript.

Reviewer #3:

I am impressed by the additional experiments that the authors performed. I think that they substantially improved the manuscript.

We would like to thank again Reviewer #3 for the thoughtful comments that were critical to improve our manuscript.

Reviewer #4:

Manuscript #NCOMMS-17-24191A is a revised manuscript by Koyuncu et. al. titled, "The ubiquitin ligase UBR5 suppresses proteostasis collapse in immortal pluripotent stem cells and Huntington's disease models." It asks an important scientific question of significant interest to the field. Namely, the authors ask whether iPSCs rely on the E3 ubiquitin ligase UBR5 to regulate polyQ-expanded mutant Htt protein levels and aggregation. This is an especially

enticing question because stem cells seem to be resistant to proteostasis decline (and polyQ length-dependent Htt aggregation) in a manner that was, until now, not well understood.

The authors found that UBR5 is naturally upregulated in iPSCs and that the knockdown of its expression significantly increased the amount of Htt protein aggregation, both in iPSCs and in *C. elegans* models of polyQ aggregation/toxicity. Ultimately, they nicely demonstrate that UBR5 is a “central modulator” of mHtt aggregation.

The authors have done a very nice job of addressing the comments of four peer reviewers. The result is a much-improved manuscript that I look forward to seeing in print. However, I would first like to see the authors address the following with respect to Fig. 9:

- Fig. 9C. The fluorescent micrographs of polyQ-YFP *C. elegans* were obtained on a compound fluorescence microscope. These images would be greatly improved with the use of confocal microscopy. Image-quality would be improved, but more than that, confocal microscopy would allow the authors to utilize FRAP to determine whether the observed foci are immobile protein aggregates. The authors show aggregation with filter trap assays, but a live-imaging/biophysical approach such as FRAP would be a nice parallel experiment to validate those findings.

We have now obtained the images using a confocal microscope. Reviewer #4 was completely right and the images were greatly improved showing more clear differences between control and ubr-5 RNAi treated worms in both head and mid-body neurons.

*We have also performed FRAP experiments. The text now says: “To assess whether the polyQ67-YFP foci were immobile protein aggregates, we performed a quantitative fluorescence recovery after photobleaching (FRAP) analysis of head neurons. In both empty vector and ubr-5 RNAi-treated worms, most of the polyQ67-YFP foci signal could not be recovered, indicating an immobile state (**Supplementary Fig. 37a**). However, ubr-5 RNAi induced a faster incorporation of new polyQ67-YFP peptides into the aggregates (**Supplementary Fig. 37b**)”.*

- Fig. 9C. The DIC images indicate that the imaged animals are caught in an air bubble and the bodies are not intact. This is not optimal and thus should be repeated with healthy animals in an aqueous environment.

Reviewer #4 is completely right. We have now immobilized the worms using the protocol described by Kim et al (Long-Term Imaging of Caenorhabditis elegans Using Nanoparticle-Mediated Immobilization, PLOS One, 2013). Briefly, worms were placed on 5% agarose-containing pads on a suspension of polystyrene beads (Polysciene, 2.5% by volume). This method does not expose the worm to toxic substances, and allows recovery of animals after immobilization.

- Fig. 9D. These filter trap assays seem to show that there is less Q67 aggregation in rrf-3 mutant animals grown on control RNAi as compared to wild type animals grown on control RNAi. The authors should indicate why this is the case. Is it consistent or was in an artifact of uneven loading?

We thank the Reviewer for raising this issue. Although the three biological experiments of the same experiment presented in our first revision showed less Q67 aggregation in rrf-3 mutant worms on control RNAi compared to wild-type animals on control RNAi, other independent experiment did not show these differences. We have now performed three new independent experiments and do not see differences between rrf-3 mutant worms on control RNAi compared to wild-type animals on control RNAi. We have now presented a more representative experiment in Fig. 9d.

REVIEWERS' COMMENTS:

Reviewer #4 (Remarks to the Author):

The authors have adequately addressed my concerns regarding Figure 9, resulting in higher-quality data. I recommend publication.